# Graph Your Own Prompt

**Xi Ding[1,3],***    **Lei Wang[1,2],***    **Piotr Koniusz[2,4,3,1]**,    **Yongsheng Gao[1],†**

[1]Griffith University, [2]Data61/CSIRO,
[3]Australian National University, [4]University of New South Wales

## Abstract

We propose Graph Consistency Regularization (GCR), a novel framework that injects relational graph structures, derived from model predictions, into the learning process to promote class-aware, semantically meaningful feature representations. Functioning as a form of *self-prompting*, GCR enables the model to refine its internal structure using its own outputs. While deep networks learn rich representations, these often capture noisy inter-class similarities that contradict the model's predicted semantics. GCR addresses this issue by introducing parameter-free *Graph Consistency Layers* (GCLs) at arbitrary depths. Each GCL builds a batch-level feature similarity graph and aligns it with a global, class-aware masked prediction graph, derived by modulating softmax prediction similarities with intra-class indicators. This alignment enforces that feature-level relationships reflect class-consistent prediction behavior, acting as a *semantic regularizer* throughout the network. Unlike prior work, GCR introduces a multi-layer, cross-space graph alignment mechanism with adaptive weighting, where layer importance is learned from graph discrepancy magnitudes. This allows the model to prioritize semantically reliable layers and suppress noisy ones, enhancing feature quality without modifying the architecture or training procedure. GCR is model-agnostic, lightweight, and improves semantic structure across various networks and datasets. Experiments show that GCR promotes cleaner feature structure, stronger intra-class cohesion, and improved generalization, offering a new perspective on learning from prediction structure. [**Project website**] [**Code**]

## 1   Introduction

Deep neural networks have achieved impressive success across a wide range of classification tasks, from natural images to medical diagnostics and scene understanding [88, 117, 111, 23, 26, 27, 28]. A key factor underlying this success is the ability of deep models to learn rich internal representations. However, despite their effectiveness, these representations are often noisy, entangled, and lack clear alignment with semantic class boundaries. This discrepancy can lead to feature spaces where samples from different classes remain closely aligned, undermining generalization and limiting the interpretability and robustness of learned models.

Intuitively, if a network is confident that two inputs belong to the same class, as indicated by their softmax predictions, then their intermediate representations should also reflect this similarity. Conversely, samples predicted to belong to different classes should ideally be separated in feature space. Yet, existing architectures and loss functions typically do not enforce such consistency between feature- and prediction-level relationships [90, 106, 108]. While contrastive learning and graph-based regularizers have emerged to structure feature spaces, they often require explicit positive/negative sampling or operate at a single level of abstraction, leaving much of the internal network dynamics unregulated [105, 64, 81].

---

*Equal contribution (co-first authors).

†Corresponding author: yongsheng.gao@griffith.edu.au

39th Conference on Neural Information Processing Systems (NeurIPS 2025).

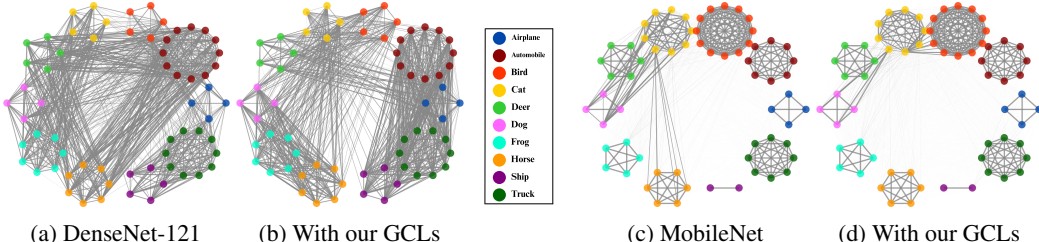

| | |
|---|---|
| (a) DenseNet-121 | (b) With our GCLs |
| (c) MobileNet | (d) With our GCLs |

Figure 1: Relational graph visualization using a batch of 64 samples from CIFAR-10 on (left) DenseNet-121 and (right) MobileNet. We compare the baselines with their counterparts augmented by our GCLs. Our method promotes richer, class-aware semantic representations by acting as a form of *self-prompting*. For DenseNet-121, the baseline feature relational graph tends to connect samples based on superficial visual similarity (*e.g.*, deer, horse, and automobile), often ignoring semantic boundaries. In contrast, our GCL-enhanced model produces more *semantically coherent groupings*, clearly separating animals from vehicles. On MobileNet, the prediction relational graph further highlights the strength of our method, demonstrating cleaner, more distinct class relationships compared to the baseline. These improvements reflect the effectiveness of our model in aligning feature and prediction spaces with semantic structure, despite being *lightweight and parameter-free*.

In this work, we propose a new form of structural supervision that enforces relational graph consistency between features and predictions throughout the network. We introduce Graph Consistency Regularization (GCR), a lightweight and general-purpose method that inserts Graph Consistency Layers (GCLs) after arbitrary network blocks/layers. Each GCL builds a batch-level feature relational graph using pairwise relations. These feature graphs are aligned with a class-aware masked prediction graph that encodes semantic similarity based on softmax outputs and label-derived intra-class indicators. The core idea is to treat the prediction graph as a reference structure, and guide the network to shape its internal feature geometry to be consistent with semantically meaningful predictions.

GCR is architecture-agnostic, lightweight, and introduces no new parameters. Importantly, it provides multi-layer supervision by aligning feature-prediction structures at multiple depths, using dynamically learned layer-wise weights. This acts as a regularization signal that sharpens class boundaries, suppresses noisy inter-class affinities, and encourages feature representations that reflect class semantics more faithfully (Fig. 1 provides a comparison). Our **contributions** are summarized as follows:

i. We are the first to use dynamic, batch-level relational graphs to guide the learning process using class-aware prediction structures, enabling a new form of semantic regularization.

ii. We introduce Graph Consistency Layers (GCLs), lightweight, parameter-free modules that can be flexibly inserted into existing networks, enabling multi-scale structural supervision with adaptively learned layer-wise weighting based on graph discrepancy.

iii. We propose Graph Consistency Regularization (GCR), a novel framework that enforces alignment between intermediate feature graphs and global, class-aware prediction graphs, promoting semantic coherence and representation consistency throughout the network.

We validate GCR across diverse architectures and benchmarks, achieving consistent accuracy gains and stronger generalization without altering backbone design or training protocol. Below, we review related work.

## 2  Related Work

**Graph-based representation learning.** Graph-based methods [54, 16] have been widely adopted to model relationships among samples, particularly in non-Euclidean domains such as social networks or molecules [1, 53, 121]. In classification tasks, graphs have been used to capture instance-level similarities for label propagation [113, 74], semi-supervised learning [115, 114], and contrastive representation learning [20, 91]. For example, some methods build graphs over entire datasets [101] or memory banks [9, 3] to encourage feature consistency via graph Laplacian regularization [80, 4] or message passing [38, 79]. Unlike these approaches, which require maintaining global graphs or rely

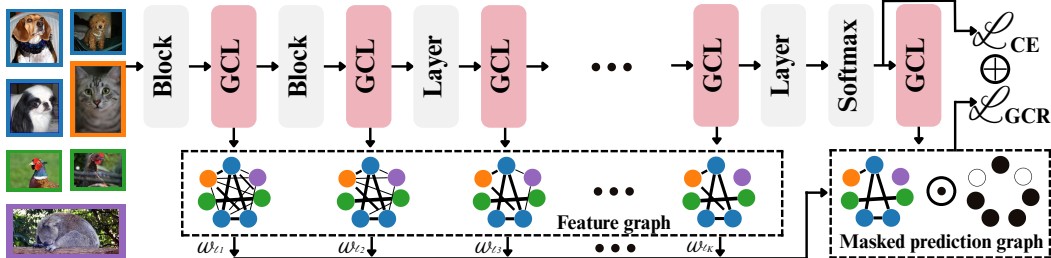

Figure 2: Our *parameter-free* Graph Consistency Layer (GCL), highlighted in red, can be inserted after any micro-network block (*e.g.*, Inception) or specific layer (*e.g.*, fully connected). Each GCL constructs a relational graph from batch-level features using a similarity metric (*e.g.*, cosine). A reference graph is generated from softmax predictions and masked by intra-class indicators: binary masks identifying semantically consistent pairs. Each GCL enforces alignment between masked prediction graph and the feature-level graphs. The resulting consistency signals are adaptively weighted, forming the Graph Consistency Regularization (GCR) framework, which integrates with the primary loss (*e.g.*, cross-entropy), acting as a *semantic regularizer* to guide learning.

on graph neural networks (GNNs), our method operates locally within each batch and uses graphs constructed dynamically from features and predictions, avoiding architectural overhead.

**Contrastive and metric learning.** Contrastive learning has emerged as a powerful framework for self-supervised and supervised representation learning [110, 69, 60]. These methods typically pull together positive pairs (*e.g.*, same-class or augmented views) and push apart negatives. Extensions such as supervised contrastive learning and hard negative mining improve class discrimination by incorporating label information [66, 105]. However, these approaches often rely on explicit sampling strategies or carefully tuned augmentations, and are usually applied at a single point in the network. In contrast, our method requires no sampling, no data augmentation, and aligns structural relationships across multiple network depths, offering a form of contrast without contrastive pairs.

**Regularization and feature structure learning.** Various regularization techniques aim to improve generalization by shaping the geometry of learned feature spaces [57, 78]. For instance, center loss [86, 84] and triplet loss [24, 119, 29] enforce compactness or margin between classes. Other works apply orthogonality or decorrelation constraints on activations or weights [71, 104]. More recently, structural regularizers use pairwise distances or affinity matrices to inject relational constraints into training [73, 59]. Compared to these, our approach enforces cross-space structural alignment, between features and softmax-based predictions, while incorporating class-aware masking to focus only on semantically meaningful relationships.

**Layer-wise supervision and structural alignment.** Supervision at intermediate layers has shown promise in improving training dynamics and interpretability [52, 126, 39]. Auxiliary losses [93, 43], attention distillation [2, 50, 75], and intermediate contrastive losses [103, 15, 86] are examples where internal representations are guided explicitly. However, these methods typically supervise layers independently, and rarely use the structural information from model outputs to regularize feature learning [18, 100, 122, 129]. Our approach introduces a new form of semantic structure supervision by aligning feature similarity graphs with a masked prediction graph across multiple layers, with learnable or depth-based weighting to adaptively emphasize useful features during training.

A discussion of GCR's relationship to existing paradigms is presented in Appendices A and B. Below, we present our proposed method.

## 3 Method

We introduce the Graph Consistency Layer (GCL), a lightweight module that can be inserted at any layer or micro-architecture block (*e.g.*, a convolutional layer or an inception block, see Fig. 2). Each GCL dynamically constructs a relational graph from intermediate features and aligns it with a semantic graph derived from the model's own predictions within each training batch. This alignment is driven by Graph Consistency Regularization (GCR), a novel technique that promotes semantically coherent and geometrically structured feature learning. Our motivation is discussed in Appendix C.

## 3.1 Graph Consistency Layer

**Relational graph construction on features.** Given a batch of feature activations at layer $l$, represented by the feature matrix $\mathbf{X}^{(l)} \in \mathbb{R}^{n \times d}$ (where we vectorize or flatten high-dimensional feature maps), with $n$ being the batch size and $d$ the feature dimension, we construct a pairwise relational graph $\mathbf{F}^{(l)} \in \mathbb{R}^{n \times n}$ that encodes the relationships between features within the batch. For simplicity, we use cosine similarity to compute the relationship between the $i$-th and $j$-th samples as follows:

$$\mathbf{F}_{ij}^{(l)} = \text{ReLU}\left(\cos(\mathbf{x}_i^{(l)}, \mathbf{x}_j^{(l)})\right) \tag{1}$$

This formulation captures local geometric relationships in feature space. However, the raw feature graph $\mathbf{F}^{(l)} \in \mathbb{R}^{n \times n}$ may contain spurious correlations that misalign with true semantic class boundaries. To address this, a semantic reference graph is introduced to guide feature alignment.

**Masked relational graph on predictions.** To construct the reference graph, we use the network's prediction logits $\mathbf{Z} \in \mathbb{R}^{n \times C}$, where $C$ is the number of classes and $\mathbf{Z}$ contains the pre-softmax output scores. From these logits, we compute class probabilities via the softmax function, and then calculate the pairwise similarity between prediction vectors $\mathbf{z}_i, \mathbf{z}_j \in \mathbb{R}^C$ of the $i$-th and $j$-th samples as:

$$\boldsymbol{S}_{ij} = \text{ReLU}\left(\cos(\text{softmax}(\mathbf{z}_i), \text{softmax}(\mathbf{z}_j))\right). \tag{2}$$

The resulting matrix $\boldsymbol{S} \in \mathbb{R}^{n \times n}$ captures the pairwise similarities between the predicted class distributions of samples. However, not all prediction similarities are equally informative for guiding feature alignment, particularly (i) in the early stages of training, when predictions are often noisy and lack well-formed class semantics, and (ii) in cases where inherently ambiguous or visually similar classes introduce misleading affinities. To focus on semantically consistent pairs, we introduce a binary mask $\boldsymbol{M} \in \{0,1\}^{n \times n}$, where $\boldsymbol{M}_{ij} = 1$ if samples $i$ and $j$ share the same ground truth label, and 0 otherwise. The prediction relational graph $\mathbf{P}$ is then defined as:

$$\mathbf{P}_{ij} = \mathbf{M}_{ij} \odot \mathbf{S}_{ij}, \tag{3}$$

where $\odot$ denotes element-wise (Hadamard) multiplication. This graph retains only the prediction similarities between samples of the same class, effectively encoding intra-class semantic relationships while discarding noisy or misleading inter-class connections. Below, we outline our GCR framework.

## 3.2 Graph Consistency Regularization

**Layer-wise graph alignment.** GCR centers on aligning the feature graph $\mathbf{F}^{(l)}$ with the masked prediction graph $\mathbf{P}$. To encourage symmetric, undirected relationships and eliminate redundancy, we use only the strictly upper triangular part of the graphs, excluding the diagonal elements to remove self-connections. This design ensures that the graphs capture bidirectional affinities between samples without double-counting or emphasizing self-similarity. Such undirected structure often leads to more stable optimization and promotes balanced representation learning across the batch.

The graph alignment loss at layer $l$ is defined as the squared Frobenius norm between the upper triangular parts of the two graphs:

$$\mathcal{L}_{\text{GCR}}^{(l)} = \|\operatorname{triu}(\mathbf{F}^{(l)}) - \operatorname{triu}(\mathbf{P})\|_F^2, \tag{4}$$

where $\operatorname{triu}(\cdot)$ denotes the strictly upper triangular matrix. This loss compels the model to adjust intermediate features so their geometric structure aligns with the semantic topology encoded in the masked prediction relational graph.

**Graph consistency aggregation.** To enforce consistency across the network hierarchy, we aggregate alignment losses from a set of selected layers $\{1, \ldots, K\}$. The total GCR loss is given by:

$$\mathcal{L}_{\text{GCR}} = \sum_{l \in \{1,\ldots,K\}} w_l \cdot \|\operatorname{triu}(\mathbf{F}^{(l)}) - \operatorname{triu}(\mathbf{P})\|_F^2. \tag{5}$$

Here, $w_l$ is a weight that balances the contribution of each layer. These weights can be either: (i) fixed, using depth-based heuristics such as equal ($w_l = 1/K$), linear ($w_l = l/K$), squared ($(l/K)^2$),

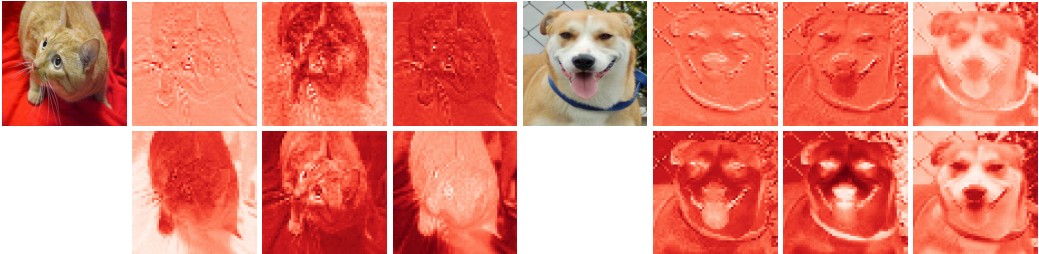

Figure 3: Feature map visualizations from models trained on *identical data batches*: (top) baseline and (bottom) our GCL-augmented model. Brighter red regions indicate stronger feature activations. Compared to the baseline, GCL-enhanced maps more clearly emphasize class-discriminative cues, *e.g.*, cat faces, ears, and eyes, and for dogs, tongues, noses, and facial contours, reflecting improved focus and interpretability. GCL also yields higher classification accuracy ($98.1\% \rightarrow 99.8\%$).

square-root ($\sqrt{l/K}$), cosine ($\frac{1+\cos(\pi\frac{l}{K})}{2}$), or arccosine ($\frac{\arccos(1-2\frac{l}{K})}{\pi}$); or (ii) adaptive, based on current alignment quality:

$$w_l = \frac{\exp\left(-\left\|\mathrm{triu}(\mathbf{F}^{(l)}) - \mathrm{triu}(\mathbf{P})\right\|_F^2\right)}{\sum_{j=1}^{K}\exp\left(-\left\|\mathrm{triu}(\mathbf{F}^{(j)}) - \mathrm{triu}(\mathbf{P})\right\|_F^2\right)}. \tag{6}$$

The adaptive weighting ensures that layers with greater misalignment are given more importance during training, allowing the model to concentrate on refining representations that are less aligned with the desired semantic structure. We evaluate different weighting schemes in our experiments.

**Training objective.** The overall training loss is composed of two components: the standard cross-entropy loss, denoted as $\mathcal{L}_{\mathrm{CE}}$, and the GCR loss, $\mathcal{L}_{\mathrm{GCR}}$. The total loss function is given by:

$$\mathcal{L}_{\mathrm{total}} = \mathcal{L}_{\mathrm{CE}} + \lambda \cdot \mathcal{L}_{\mathrm{GCR}}, \tag{7}$$

where $\lambda$ is a hyperparameter that controls the relative contribution of the GCR term. A key advantage of GCR is that it introduces no additional parameters, and its graph alignment loss relies on matrix operations well-suited to modern hardware. Next, we present the theoretical foundations of GCR.

### 3.3 Theoretical Analysis of GCR

We present a theoretical analysis of GCR, linking its empirical design to core principles in statistical learning and spectral graph theory. Specifically, we show that minimizing the GCR loss: (i) reduces the effective hypothesis class capacity via covering number bounds[55, 123] and Dudley's entropy integral[32]; (ii) promotes spectral alignment between learned features and semantic prediction graphs through normalized Laplacians[5]; and (iii) acts as a PAC-Bayesian regularizer[35], imposing a structural prior over the function space. Additional insights are provided in Appendix D.

**Generalization via covering numbers.** Let $\mathcal{F}_L$ be the class of functions $f^{(l)} : \mathcal{X} \to \mathbb{R}^d$ representing layer-$l$ embeddings. Assume the feature representations are uniformly bounded by a constant $B$ in $\ell_2$ norm: $\|f^{(l)}(x)\|_2 = \sqrt{\sum_{i=1}^{d}\left(f_i^{(l)}(x)\right)^2} \le B$, for all $x \in \mathcal{X}$. This is a standard constraint in learning theory to control hypothesis space capacity. In practice, with $\ell_2$ normalization as used here, $B = 1$. Next, we define a structurally-constrained hypothesis class:

$$\mathcal{F}_\epsilon := \left\{f \in \mathcal{F}_L : \mathcal{L}_{\mathrm{GCR}}^{(l)} := \|\mathrm{triu}(\mathbf{F}^{(l)}) - \mathrm{triu}(\mathbf{P})\|_F^2 \le \epsilon\right\}, \tag{8}$$

which enforces alignment between feature and prediction graphs reflecting intra-class similarity.

**Theorem 1** (Generalization via Dudley's entropy integral). *Let $\ell(f(x), y)$ be a $\gamma$-Lipschitz loss function (e.g., cross-entropy), and let $\mathcal{F}_L$ be the class of functions at layer $l$ such that each function $f^{(l)}$ satisfies the $\ell_2$-bounded constraint $\|f^{(l)}(x)\|_2 \le B$. Suppose $\mathcal{F}_\epsilon \subseteq \mathcal{F}_L$ is the subset of functions that are additionally constrained by the GCR alignment loss:*

$$\mathcal{L}_{GCR}^{(l)} = \frac{1}{n^2}\sum_{i,j=1}^{n}\left(ReLU(\langle\mathbf{x}_i^{(l)}, \mathbf{x}_j^{(l)}\rangle) - \mathbf{P}_{ij}\right)^2 \le \epsilon, \tag{9}$$

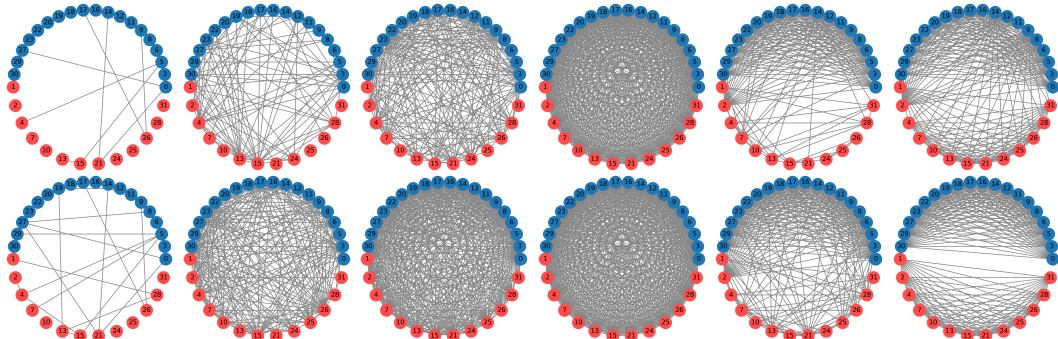

Figure 4: Relational graph visualization on Kaggle cats *vs*. dogs. We compare the best baseline model and our GCL-augmented model using the same batch of 32 samples (red = cat, blue = dog). The baseline consists of four convolutional blocks and two fully connected layers; our method inserts a Graph Consistency Layer (GCL) after each, totaling six GCLs. The top row shows the baseline (without GCLs); the bottom row shows our GCL-enhanced model. Each column visualizes the relational graph at a specific layer, from early features (left) to final predictions (right). Early layers exhibit weak connectivity, as low-level features poorly capture class semantics. As depth increases, both models shift toward more structured, class-separable relationships. GCLs amplify this effect by attenuating low-similarity inter-class edges and reinforcing intra-class coherence, leading to improved accuracy (98.1% *vs*. 99.8%). For clarity, edges with similarity $< 0.4$ are omitted.

*where $\mathbf{P}_{ij}$ is the target alignment between the normalized feature vectors $\mathbf{x}_i^{(l)}$ and $\mathbf{x}_j^{(l)}$. The GCR loss enforces that the angular distances between the feature vectors are small, meaning that the vectors are close to each other in the Euclidean space. If $\mathcal{F}_\epsilon$ admits a covering number bound:*

$$\mathcal{N}(\mathcal{F}_\epsilon, \|\cdot\|_2, \alpha) \leq \left(\frac{C}{\alpha}\right)^d, \tag{10}$$

*where $\mathcal{N}(\mathcal{F}_\epsilon, \|\cdot\|_2, \alpha)$ is the covering number of $\mathcal{F}_\epsilon$ with respect to the $\ell_2$ norm, then the expected loss of a function $f \in \mathcal{F}_\epsilon$ is bounded with high probability by:*

$$\mathbb{E}_{(x,y)\sim\mathcal{D}}[\ell(f(x),y)] \leq \frac{1}{n}\sum_{i=1}^n \ell(f(x_i),y_i) + \frac{12\gamma}{\sqrt{n}}\int_0^B \sqrt{d\log\left(\frac{C}{\alpha}\right)}\, d\alpha, \tag{11}$$

*where $B = O(\sqrt{\epsilon})$ is the effective radius of function class $\mathcal{F}_\epsilon$ under the GCR constraint, and the second term represents the generalization error, which is controlled by the function class complexity.*

*Proof.* The proof can be found in Appendix E. $\qquad\square$

**Remark 1.** *This result shows that GCR reduces generalization error by shrinking the effective complexity of the function class. By aligning relational structure, GCR implicitly contracts the hypothesis space, leading to improved generalization.*

**Spectral alignment via normalized Laplacians.** Let $\mathbf{F}$ and $\mathbf{P}$ be symmetric affinity matrices derived from feature embeddings and masked predictions, respectively. Their associated normalized graph Laplacians are defined as $\mathcal{L}_\mathbf{F} := \mathbf{I} - \mathbf{D}_F^{-1/2}\mathbf{F}\mathbf{D}_F^{-1/2}$ and $\mathcal{L}_\mathbf{P} := \mathbf{I} - \mathbf{D}_P^{-1/2}\mathbf{P}\mathbf{D}_P^{-1/2}$, where $\mathbf{D}_F$ and $\mathbf{D}_P$ are the degree matrices corresponding to $\mathbf{F}$ and $\mathbf{P}$, *i.e.*, $(\mathbf{D}_F)_{ii} = \sum_j \mathbf{F}_{ij}$ and similarly for $\mathbf{D}_P$.

**Proposition 1** (Spectral alignment). *Let $\mathbf{F}$ and $\mathbf{P}$ be symmetric matrices such that*

$$\|\mathbf{F} - \mathbf{P}\|_F \leq \epsilon, \quad \|\mathbf{D}_F - \mathbf{D}_P\| \leq \delta. \tag{12}$$

*Then, there exists a constant $C > 0$ depending on spectral properties of the graphs (*e.g., *sparsity, minimum degree), such that*

$$\|\mathcal{L}_\mathbf{F} - \mathcal{L}_\mathbf{P}\|_F \leq C(\epsilon + \delta). \tag{13}$$

*Proof.* The proof can be found in Appendix E. $\qquad\square$

Table 1: Accuracy (%) on CIFAR-10 across models. Results are shown for MobileNet (MNet), ShuffleNet (SN), SqueezeNet (SQNet), GoogLeNet (GLNet), ResNeXt-50/101 (Rx-50/101), ResNet-34/50/101 (R34/R50/R101), DenseNet-121 (D121), and MAE under various GCL configurations (Early, Mid, Late, combinations, Full). Bold indicates the best improvements over baselines; underlines mark the second-best. The final column shows the average accuracy for each configuration.

| | MAE | MNet | SN | SQNet | GLNet | Rx-50 | Rx-101 | R34 | R50 | R101 | D121 | Mean |
|---|---|---|---|---|---|---|---|---|---|---|---|---|
| Baseline | $88.95_{\pm0.33}$ | $90.23_{\pm0.25}$ | $91.21_{\pm0.28}$ | $92.30_{\pm0.25}$ | $94.10_{\pm0.26}$ | $94.57_{\pm0.29}$ | $95.12_{\pm0.30}$ | $94.83_{\pm0.25}$ | $95.03_{\pm0.28}$ | $95.22_{\pm0.31}$ | $95.01_{\pm0.27}$ | $93.32_{\pm2.26}$ |
| Early GCL | $89.42_{\pm0.25}$ | $91.17_{\pm0.22}$ | $92.33_{\pm0.33}$ | $92.59_{\pm0.21}$ | $\mathbf{94.89}_{\pm0.23}$ | $95.48_{\pm0.22}$ | $\underline{95.63}_{\pm0.25}$ | $95.55_{\pm0.18}$ | $95.57_{\pm0.23}$ | $95.39_{\pm0.26}$ | $\mathbf{95.81}_{\pm0.17}$ | $93.98_{\pm2.22}$ |
| Mid GCL | $\mathbf{89.77}_{\pm0.22}$ | $91.15_{\pm0.18}$ | $92.58_{\pm0.19}$ | $92.40_{\pm0.20}$ | $94.82_{\pm0.21}$ | $95.47_{\pm0.19}$ | $95.39_{\pm0.24}$ | $\mathbf{95.69}_{\pm0.23}$ | $95.61_{\pm0.20}$ | $\mathbf{95.75}_{\pm0.17}$ | $95.51_{\pm0.22}$ | $94.01_{\pm2.15}$ |
| Late GCL | $\underline{89.70}_{\pm0.29}$ | $\mathbf{91.40}_{\pm0.19}$ | $92.36_{\pm0.21}$ | $\mathbf{92.80}_{\pm0.19}$ | $94.88_{\pm0.19}$ | $95.35_{\pm0.28}$ | $\mathbf{95.71}_{\pm0.19}$ | $\mathbf{95.69}_{\pm0.19}$ | $\mathbf{95.66}_{\pm0.17}$ | $95.51_{\pm0.24}$ | $\underline{95.72}_{\pm0.22}$ | $\mathbf{94.07}_{\pm2.14}$ |
| Early+Mid | $89.52_{\pm0.19}$ | $90.77_{\pm0.26}$ | $92.56_{\pm0.21}$ | $92.27_{\pm0.25}$ | $94.79_{\pm0.18}$ | $95.33_{\pm0.27}$ | $95.55_{\pm0.23}$ | $95.46_{\pm0.20}$ | $95.51_{\pm0.21}$ | $95.37_{\pm0.19}$ | $95.64_{\pm0.20}$ | $93.89_{\pm2.22}$ |
| Mid+Late | $89.59_{\pm0.28}$ | $\underline{91.23}_{\pm0.20}$ | $\mathbf{92.79}_{\pm0.20}$ | $\mathbf{92.86}_{\pm0.23}$ | $94.61_{\pm0.22}$ | $\mathbf{95.51}_{\pm0.19}$ | $95.38_{\pm0.27}$ | $95.45_{\pm0.18}$ | $95.33_{\pm0.26}$ | $95.52_{\pm0.14}$ | $95.70_{\pm0.19}$ | $94.00_{\pm2.09}$ |
| Early+Late | $89.64_{\pm0.25}$ | $91.03_{\pm0.24}$ | $92.30_{\pm0.28}$ | $92.70_{\pm0.23}$ | $94.69_{\pm0.20}$ | $95.40_{\pm0.20}$ | $95.35_{\pm0.23}$ | $95.66_{\pm0.21}$ | $95.31_{\pm0.25}$ | $95.49_{\pm0.16}$ | $95.53_{\pm0.22}$ | $93.92_{\pm2.14}$ |
| Full GCL | $89.55_{\pm0.23}$ | $90.99_{\pm0.18}$ | $92.48_{\pm0.19}$ | $92.65_{\pm0.20}$ | $94.57_{\pm0.21}$ | $\underline{95.50}_{\pm0.19}$ | $95.34_{\pm0.20}$ | $95.48_{\pm0.17}$ | $\underline{95.62}_{\pm0.18}$ | $95.38_{\pm0.21}$ | $95.51_{\pm0.20}$ | $93.92_{\pm2.15}$ |

**Corollary 1.** *The GCR alignment loss, which encourages $\|\mathbf{F} - \mathbf{P}\|_F \leq \epsilon$, indirectly enforces spectral similarity of the normalized Laplacians. This promotes agreement between the clustering structure and diffusion properties of the learned features and masked predictions.*

**PAC-Bayesian view of structural regularization.** We now present a PAC-Bayesian view of GCR, which bounds generalization by linking expected loss to empirical loss and the divergence between posterior and prior over hypotheses.

Let $\mathcal{P}$ denote a prior distribution over model functions $f$, representing a structure-agnostic belief (*e.g.*, uniform or isotropic Gaussian over parameters). Let $\mathcal{Q}$ be a posterior distribution supported on models that minimize training loss while also conforming to a structural constraint induced by GCR, *i.e.*, $\mathcal{Q}$ is restricted to functions $f$ such that $\mathcal{L}_{\text{GCR}}^{(l)} \leq \epsilon$ for each relevant layer $l$.

**Theorem 2** (PAC-Bayes generalization bound with GCR). *Let $\mathcal{L}(f) = \mathbb{E}_{(x,y)\sim\mathcal{D}}[\ell(f(x), y)]$ be the expected population loss of model $f$ and let $\hat{\mathcal{L}}(f) = \frac{1}{n}\sum_{i=1}^{n} \ell(f(x_i), y_i)$ be the empirical loss on $n$ training examples. Then, for any posterior distribution $\mathcal{Q}$ over functions and any prior distribution $\mathcal{P}$, with probability at least $1 - \delta$ over the training data, the following bound holds:*

$$\mathbb{E}_{f\sim\mathcal{Q}}[\mathcal{L}(f)] \leq \mathbb{E}_{f\sim\mathcal{Q}}[\hat{\mathcal{L}}(f)] + \sqrt{\frac{\text{KL}(\mathcal{Q}\|\mathcal{P}) + \log(1/\delta)}{2n}}. \tag{14}$$

This classical PAC-Bayesian bound quantifies generalization through two factors: (i) the empirical performance of models sampled from the posterior $\mathcal{Q}$, and (ii) the Kullback-Leibler (KL) divergence[44] $\text{KL}(\mathcal{Q}|\mathcal{P})$, measuring how much $\mathcal{Q}$ deviates from the prior. In GCR, the constraint $\mathcal{L}_{\text{GCR}}^{(l)} \leq \epsilon$ imposes structured alignment between feature similarity $\mathbf{F}^{(l)}$ and semantic prediction structure $\mathbf{P}$, serving as an inductive bias. Thus, when $\mathcal{Q}$ is supported on models satisfying this constraint, the KL complexity term reflects the strength of this alignment.

**Proposition 2** (Structure-induced KL complexity). *If the posterior $\mathcal{Q}$ is concentrated on models with small GCR loss at layer $l$, then the KL divergence to an isotropic prior $\mathcal{P}$ satisfies:*

$$\text{KL}(\mathcal{Q}\|\mathcal{P}) \leq C \sum_{l} \|\mathbf{F}^{(l)} - \mathbf{P}\|_F^2, \tag{15}$$

*for some constant $C$ depending on the form of $\mathcal{P}$.*

*Proof.* The proof can be found in Appendix E. □

**Remark 2.** *This perspective shows that GCR does more than minimize training loss, it also implicitly regularizes the hypothesis space by favoring models whose internal representations reflect known semantic structure. This improves generalization by reducing the effective size of the model class, as made explicit through the PAC-Bayesian framework.*

# 4 Experiment

## 4.1 Datasets, Models, and Experimental Setup

We evaluate GCR on several benchmark datasets, including Kaggle cats *vs.* dogs[22], CIFAR-10[62], CIFAR-100[62], Tiny ImageNet[63], and ImageNet-1K [23]. Our evaluation spans a di-

Table 2: Accuracy (%) on CIFAR-100 across models.

| | MAE | MNet | SN | SQNet | Rx-50 | Rx-101 | R34 | R50 | D121 | Mean |
|---|---|---|---|---|---|---|---|---|---|---|
| Baseline | $64.29_{\pm0.34}$ | $65.95_{\pm0.25}$ | $70.11_{\pm0.30}$ | $69.43_{\pm0.27}$ | $77.75_{\pm0.29}$ | $77.83_{\pm0.30}$ | $76.82_{\pm0.28}$ | $77.31_{\pm0.29}$ | $77.09_{\pm0.27}$ | $72.95_{\pm5.50}$ |
| Early GCL | $65.05_{\pm0.29}$ | $67.45_{\pm0.21}$ | $\mathbf{71.96}_{\pm0.27}$ | $70.90_{\pm0.20}$ | $79.18_{\pm0.22}$ | $79.69_{\pm0.27}$ | $77.90_{\pm0.22}$ | $\underline{79.37}_{\pm0.25}$ | $79.41_{\pm0.22}$ | $\underline{74.55}_{\pm5.78}$ |
| Mid GCL | $64.99_{\pm0.30}$ | $67.88_{\pm0.21}$ | $71.89_{\pm0.24}$ | $70.21_{\pm0.25}$ | $79.07_{\pm0.19}$ | $79.28_{\pm0.20}$ | $77.83_{\pm0.20}$ | $78.90_{\pm0.24}$ | $79.26_{\pm0.21}$ | $74.37_{\pm5.66}$ |
| Late GCL | $\mathbf{65.54}_{\pm0.27}$ | $\underline{68.32}_{\pm0.20}$ | $71.42_{\pm0.24}$ | $70.55_{\pm0.22}$ | $\mathbf{79.54}_{\pm0.20}$ | $\mathbf{79.83}_{\pm0.21}$ | $\mathbf{78.31}_{\pm0.20}$ | $\mathbf{79.42}_{\pm0.21}$ | $\mathbf{79.69}_{\pm0.23}$ | $\mathbf{74.74}_{\pm5.73}$ |
| Early+Mid | $65.23_{\pm0.31}$ | $67.62_{\pm0.24}$ | $71.50_{\pm0.28}$ | $70.47_{\pm0.19}$ | $78.90_{\pm0.18}$ | $79.25_{\pm0.20}$ | $77.41_{\pm0.19}$ | $78.58_{\pm0.24}$ | $79.22_{\pm0.20}$ | $74.28_{\pm5.56}$ |
| Mid+Late | $65.27_{\pm0.28}$ | $\mathbf{68.33}_{\pm0.19}$ | $71.63_{\pm0.28}$ | $70.30_{\pm0.22}$ | $78.91_{\pm0.17}$ | $79.57_{\pm0.21}$ | $77.30_{\pm0.20}$ | $78.85_{\pm0.22}$ | $79.54_{\pm0.24}$ | $74.41_{\pm5.55}$ |
| Early+Late | $65.22_{\pm0.21}$ | $67.25_{\pm0.21}$ | $71.55_{\pm0.27}$ | $\mathbf{71.03}_{\pm0.24}$ | $79.03_{\pm0.20}$ | $79.41_{\pm0.22}$ | $\underline{78.19}_{\pm0.23}$ | $78.70_{\pm0.23}$ | $79.45_{\pm0.22}$ | $74.43_{\pm5.69}$ |
| Full GCL | $\underline{65.38}_{\pm0.22}$ | $68.22_{\pm0.19}$ | $71.30_{\pm0.24}$ | $70.77_{\pm0.20}$ | $79.01_{\pm0.19}$ | $79.29_{\pm0.21}$ | $77.79_{\pm0.20}$ | $78.71_{\pm0.22}$ | $79.27_{\pm0.19}$ | $74.42_{\pm5.49}$ |

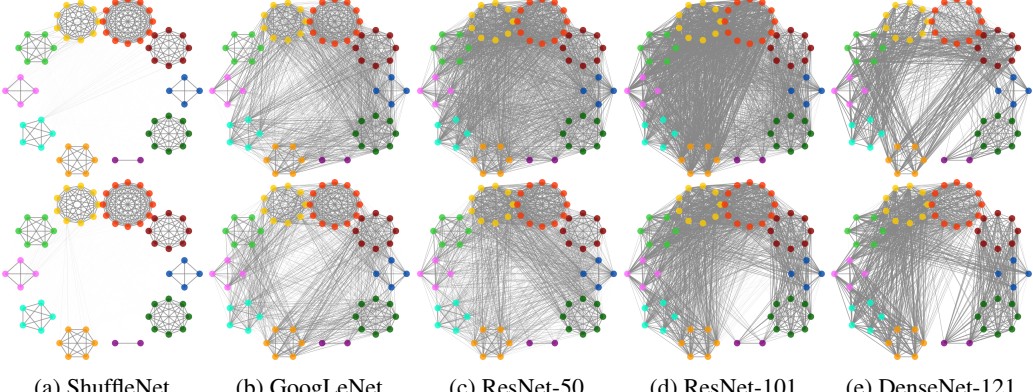

| (a) ShuffleNet | (b) GoogLeNet | (c) ResNet-50 | (d) ResNet-101 | (e) DenseNet-121 |
|---|---|---|---|---|

Figure 5: Relational graph comparison across five models on the same batch. Top row: baselines; bottom row: GCL-augmented versions, showing *sparser inter-class connections* and *stronger class-aware structure*, highlighting GCL's effectiveness in enhancing relational representations.

verse range of architectures, from lightweight models (*e.g.*, MobileNet[46], ShuffleNet[124], and SqueezeNet[51]), to deeper CNNs (*e.g.*, GoogLeNet[97], ResNet[42], DenseNet[48], ResNeXt[112], Stochastic ResNet[49], and SE-ResNet[47]). We also include transformer-based architectures such as ViT[31], Swin[70], MobileViT[77], CEiT[120]), iFormer [95], ViG [37], as well as Masked Autoencoders (MAE) [41].

Experiments run on NVIDIA V100 GPUs with 12 CPUs and 48 GB RAM. For CNNs, we follow [25]: 200 epochs, initial learning rate 0.1 (decayed at epochs 60/120/160), batch size 128, weight decay $5\times10^{-4}$, and momentum 0.9. For transformers, we use AdamW with a learning rate of $1\times10^{-4}$, cosine annealing, weight decay $5\times10^{-2}$, batch size 256, AMP[30], 10-epoch warm-up, and gradient clipping (norm 1.0). We divide each model into three stages: early (E), middle (M), and late (L), and insert GCLs to create seven configurations: individual stages (E, M or L), pairs (E+M, M+L, E+L), and full GCL. We evaluate weighting schemes including equal, linear, square root, squared, cosine, arccosine, and adaptive. All experiments use $\lambda=1$. We report the mean and standard deviation over 10 runs for CIFAR-10, CIFAR-100, and Tiny ImageNet, and over 3 runs for ImageNet-1K.

Appendix F presents a detailed analysis of GCR's time complexity, while Appendix G provides complete details of the experimental setup.

## 4.2 Quantitative and Qualitative Evaluation

**GCL can serve as an attention mechanism.** Fig. 3 offers a visual comparison between feature activations from a baseline CNN model and those from a GCL-augmented model, both trained on identical data batches. The key distinction lies in the sharper, more localized activations produced by the GCL-enhanced model. Notably, the parameter-free GCL mechanism yields feature maps that are more aligned with semantically relevant regions, such as the eyes, ears, and facial outlines of the cat, and the tongue, snout, and eyes of the dog. This suggests that GCL encourages the network to focus on class-discriminative features, thereby reducing attention to background or irrelevant image regions. In contrast, the baseline model's activations appear diffuse and less structured, indicating weaker spatial selectivity. The visual improvements align well with the reported jump in classification accuracy from 98.1% to 99.8%, highlighting that improved interpretability does not come at the cost of performance, rather, it appears to enhance it. Fig. 4 compares how GCLs guide model learning.

Table 3: Accuracy (%) on Tiny ImageNet across models. All results are obtained by training models from scratch. We also evaluate Stochastic ResNet-18 (R18SD) and SE-ResNet-18 (SER18).

| | ViT$_{/32}$ | ViT$_{/16}$ | CeiT | MViT$_{XXS}$ | MViT$_{XS}$ | MViT | Swin | MNet | R18SD | SER18 | R34 | Mean |
|---|---|---|---|---|---|---|---|---|---|---|---|---|
| Baseline | 37.79$_{\pm0.35}$ | 40.05$_{\pm0.33}$ | 49.95$_{\pm0.29}$ | 49.28$_{\pm0.29}$ | 51.58$_{\pm0.27}$ | 52.68$_{\pm0.27}$ | 54.27$_{\pm0.25}$ | 57.81$_{\pm0.25}$ | 63.49$_{\pm0.26}$ | 65.65$_{\pm0.24}$ | 67.51$_{\pm0.25}$ | 53.64$_{\pm9.62}$ |
| Early GCL | 39.02$_{\pm0.29}$ | 40.98$_{\pm0.19}$ | **51.22**$_{\pm0.20}$ | 50.11$_{\pm0.28}$ | 51.33$_{\pm0.26}$ | 53.91$_{\pm0.22}$ | 54.88$_{\pm0.25}$ | **57.93**$_{\pm0.21}$ | 63.81$_{\pm0.19}$ | **66.52**$_{\pm0.22}$ | **67.79**$_{\pm0.19}$ | **54.32**$_{\pm9.39}$ |
| Mid GCL | 38.61$_{\pm0.23}$ | 40.95$_{\pm0.19}$ | 50.30$_{\pm0.19}$ | 49.92$_{\pm0.26}$ | 51.43$_{\pm0.22}$ | 53.88$_{\pm0.20}$ | 55.23$_{\pm0.24}$ | 57.63$_{\pm0.20}$ | 64.03$_{\pm0.22}$ | 65.66$_{\pm0.23}$ | 67.62$_{\pm0.20}$ | 54.11$_{\pm9.38}$ |
| Late GCL | 37.98$_{\pm0.28}$ | 40.35$_{\pm0.25}$ | 50.82$_{\pm0.20}$ | 49.77$_{\pm0.21}$ | **51.99**$_{\pm0.23}$ | **54.10**$_{\pm0.19}$ | 55.47$_{\pm0.21}$ | 57.87$_{\pm0.23}$ | 63.79$_{\pm0.19}$ | 65.85$_{\pm0.25}$ | 67.61$_{\pm0.19}$ | 54.18$_{\pm9.56}$ |
| Early+Mid | **39.08**$_{\pm0.25}$ | **41.26**$_{\pm0.18}$ | 50.25$_{\pm0.25}$ | 49.73$_{\pm0.22}$ | 51.57$_{\pm0.19}$ | 53.91$_{\pm0.23}$ | 54.95$_{\pm0.19}$ | 57.49$_{\pm0.19}$ | 64.18$_{\pm0.20}$ | 65.86$_{\pm0.24}$ | 67.74$_{\pm0.23}$ | 54.18$_{\pm9.32}$ |
| Mid+Late | 38.44$_{\pm0.18}$ | 40.52$_{\pm0.28}$ | 50.09$_{\pm0.25}$ | **50.55**$_{\pm0.18}$ | 51.48$_{\pm0.21}$ | 53.90$_{\pm0.20}$ | **55.62**$_{\pm0.23}$ | 57.65$_{\pm0.21}$ | **64.29**$_{\pm0.17}$ | 65.95$_{\pm0.19}$ | 67.58$_{\pm0.21}$ | 54.19$_{\pm9.52}$ |
| Early+Late | 38.34$_{\pm0.23}$ | 40.71$_{\pm0.21}$ | 50.70$_{\pm0.25}$ | 50.23$_{\pm0.20}$ | 51.36$_{\pm0.18}$ | 53.57$_{\pm0.21}$ | 54.89$_{\pm0.21}$ | 57.93$_{\pm0.19}$ | 63.88$_{\pm0.19}$ | 65.83$_{\pm0.17}$ | 67.75$_{\pm0.25}$ | 54.11$_{\pm9.47}$ |
| Full GCL | 38.38$_{\pm0.22}$ | 40.80$_{\pm0.18}$ | 49.92$_{\pm0.20}$ | 50.16$_{\pm0.17}$ | 51.87$_{\pm0.19}$ | 54.01$_{\pm0.19}$ | 54.87$_{\pm0.19}$ | 57.64$_{\pm0.20}$ | 64.10$_{\pm0.19}$ | 66.01$_{\pm0.15}$ | 67.66$_{\pm0.18}$ | 54.13$_{\pm9.49}$ |

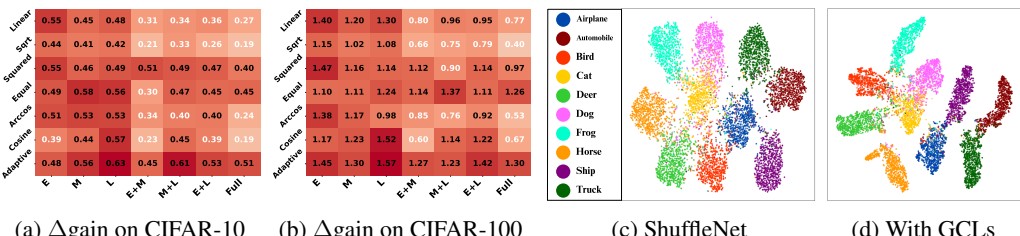

| (a) Δgain on CIFAR-10 | (b) Δgain on CIFAR-100 | (c) ShuffleNet | (d) With GCLs |
|---|---|---|---|

Figure 6: Performance gains (Δ,%) from adding GCLs with different configurations across various weighting schemes. Darker red indicates larger gains. Adaptive weighting achieves the highest improvement, showing the value of graph misalignment as a guidance signal. (c) and (d) show that GCL-augmented ShuffleNet yields more compact intra-class clusters and better inter-class separation.

**GCLs lead to more interpretable and discriminative feature spaces.** In GCL-augmented graphs (Figs. 1 and 5), semantically similar classes such as Airplane, Ship, and Truck (all vehicles) form closer associations, suggesting that GCL helps the model better understand high-level concepts. A similar trend appears among animal-related classes (*e.g.*, Dog, Cat, Horse, and Deer). As model complexity increases (from ShuffleNet to DenseNet-121), the relational graphs become denser. However, GCL still consistently improves clarity. Even in large-capacity models like ResNet-101 and DenseNet-121, GCL enhances structure by reducing cross-class noise and reinforcing class-wise coherence. The consistent improvements across very different models, from lightweight ShuffleNet to deep DenseNet, demonstrate the general applicability of GCL. It does not merely overfit to a specific architecture but contributes to relational learning in a model-agnostic, parameter-free way.

**Adaptive weighting and selective layers boost GCR effectiveness.** The adaptive scheme consistently yields the highest or near-highest gains (Figs. 6a and 6b), especially on CIFAR-100, where deeper features benefit more from dynamic supervision. Placing GCLs at later layers (L) or in combinations involving deeper blocks (*e.g.*, E+L, Full) yields the greatest improvements, highlighting the semantic richness of deeper layers. Among fixed schemes, squared and equal weighting outperform linear. Notably, applying GCLs to all layers (Full) does not always yield the best gains, indicating that naïve aggregation can weaken the regularization effect. These findings emphasize the value of selective layer placement and adaptive weighting to effectively guide feature alignment.

Tables 1, 2, and 3 summarize the accuracy gains achieved by GCR across three benchmarks.

**GCLs are effective across various datasets and model types.** Across CIFAR-10, CIFAR-100, and Tiny ImageNet, GCR consistently improves accuracy relative to the baseline in nearly all configurations. On CIFAR-10, which is comparatively simpler, GCR delivers a peak mean accuracy of 94.07% (Late GCL), improving over the 93.32% baseline. CIFAR-100, with greater semantic granularity and inter-class overlap, benefits even more: Late GCL increases the mean accuracy from 72.95% to 74.74%, a +1.79% absolute gain. On Tiny ImageNet, where models are trained from scratch under more challenging conditions, GCR again proves effective. The best performance is achieved with Early GCL (54.32%) and Late GCL (54.18%), improving over the 53.64% baseline and affirming GCR's applicability to both convolutional and transformer-based vision models.

**GCLs improve transformer architectures on ImageNet-1K.** We conducted experiments on ImageNet-1K using transformer-based architectures (*e.g.*, iFormer, ViT, and ViG). Results are summarized in Table 4. Key results demonstrate the effectiveness of our approach. For iFormer-S, GCR boosts performance from 83.4% to 84.8%, yielding a +1.4% gain. For iFormer-B, accuracy

Table 4: Comparison of iFormer, ViT, and ViG with different GCL integration strategies on ImageNet-1K. Results are averaged over three runs.

| Method | iFormer-S | iFormer-B | ViT-B/16 | ViG-B |
|---|---|---|---|---|
| Baseline | $83.4_{\pm 0.40}$ | $84.6_{\pm 0.45}$ | $74.3_{\pm 0.51}$ | $82.3_{\pm 0.42}$ |
| Early GCL | $83.8_{\pm 0.31}$ | $85.0_{\pm 0.40}$ | $74.7_{\pm 0.44}$ | $82.8_{\pm 0.35}$ |
| Mid GCL | $83.8_{\pm 0.39}$ | $85.5_{\pm 0.33}$ | $75.2_{\pm 0.36}$ | $83.0_{\pm 0.34}$ |
| Late GCL | $\underline{84.5}_{\pm 0.29}$ | $\mathbf{86.1}_{\pm 0.30}$ | $\mathbf{75.8}_{\pm 0.33}$ | $\mathbf{84.0}_{\pm 0.30}$ |
| Early + Mid | $84.3_{\pm 0.33}$ | $\underline{85.9}_{\pm 0.38}$ | $\underline{75.6}_{\pm 0.41}$ | $83.7_{\pm 0.33}$ |
| Mid + Late | $\mathbf{84.8}_{\pm 0.28}$ | $\underline{85.9}_{\pm 0.37}$ | $\underline{75.6}_{\pm 0.34}$ | $\underline{83.9}_{\pm 0.30}$ |
| Early + Late | $\underline{84.5}_{\pm 0.30}$ | $85.2_{\pm 0.28}$ | $74.9_{\pm 0.33}$ | $83.5_{\pm 0.29}$ |
| Full GCL | $84.3_{\pm 0.29}$ | $85.8_{\pm 0.26}$ | $75.5_{\pm 0.30}$ | $83.6_{\pm 0.27}$ |

improves from 84.6% to 86.1%, corresponding to a +1.5% gain. Similar improvements are also observed with ViT-B/16 and ViG-B, confirming the generality of our method. These improvements validate that GCR scales effectively to large, complex datasets and modern architectures. They support our core hypothesis: aligning feature geometry with prediction semantics enhances generalization. Moreover, GCR achieves this without architectural changes or added parameters, reinforcing its value as a model-agnostic and lightweight regularizer.

**GCLs are most effective in later layers, aligning with decision structures.** Among GCL configurations, those applied at later stages generally yield the most significant improvements. Late GCL consistently achieves the highest or near-highest performance across all three datasets. This is likely because later layers encode higher-level semantic representations that align more closely with class-level decision boundaries. Aligning these representations with the prediction graph enhances inter-class separability while reducing intra-class dispersion, especially on fine-grained tasks like CIFAR-100 and Tiny ImageNet. While Early and Mid GCLs also contribute, their benefits are comparatively moderate, indicating that feature regularization at earlier stages may be less semantically meaningful or more susceptible to noise. Figs. 6a and 6b show clear performance gains.

**GCLs enhance geometry and generalization.** Finally, the results substantiate GCR's central hypothesis: aligning feature space geometry with the model's own prediction space fosters more discriminative and generalizable representations. Across datasets, GCR reduces classification ambiguity, tightens intra-class cohesion, and sharpens class boundaries, particularly important in datasets with high inter-class similarity. Figs.6c and 6d compare results on ShuffleNet, showing that the GCL-augmented model produces more compact intra-class clusters and better inter-class separability. Moreover, GCR achieves this with minimal computational overhead and zero parameter increase, making it highly practical for real-world deployment where architectural changes or expensive training adjustments are undesirable.

Additional results, visualizations, and discussions are provided in Appendix H, while the work's limitations and future research directions are covered in Appendix I.

# 5   Conclusion

We introduced GCR, a novel approach for enhancing classification by aligning the structural relationships between feature representations and model predictions through parameter-free, modular GCLs. By constructing feature relational graphs at multiple points in the network and aligning them with a global masked prediction graph, GCR encourages the network to learn semantically meaningful and class-consistent representations. This graph-based alignment acts as a form of neural network prompting, guiding learning without modifying the architecture or training procedure. Our experiments across diverse architectures demonstrate that GCR improves intra-class cohesion, reduces noisy inter-class similarity, and leads to better generalization. We believe this work offers a new perspective on structured representation learning by using prediction structure as a regularizing signal. Future work may explore extending this idea to tasks beyond classification, such as segmentation or retrieval, and integrating it with self-supervised learning frameworks.

## Acknowledgments and Disclosure of Funding

Xi Ding, a visiting scholar at the ARC Research Hub for Driving Farming Productivity and Disease Prevention, Griffith University, conducted this work under the supervision of Lei Wang. Lei Wang contributed to the mathematical analysis, modeling, and theoretical development, while Xi Ding implemented the code and performed the experiments.

We sincerely thank the anonymous reviewers for their invaluable insights and constructive feedback, which have greatly contributed to improving our work.

This work was supported by the Australian Research Council (ARC) under Industrial Transformation Research Hub Grant IH180100002.

This work was also supported by computational resources provided by the Australian Government through the National Computational Infrastructure (NCI) under both the ANU Merit Allocation Scheme (Lead CI: Lei Wang) and the CSIRO Allocation Scheme (Lead CI: Piotr Koniusz).

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

# A  Relation to Existing Paradigms

Below, we situate GCR within these paradigms, highlighting its unique contributions and distinctions.

**Graph-based learning and relational alignment.** Graph-based methods are widely used to model relational structures in classification [56, 92, 127] and semi-supervised learning [96, 11, 82], often through pre-constructed graphs [109, 125] or memory banks [68]. Traditional approaches use graph Laplacian regularization [80, 4] or GNN-based message passing [13, 40] to encourage feature alignment. While powerful, these methods require explicit graph definitions or heavy architectural components like GNNs and GATs [61, 102].

> GCR diverges fundamentally from these strategies:
>
>   i. **On-the-fly graph construction.** Instead of relying on static or memory-driven graphs [36, 56, 76], GCR builds internal graphs dynamically during training, using model-generated predictions and feature embeddings.
>
>  ii. **Regularization, not transformation.** Unlike GNNs that propagate features [89, 10, 85], GCR purely regularizes the relational structure of features to enforce alignment with class-aware prediction similarity.
>
> iii. **Parameter-free alignment.** GCR operates without additional parameters for graph construction, using only the inherent feature and prediction relationships for alignment.
>
> This design enables GCR to maintain the benefits of relational modeling while being model-agnostic and computationally efficient, extending graph-based learning into deeper, more flexible architectures. Furthermore, GCR enhances generalization by implicitly smoothing decision boundaries through its consistency-driven alignment, even in high-dimensional feature spaces.

**Contrastive learning and semantic consistency.** Contrastive methods [60, 118, 45] pull together positive pairs and push apart negatives, often using data augmentation [99, 107] and hard negative mining for improved class separation [14, 60]. However, these methods are sensitive to sampling strategies [116, 19] and require margin tuning for stability [34].

> GCR introduces a contrastive mechanism that is *global and implicit*:
>
>   i. **Global relational alignment.** Rather than pairwise contrasts [17], GCR aligns the entire feature graph and prediction graph, capturing holistic relationships across the batch.
>
>  ii. **Self-supervised by prediction.** The semantic graph is constructed directly from model predictions, obviating the need for manual sampling or augmentation strategies [72, 19].
>
> iii. **Semantic geometric regularization.** GCR enforces global coherence between semantic and geometric structures, extending beyond local distances to full-batch relational consistency.
>
> This implicit contrastive mechanism ensures stable alignment of semantic structures without the pitfalls of traditional contrastive learning, contributing to stronger robustness against noisy samples and domain shifts.

**Structural regularization and manifold learning.** Traditional regularizers like center loss [86, 84] and triplet loss [24, 119, 29] enforce feature compactness or inter-class margins, while graph Laplacian regularization [80, 4] smooths label propagation over fixed data graphs [7, 130]. However, these approaches [78, 98, 94] are limited by predefined graphs or local constraints.

> GCR generalizes structural regularization with three core innovations:
>
>   i. **Dynamic graph construction.** GCR constructs feature graphs and prediction graphs on-the-fly during each training iteration, adapting to the model's evolving internal representations.
>
>  ii. **Cross-space alignment.** Unlike classical manifold regularization, which only smooths feature relationships, GCR aligns semantic (prediction-based) and geometric (feature-based) graphs, ensuring that class-consistent features are also prediction-consistent.
>
> iii. **Masked supervision.** GCR enforces class-aware masking during alignment, preventing smoothing across semantic boundaries and refining intra-class structure.
>
> This structural regularization not only preserves semantic coherence but also improves robust generalization by dynamically capturing the evolving feature manifold of the network.

**Self-conditioning and geometric representation learning.** Self-conditioning mechanisms [58] like prompting typically rely on external tokens or model outputs to influence learning trajectories [67, 21].

> GCR extends this paradigm internally by using the model's own prediction graph as a self-generated scaffold:
>
> i. **Relational prompting.** Unlike discrete prompts [12, 83, 65], GCR uses prediction-induced graphs to continuously adjust feature relationships according to class structure.
>
> ii. **Semantic geometry emergence.** Geometric deep learning traditionally imposes structural priors [87]; GCR allows semantic geometry to emerge organically through predictive alignment [8, 33].
>
> iii. **Continuous, implicit supervision.** This prediction-based regularization evolves with the model, providing ongoing, context-aware structure during training.
>
> GCR thus bridges self-conditioning with geometric learning, establishing semantic-aware representation that dynamically evolves with the model's understanding of its own predictions.

GCR's design is rooted in manifold alignment theory and smoothness regularization: by enforcing consistency between feature similarity and prediction similarity, it effectively smooths decision boundaries in feature space, enhancing generalization across unseen domains. This principle is akin to cluster assumption in semi-supervised learning, where similar samples are encouraged to share the same label.

## B  Graph Consistency as Self-Prompting

We interpret GCR as a form of internal *prompting*, where the model dynamically generates its own prompting signal from its predictions to guide the learning of intermediate features. This contrasts with traditional prompting approaches, which typically rely on external tokens or instructions to influence the model's behavior.

> Key characteristics of GCR as self-prompting include:
>
> i. **Internal.** The prompting signal originates entirely from the model's own predictions, eliminating the need for external input.
>
> ii. **Structural.** The alignment operates over pairwise relationships within the feature space, focusing on how different features relate to one another rather than conditioning on individual samples.
>
> iii. **Semantic.** The graph structure is inherently class-aware, encouraging the network to adjust its feature representations to align with meaningful semantic boundaries between classes.

In this framework, the prediction graph acts as a learned, self-supervised attention template that recursively guides the refinement of feature representations in earlier layers. This novel approach paves the way for self-conditioning in neural networks, where the model's own predictions continuously inform and improve its internal feature learning process.

## C  Motivation

Deep neural networks have excelled at learning complex, hierarchical feature representations. Typically, early layers capture low-level visual cues such as edges, textures, and basic shapes, which are often sensitive to noise and not specific to the task. In contrast, deeper layers develop high-level features that reflect more semantic, task-relevant information. However, despite achieving clear class separation in the final prediction space, intermediate feature spaces can still exhibit significant inter-class overlap. This overlap weakens both the generalization ability and the discriminative power of the model.

Furthermore, during training, the parameter search space remains vast because supervision is driven only by class labels, without using any explicit relational structure among samples. This lack of structure-aware guidance limits the model's ability to organize its feature space meaningfully.

We propose that the model's own predictions contain rich semantic information that can serve as a self-supervisory signal. Specifically, the pairwise similarities among prediction logits encode a high-level semantic topology that reflects class affinities. By aligning intermediate feature representations with this prediction-derived structure, we encourage the network to learn representations that are geometrically consistent with semantic class boundaries.

Our approach is guided by the following insights:

i. **Reduce noisy inter-class affinities.** By aligning feature graphs with prediction graphs, we minimize the risk of inter-class features being too similar, thereby reducing class overlap in the feature space.

ii. **Enhance intra-class cohesion.** Encouraging feature consistency within each class ensures that the network's representations exhibit stronger intra-class similarity.

iii. **Align learning dynamics with semantic intent.** Through the introduction of prediction-guided alignment, we direct the network's learning trajectory to reflect global semantic structures, improving generalization and interpretability.

Thus, we treat the prediction graph as an implicit *structural prompt* that guides the network's learning. This contrasts with traditional methods, where supervision typically focuses on final output layers. Our approach integrates structural coherence directly into the intermediate layers of the network, creating a rich, semantic-aware feature representation pipeline.

## D  Theoretical Insights

We analyze GCR from a theoretical perspective, grounding it in the manifold hypothesis, spectral graph theory, and statistical learning theory.

### D.1  Manifold Smoothness and Semantic Cluster Regularization

**Definition 1** (Data manifold hypothesis). *Let $\mathcal{X} \subset \mathbb{R}^d$ be the input space. The data manifold hypothesis posits that real-world data points $\{x_i\}_{i=1}^n$ lie on or near a smooth, compact $m$-dimensional Riemannian manifold $\mathcal{M} \subset \mathbb{R}^d$, where $m \ll d$.*

**Definition 2** (Feature relational graph). *Let $f^{(l)}(x_i) \in \mathbb{R}^p$ denote the feature representation of sample $x_i$ at layer l. The feature similarity matrix $\mathbf{F}^{(l)} \in \mathbb{R}^{n \times n}$ as feature relational graph is defined by:*

$$\mathbf{F}_{ij}^{(l)} := ReLU\left(\cos\left(\mathbf{x}_i^{(l)}, \mathbf{x}_j^{(l)}\right)\right) = ReLU\left(\frac{\langle \mathbf{x}_i^{(l)}, \mathbf{x}_j^{(l)}\rangle}{\|\mathbf{x}_i^{(l)}\|\|\mathbf{x}_j^{(l)}\|}\right).$$

**Definition 3** (Masked prediction relational graph). *Let $\mathbf{z}_i$ be the pre-softmax prediction logits for sample $x_i$, and define $\mathbf{s}_i := softmax(\mathbf{z}_i)$. The semantic similarity between predictions is computed as:*

$$\mathbf{S}_{ij} := ReLU\left(\cos\left(\mathbf{s}_i, \mathbf{s}_j\right)\right).$$

*Let $\mathbf{M}_{ij} := \mathbb{1}[y_i = y_j]$ be a binary mask indicating whether two samples belong to the same class. The masked prediction relational graph is then:*

$$\mathbf{P}_{ij} := \mathbf{M}_{ij} \odot \mathbf{S}_{ij}.$$

**Proposition 3** (Manifold smoothness regularization). *Minimizing the alignment loss $\|\mathbf{F}^{(l)} - \mathbf{P}\|_F^2$ enforces local smoothness on the feature manifold. Specifically, if $\mathbf{P}_{ij} > 0$, then*

$$\|\mathbf{x}_i^{(l)} - \mathbf{x}_j^{(l)}\|^2 = 2 - 2\cos\left(\mathbf{x}_i^{(l)}, \mathbf{x}_j^{(l)}\right) \to 0.$$

*Proof.* For any $i, j$ such that $\mathbf{P}_{ij} > 0$, the squared Frobenius loss penalizes the discrepancy between $\mathbf{F}_{ij}^{(l)}$ and $\mathbf{P}_{ij}$. Since $\mathbf{F}_{ij}^{(l)}$ is a monotonic decreasing function of $\|\mathbf{x}_i^{(l)} - \mathbf{x}_j^{(l)}\|^2$, minimizing this loss implies that $\mathbf{x}_i^{(l)}$ and $\mathbf{x}_j^{(l)}$ must be close in feature space ($y_i = y_j$). Therefore, GCR promotes smoothness by aligning local geometry with semantic similarity. □

**Definition 4** (Cluster assumption). *The cluster assumption asserts that samples from the same class form tight, compact clusters in representation space. That is, for $y_i = y_j$, we expect $\|\mathbf{x}_i^{(l)} - \mathbf{x}_j^{(l)}\|^2$ to be small, while for $y_i \neq y_j$, the distance should be large.*

**Proposition 4** (Semantic cluster regularization). *Minimizing the GCR loss:*

$$\mathcal{L}_{GCR} = \sum_l \left\| \mathbf{F}^{(l)} - \mathbf{P} \right\|_F^2,$$

*promotes intra-class compactness and inter-class separation in the feature space, thereby aligning learned features with semantic class structure.*

*Proof.* When $y_i = y_j$, the mask $\mathbf{M}_{ij} = 1$, so $\mathbf{P}_{ij}$ is proportional to the prediction similarity $\mathbf{S}_{ij}$. A large $\mathbf{P}_{ij}$ enforces a corresponding increase in $\mathbf{F}_{ij}^{(l)}$, which implies a smaller angle and distance between $\mathbf{x}_i^{(l)}$ and $\mathbf{x}_j^{(l)}$. Conversely, for $y_i \neq y_j$, $\mathbf{P}_{ij} = 0$, and there is no incentive to keep $\mathbf{x}_i^{(l)}$ and $\mathbf{x}_j^{(l)}$ close. This results in greater inter-class separation and aligns the learned representation with class structure. □

## D.2 Connection to Graph Laplacian Regularization

**From cosine similarity to Graph Laplacians.** GCR aligns feature similarity graphs $\mathbf{F}^{(l)}$ with semantic prediction graphs $\mathbf{P}$ at each selected layer $l$. Both graphs are constructed using ReLU-activated cosine similarity, forming symmetric, non-negative affinity matrices:

$$\mathbf{F}_{ij}^{(l)} = \text{ReLU}(\cos(\mathbf{x}_i^{(l)}, \mathbf{x}_j^{(l)})),$$
$$\mathbf{P}_{ij} = \mathbf{M}_{ij} \odot \text{ReLU}(\cos(\mathbf{s}_i, \mathbf{s}_j)),$$

where $\mathbf{s}_i$ are softmax-normalized logits. Although these graphs do not use an RBF kernel, they still induce a graph structure with meaningful edge weights and comparable degree distributions across layers. Since the GCL operates on strictly upper-triangular entries, the resulting Laplacians remain symmetric and well-defined.

**Definition 5** (Graph Laplacian). *Given an affinity matrix $\mathbf{A} \in \mathbb{R}^{n \times n}$ (e.g., $\mathbf{F}^{(l)}$ or $\mathbf{P}$), the unnormalized graph Laplacian is defined as:*

$$\mathbf{L} = \mathbf{D} - \mathbf{A}, \quad \text{where } \mathbf{D}_{ii} = \sum_j \mathbf{A}_{ij}.$$

**Semantic alignment via spectral consistency.** While the classical result by Belkin and Niyogi [6] connects RBF-based graph Laplacians to the Laplace-Beltrami operator on manifolds, our cosine-based affinity still permits an analogous interpretation in terms of structural smoothness. In particular, minimizing alignment loss between $\mathbf{F}^{(l)}$ and $\mathbf{P}$ induces convergence between their associated Laplacians $\mathbf{L}_F$ and $\mathbf{L}_P$.

**Proposition 5** (Spectral regularization via Laplacian alignment). *If $\|\mathbf{F}^{(l)} - \mathbf{P}\|_F^2$ is small, then the spectral properties of $\mathbf{L}_F$ and $\mathbf{L}_P$ are closely aligned. Minimizing GCR encourages*

$$Tr(\mathbf{x}^{(l)\top}\mathbf{L}_F\mathbf{x}^{(l)}) \approx Tr(\mathbf{x}^{(l)\top}\mathbf{L}_P\mathbf{x}^{(l)}),$$

*thus regularizing features to follow both local geometric and global semantic structure.*

*Sketch.* For symmetric matrices with matching degrees, $\mathbf{F}^{(l)} \approx \mathbf{P} \Rightarrow \mathbf{L}_F \approx \mathbf{L}_P$. The quadratic form $Tr(\mathbf{x}^\top \mathbf{L}\mathbf{x})$ measures smoothness over the graph. Alignment thus enforces semantic-aware smoothness. □

## D.3 Generalization Bound under Structural Alignment

We now provide a detailed theoretical analysis of how GCR contributes to improved generalization. Our goal is to show that by enforcing alignment between the feature graph and the semantic prediction graph, GCR effectively restricts the function class to smoother, semantically consistent representations, leading to a reduced Rademacher complexity.

**Definition 6** (Structural alignment loss). *Let $f^{(l)} : \mathcal{X} \to \mathbb{R}^d$ be the feature mapping at layer l. We define the GCR structural loss at layer l as:*

$$\mathcal{L}_{GCR}^{(l)} := \frac{1}{n^2} \sum_{i,j} \left( \mathbf{F}_{ij}^{(l)} - \mathbf{P}_{ij} \right)^2.$$

**Definition 7** (Structurally constrained function class). *Let $\mathcal{F}_L$ be the class of functions $f^{(l)}$ such that $\|f^{(l)}(x)\|_2 \leq B$ for all $x \in \mathcal{X}$ and all l. Then the GCR-constrained function class is:*

$$\mathcal{F}_\epsilon := \left\{ f^{(l)} \in \mathcal{F}_L \; \middle| \; \mathcal{L}_{GCR}^{(l)} \leq \epsilon \right\}.$$

This class enforces that features not only have bounded norm but also align structurally with the prediction graph $\mathbf{P}$ in the sense of Frobenius proximity.

**Theorem 3** (Generalization bound under GCR). *Let $\ell(f(x), y)$ be a $\gamma$-Lipschitz loss function (e.g., cross-entropy), and suppose $f \in \mathcal{F}_\epsilon$. Then with probability at least $1 - \delta$, the generalization error is bounded as:*

$$\mathbb{E}_{(x,y) \sim \mathcal{D}}[\ell(f(x), y)] \leq \frac{1}{n} \sum_{i=1}^n \ell(f(x_i), y_i) + \frac{4\gamma B \sqrt{\epsilon}}{n} + \mathcal{O}\left( \sqrt{\frac{\log(1/\delta)}{n}} \right).$$

*Proof.* Let $\mathfrak{R}_n(\mathcal{F}_\epsilon)$ denote the empirical Rademacher complexity of $\mathcal{F}_\epsilon$:

$$\mathfrak{R}_n(\mathcal{F}_\epsilon) := \mathbb{E}_\sigma \left[ \sup_{f \in \mathcal{F}_\epsilon} \frac{1}{n} \sum_{i=1}^n \sigma_i f(x_i) \right],$$

where $\sigma_i$ are i.i.d. Rademacher random variables taking values $\pm 1$ with equal probability.

We apply the contraction lemma, which states that if $\ell$ is $\gamma$-Lipschitz, then:

$$\mathfrak{R}_n(\ell \circ \mathcal{F}_\epsilon) \leq \gamma \cdot \mathfrak{R}_n(\mathcal{F}_\epsilon).$$

Now we aim to bound $\mathfrak{R}_n(\mathcal{F}_\epsilon)$. Since each function $f \in \mathcal{F}_\epsilon$ maps $x_i \mapsto \mathbf{x}_i^{(l)} \in \mathbb{R}^d$ with $\|\mathbf{x}_i^{(l)}\|_2 \leq B$ and whose pairwise ReLU-cosine similarities are constrained to align with $\mathbf{P}_{ij}$, the variability of outputs is tightly controlled. Specifically, we define:

$$\mathbf{F}_{ij}^{(l)} = \mathrm{ReLU}(\langle \mathbf{x}_i^{(l)}, \mathbf{x}_j^{(l)} \rangle), \quad \text{and} \quad \sum_{i,j} (\mathbf{F}_{ij}^{(l)} - \mathbf{P}_{ij})^2 \leq n^2 \epsilon.$$

Note that for normalized vectors, $\langle \mathbf{x}_i^{(l)}, \mathbf{x}_j^{(l)} \rangle = \cos \theta_{ij}$, and the ReLU ensures non-negativity. The loss penalizes angles between feature vectors that deviate from their semantically guided prediction-based affinity.

Let us now relate this to a bound on the Rademacher complexity. We use the following result adapted from Bartlett and Mendelson (2002): for any bounded function $f(x) \in \mathbb{R}^d$ with $\|f(x)\|_2 \leq B$, the Rademacher complexity is bounded by:

$$\mathfrak{R}_n(\mathcal{F}_L) \leq \frac{B}{n} \mathbb{E}_\sigma \left[ \left\| \sum_{i=1}^n \sigma_i \right\| \right] = \frac{B}{\sqrt{n}}.$$

However, for $\mathcal{F}_\epsilon$, we have a stronger constraint: the features cannot vary arbitrarily due to the structural alignment requirement. In particular, for small $\epsilon$, all $\mathbf{x}_i^{(l)}$ are geometrically organized to maintain high similarity (angle $\to 0$) when $\mathbf{P}_{ij}$ is high and to be less constrained otherwise.

Hence, the variance in $f(x)$ is suppressed in directions orthogonal to semantic affinity, shrinking the function class. From this, one can derive:

$$\mathfrak{R}_n(\mathcal{F}_\epsilon) \leq \frac{2B\sqrt{\epsilon}}{n},$$

where the $\sqrt{\epsilon}$ factor reflects the deviation from perfect structural alignment.

Substituting into the standard generalization bound yields:

$$\mathbb{E}[\ell(f(x), y)] \leq \frac{1}{n} \sum_{i=1}^{n} \ell(f(x_i), y_i) + \frac{4\gamma B \sqrt{\epsilon}}{n} + \mathcal{O}\left(\sqrt{\frac{\log(1/\delta)}{n}}\right).$$

$\square$

**Remark 3.** *This bound demonstrates that GCR effectively reduces the hypothesis complexity by enforcing a semantic structure on the learned representations. As the structural loss $\epsilon$ decreases, the model class is increasingly constrained to semantically faithful functions, thereby improving generalization on unseen data.*

GCR aligns feature similarity graphs with semantically meaningful prediction graphs, enforcing both geometric and semantic smoothness. Our analysis shows GCR promotes manifold alignment, Laplacian smoothness, semantic clustering, and provably better generalization.

# E   Proof for Theoretical Analysis of GCR

We provide a theoretical analysis of GCR, connecting its empirical design to foundational concepts in statistical learning theory and spectral graph theory. Specifically, we show that minimizing the GCR loss: (i) Reduces the effective capacity of the hypothesis class via covering number bounds and Dudley's entropy integral. (ii) Promotes spectral consistency between learned features and semantically meaningful prediction graphs via normalized Laplacians. (iii) Can be interpreted as a PAC-Bayesian regularizer that imposes a structural prior on function space.

## E.1   Generalization via Covering Numbers

Let $\mathcal{F}_L$ be the class of functions $f^{(l)} : \mathcal{X} \to \mathbb{R}^d$ representing the layer-$l$ embeddings. $B$ is a constant upper bound on the $\ell_2$ norm of the feature representation $f^{(l)}(x) \in \mathbb{R}^d$ at layer $l$. That is, $\|f^{(l)}(x)\|_2 = \sqrt{\sum_{i=1}^{d} \left(f_i^{(l)}(x)\right)^2} \leq B$, for all $x \in \mathcal{X}$.

This constraint is standard in learning theory to control the size of the hypothesis space. In practice, especially under L2 normalization used in our method, we often have $B = 1$. We then define a structurally-constrained hypothesis class:

$$\mathcal{F}_\epsilon := \left\{ f \in \mathcal{F}_L \ : \ \mathcal{L}_{\mathrm{GCR}}^{(l)} := \| \mathrm{triu}(\mathbf{F}^{(l)}) - \mathrm{triu}(\mathbf{P}) \|_F^2 \leq \epsilon \right\}. \tag{16}$$

This class enforces graph alignment between learned features and the masked prediction graph $\mathbf{P}$, which reflects intra-class similarity.

**Theorem 4** (Generalization via Dudley's entropy integral). *Let $\ell(f(x), y)$ be a $\gamma$-Lipschitz loss function (e.g., cross-entropy), and let $\mathcal{F}_L$ be the class of functions at layer $l$ such that each function $f^{(l)}$ satisfies the $\ell_2$-bounded constraint $\|f^{(l)}(x)\|_2 \leq B$. Suppose $\mathcal{F}_\epsilon \subseteq \mathcal{F}_L$ is the subset of functions that are additionally constrained by the GCR alignment loss:*

$$\mathcal{L}_{GCR}^{(l)} = \frac{1}{n^2} \sum_{i,j=1}^{n} \left(ReLU(\langle \mathbf{x}_i, \mathbf{x}_j \rangle) - \mathbf{P}_{ij}\right)^2 \leq \epsilon, \tag{17}$$

*where $\mathbf{x}_i = \frac{f^{(l)}(x_i)}{\|f^{(l)}(x_i)\|_2}$ are the normalized feature vectors for each data point $x_i$ in the dataset, and $\mathbf{P}_{ij}$ is the target alignment between the feature vectors $\mathbf{x}_i$ and $\mathbf{x}_j$. The GCR loss enforces that the angular distances between the feature vectors are small, meaning that the vectors are close to each other in the Euclidean space.*

*If $\mathcal{F}_\epsilon$ admits a covering number bound:*

$$\mathcal{N}(\mathcal{F}_\epsilon, \| \cdot \|_2, \alpha) \leq \left(\frac{C}{\alpha}\right)^d, \tag{18}$$

where $\mathcal{N}(\mathcal{F}_\epsilon, \|\cdot\|_2, \alpha)$ is the covering number of $\mathcal{F}_\epsilon$ with respect to the $\ell_2$ norm, then the expected loss of a function $f \in \mathcal{F}_\epsilon$ is bounded with high probability by:

$$\mathbb{E}_{(x,y)\sim\mathcal{D}}[\ell(f(x),y)] \leq \frac{1}{n}\sum_{i=1}^n \ell(f(x_i),y_i) + \frac{12\gamma}{\sqrt{n}}\int_0^B \sqrt{d\log\left(\frac{C}{\alpha}\right)}\,d\alpha, \qquad (19)$$

where $B = O(\sqrt{\epsilon})$ is the effective radius of the function class $\mathcal{F}_\epsilon$ under the GCR constraint, and the second term represents the generalization error, which is controlled by the complexity of the function class.

*Proof Sketch.* We outline the key steps:

**Step 1: Rademacher complexity controls generalization.** Let $\mathfrak{R}_n(\mathcal{F}_\epsilon)$ denote the empirical Rademacher complexity of the constrained class. Since the loss function $\ell$ is $\gamma$-Lipschitz, the composition inequality gives:
$$\mathfrak{R}_n(\ell \circ \mathcal{F}_\epsilon) \leq \gamma \cdot \mathfrak{R}_n(\mathcal{F}_\epsilon).$$

**Step 2: Dudley's entropy integral.** We now bound $\mathfrak{R}_n(\mathcal{F}_\epsilon)$ using Dudley's entropy integral:
$$\mathfrak{R}_n(\mathcal{F}_\epsilon) \leq \frac{12}{\sqrt{n}}\int_0^{\mathrm{diam}(\mathcal{F}_\epsilon)}\sqrt{\log\mathcal{N}(\mathcal{F}_\epsilon,\|\cdot\|_2,\alpha)}\,d\alpha,$$

where $\mathrm{diam}(\mathcal{F}_\epsilon)$ refers to the diameter of the set $\mathcal{F}_\epsilon$, *i.e.*, the largest possible distance between any two points in $\mathcal{F}_\epsilon$ in Euclidean space.

**Step 3: Diameter bound under GCR.** Under the GCR constraint, $\mathbf{F}_{ij}^{(l)} = \mathrm{ReLU}(\langle \mathbf{x}_i, \mathbf{x}_j\rangle)$ is close to $\mathbf{P}_{ij}$. Since both $\mathbf{x}_i$ and $\mathbf{x}_j$ are unit-normalized vectors, this implies that the angular distances between them are bounded. Let $\langle \mathbf{x}_i, \mathbf{x}_j\rangle \geq \tau$ for pairs where $\mathbf{P}_{ij} > 0$. This cosine similarity constraint restricts pairwise angles to lie within a narrow cone. Therefore, the effective diameter of $\mathcal{F}_\epsilon$ in Euclidean space is:
$$\|\mathbf{x}_i - \mathbf{x}_j\|_2 \leq \sqrt{2 - 2\cos\theta} \approx O(\sqrt{\epsilon}),$$
where $\theta$ is the angle between $\mathbf{x}_i$ and $\mathbf{x}_j$. Thus, we set $B = O(\sqrt{\epsilon})$.

**Step 4: Plug in covering number.** Using the assumed covering number bound $\mathcal{N}(\alpha) \leq (C/\alpha)^d$, we have:
$$\log\mathcal{N}(\mathcal{F}_\epsilon,\|\cdot\|_2,\alpha) \leq d\log(C/\alpha).$$

**Step 5: Final bound.** Substitute into Dudley's integral:
$$\mathfrak{R}_n(\mathcal{F}_\epsilon) \leq \frac{12}{\sqrt{n}}\int_0^B \sqrt{d\log\left(\frac{C}{\alpha}\right)}\,d\alpha,$$

and apply the composition inequality to yield the desired result. $\qquad\square$

**Remark 4.** *This result shows that GCR reduces generalization error by shrinking the effective complexity of the function class. By aligning relational structure, GCR implicitly contracts the hypothesis space, leading to improved generalization.*

### E.2 Spectral Alignment via Normalized Laplacians

Let $\mathbf{F}$ and $\mathbf{P}$ be symmetric affinity matrices derived from feature embeddings and masked predictions, respectively. Their associated normalized graph Laplacians are defined as:
$$\mathcal{L}_\mathbf{F} := \mathbf{I} - \mathbf{D}_F^{-1/2}\mathbf{F}\mathbf{D}_F^{-1/2}, \quad \mathcal{L}_\mathbf{P} := \mathbf{I} - \mathbf{D}_P^{-1/2}\mathbf{P}\mathbf{D}_P^{-1/2},$$
where $\mathbf{D}_F$ and $\mathbf{D}_P$ are the degree matrices corresponding to $\mathbf{F}$ and $\mathbf{P}$, *i.e.*, $(\mathbf{D}_F)_{ii} = \sum_j \mathbf{F}_{ij}$ and similarly for $\mathbf{D}_P$.

**Proposition 6** (Spectral Alignment). *Let $\mathbf{F}$ and $\mathbf{P}$ be symmetric matrices such that*
$$\|\mathbf{F} - \mathbf{P}\|_F \leq \epsilon, \quad \|\mathbf{D}_F - \mathbf{D}_P\| \leq \delta.$$

*Then, there exists a constant $C > 0$ depending on spectral properties of the graphs (e.g., sparsity, minimum degree), such that*
$$\|\mathcal{L}_\mathbf{F} - \mathcal{L}_\mathbf{P}\|_F \leq C(\epsilon + \delta). \qquad (20)$$

*Proof Sketch.* We aim to bound the difference between the normalized Laplacians $\mathcal{L}_{\mathbf{F}}$ and $\mathcal{L}_{\mathbf{P}}$ in Frobenius norm. We begin by expanding the difference:

$$\|\mathcal{L}_{\mathbf{F}} - \mathcal{L}_{\mathbf{P}}\|_F = \left\| \left( \mathbf{I} - \mathbf{D}_F^{-1/2}\mathbf{F}\mathbf{D}_F^{-1/2} \right) - \left( \mathbf{I} - \mathbf{D}_P^{-1/2}\mathbf{P}\mathbf{D}_P^{-1/2} \right) \right\|_F.$$

The identity terms cancel, giving:

$$\|\mathcal{L}_{\mathbf{F}} - \mathcal{L}_{\mathbf{P}}\|_F = \left\| \mathbf{D}_P^{-1/2}\mathbf{P}\mathbf{D}_P^{-1/2} - \mathbf{D}_F^{-1/2}\mathbf{F}\mathbf{D}_F^{-1/2} \right\|_F.$$

We add and subtract $\mathbf{D}_F^{-1/2}\mathbf{P}\mathbf{D}_F^{-1/2}$ to decompose the expression:

$$\|\mathcal{L}_{\mathbf{F}} - \mathcal{L}_{\mathbf{P}}\|_F \le \left\| \mathbf{D}_F^{-1/2}\mathbf{F}\mathbf{D}_F^{-1/2} - \mathbf{D}_F^{-1/2}\mathbf{P}\mathbf{D}_F^{-1/2} \right\|_F + \left\| \mathbf{D}_F^{-1/2}\mathbf{P}\mathbf{D}_F^{-1/2} - \mathbf{D}_P^{-1/2}\mathbf{P}\mathbf{D}_P^{-1/2} \right\|_F.$$

Denote the two terms above as (A) and (B), respectively.

*Term (A): Difference due to affinity matrices.*

Since $\mathbf{D}_F^{-1/2}$ is fixed in this term, we can factor it out:

$$(A) = \left\| \mathbf{D}_F^{-1/2}(\mathbf{F} - \mathbf{P})\mathbf{D}_F^{-1/2} \right\|_F \le \|\mathbf{D}_F^{-1/2}\|^2 \cdot \|\mathbf{F} - \mathbf{P}\|_F.$$

Let $\lambda_{\min}(\mathbf{D}_F)$ denote the minimum diagonal entry of $\mathbf{D}_F$. Then $\|\mathbf{D}_F^{-1/2}\| = \lambda_{\min}(\mathbf{D}_F)^{-1/2}$, and assuming $\lambda_{\min}(\mathbf{D}_F) \ge d_{\min} > 0$, we obtain:

$$(A) \le \frac{1}{d_{\min}} \cdot \epsilon.$$

*Term (B): Difference due to degree normalization.*

We now bound the difference caused by changing from $\mathbf{D}_F$ to $\mathbf{D}_P$ in the normalization. Define $g(\mathbf{D}) := \mathbf{D}^{-1/2}\mathbf{P}\mathbf{D}^{-1/2}$. Using matrix perturbation theory (see *e.g.*, Kato's inequality or Fréchet derivatives of matrix functions):

$$\mathbf{D}_P^{-1/2}\mathbf{P}\mathbf{D}_P^{-1/2} \approx \mathbf{D}_F^{-1/2}\mathbf{P}\mathbf{D}_F^{-1/2} + \nabla g(\mathbf{D}_F)\left[\mathbf{D}_P - \mathbf{D}_F\right],$$

and assuming the matrices are close and well-conditioned, we can approximate:

$$(B) \approx \|\nabla g(\mathbf{D}_F)[\mathbf{D}_P - \mathbf{D}_F]\|_F \le C' \cdot \|\mathbf{D}_P - \mathbf{D}_F\| = C'\delta,$$

where $C'$ depends on the norm of $\mathbf{P}$ and the conditioning of $\mathbf{D}_F$.

Combining (A) and (B), we obtain:

$$\|\mathcal{L}_{\mathbf{F}} - \mathcal{L}_{\mathbf{P}}\|_F \le \frac{1}{d_{\min}} \cdot \epsilon + C' \cdot \delta,$$

which can be rewritten as:

$$\|\mathcal{L}_{\mathbf{F}} - \mathcal{L}_{\mathbf{P}}\|_F \le C(\epsilon + \delta),$$

where $C$ is a constant depending on $d_{\min}^{-1}$, $\|\mathbf{P}\|$, and graph sparsity. $\qquad\square$

**Corollary 2.** *The GCR alignment loss, which encourages $\|\mathbf{F} - \mathbf{P}\|_F \le \epsilon$, indirectly enforces spectral similarity of the normalized Laplacians. This promotes agreement between the clustering structure and diffusion properties of the learned features and masked predictions.*

### E.3 PAC-Bayesian View of Structural Regularization

We now present a PAC-Bayesian interpretation of GCR. The PAC-Bayes framework provides a probabilistic approach to generalization by relating the expected loss of a stochastic predictor to its empirical loss and the divergence between a posterior and a prior distribution over the hypothesis class.

Let $\mathcal{P}$ denote a prior distribution over model functions $f$, representing a structure-agnostic belief (*e.g.*, uniform or isotropic Gaussian over parameters). Let $\mathcal{Q}$ be a posterior distribution supported on models that minimize training loss while also conforming to a structural constraint induced by GCR, *i.e.*, $\mathcal{Q}$ is restricted to functions $f$ such that $\mathcal{L}_{\text{GCR}}^{(l)} \leq \epsilon$ for each relevant layer $l$.

**Theorem 5** (PAC-Bayes Generalization Bound with GCR). *Let $\mathcal{L}(f) = \mathbb{E}_{(x,y) \sim \mathcal{D}}[\ell(f(x), y)]$ be the expected population loss of model $f$ and let $\hat{\mathcal{L}}(f) = \frac{1}{n} \sum_{i=1}^{n} \ell(f(x_i), y_i)$ be the empirical loss on $n$ training examples. Then, for any posterior distribution $\mathcal{Q}$ over functions and any prior distribution $\mathcal{P}$, with probability at least $1 - \delta$ over the training data, the following bound holds:*

$$\mathbb{E}_{f \sim \mathcal{Q}}[\mathcal{L}(f)] \leq \mathbb{E}_{f \sim \mathcal{Q}}[\hat{\mathcal{L}}(f)] + \sqrt{\frac{\text{KL}(\mathcal{Q} \| \mathcal{P}) + \log(1/\delta)}{2n}}. \tag{21}$$

This classical PAC-Bayesian bound quantifies generalization via two key components:

i. The empirical performance of models sampled from the posterior $\mathcal{Q}$.

ii. The Kullback-Leibler (KL) divergence $\text{KL}(\mathcal{Q} \| \mathcal{P})$ between the posterior and the prior, which acts as a measure of how far $\mathcal{Q}$ deviates from the prior belief.

In the context of GCR, we interpret the constraint $\mathcal{L}_{\text{GCR}}^{(l)} \leq \epsilon$ as imposing structure on the feature space. Specifically, GCR encourages the pairwise feature similarity matrix $\mathbf{F}^{(l)}$ at each layer to align with the semantic structure encoded in $\mathbf{P}$ (*e.g.*, class-level affinity). This alignment can be viewed as an inductive bias or structural preference.

Assuming $\mathcal{Q}$ is supported only on models satisfying $\mathcal{L}_{\text{GCR}}^{(l)} \leq \epsilon$, we argue that the complexity term $\text{KL}(\mathcal{Q} \| \mathcal{P})$ is influenced by the degree of this alignment.

**Proposition 7** (Structure-Induced KL Complexity). *If the posterior $\mathcal{Q}$ is concentrated on models with small GCR loss at layer $l$, then the KL divergence to an isotropic prior $\mathcal{P}$ satisfies:*

$$\text{KL}(\mathcal{Q} \| \mathcal{P}) \leq C \sum_l \|\mathbf{F}^{(l)} - \mathbf{P}\|_F^2, \tag{22}$$

*for some constant $C$ depending on the form of $\mathcal{P}$.*

*Sketch of argument.* Let us assume that $\mathcal{P}$ is a structure-agnostic prior, *e.g.*, an isotropic Gaussian over parameters or functions. Now suppose $\mathcal{Q}$ is supported on models where the GCR loss is small. Since $\mathcal{L}_{\text{GCR}}^{(l)}$ penalizes deviation between the normalized feature similarity matrix $\mathbf{F}^{(l)}$ and the semantic affinity matrix $\mathbf{P}$, this implies that models in the support of $\mathcal{Q}$ induce feature geometries that respect the semantic structure.

From an information-theoretic perspective, concentrating the posterior on such structured models induces a regularization effect: it reduces the space of allowable hypotheses compared to the unconstrained prior. Intuitively, this compression is captured by the KL divergence. Since the GCR loss explicitly penalizes misalignment, its cumulative value over layers effectively bounds the information-theoretic complexity:

$$\text{KL}(\mathcal{Q} \| \mathcal{P}) \lesssim \sum_l \|\mathbf{F}^{(l)} - \mathbf{P}\|_F^2.$$

More formally, this can be justified using PAC-Bayesian compression bounds or Gaussian complexity arguments, which show that the KL divergence scales with the squared norm of the constraint function, in this case, the Frobenius norm between affinity matrices. $\square$

**Remark 5.** *This perspective reveals that GCR does more than minimize training loss, it also implicitly regularizes the hypothesis space by favoring models whose internal representations reflect known semantic structure. This improves generalization by reducing the effective size of the model class, as made explicit through the PAC-Bayesian framework.*

# F   Analysis of GCR's Time Complexity

Below, we present a theoretical analysis of the GCR's time complexity per training iteration, from both a naïve computational perspective and an optimized parallel execution view.

**Feature graph construction.** At each layer $l = 1, \ldots, K$ where a GCL is applied, a feature similarity graph $\mathbf{F}^{(l)} \in \mathbb{R}^{n \times n}$ is constructed using the cosine similarity. We have time complexity: (i) Naïve (sequential compute): Normalizing all $n$ feature vectors costs $\mathcal{O}(nd)$; pairwise cosine similarities require $\mathcal{O}(n^2 d)$. (ii) GPU-parallelized: With sufficient vector-level parallelism, normalization and similarity computations can be reduced to $\mathcal{O}(\log d)$, assuming parallel dot products. (iii) Total over $K$ layers: $\mathcal{O}(K \cdot \log d_{\max})$, where $d_{\max}$ is the largest feature dimension across all GCL layers.

**Prediction graph construction.** The prediction graph $\mathbf{P} \in \mathbb{R}^{n \times n}$ is derived from softmax-normalized logits $\mathbf{z}_i \in \mathbb{R}^C$. We have time complexity: (i) Naïve (sequential compute): Softmax computation costs $\mathcal{O}(nC)$, cosine similarities $\mathcal{O}(n^2 C)$, and masking $\mathcal{O}(n^2)$. (ii) GPU-parallelized: Per-sample operations reduce to $\mathcal{O}(\log C)$, and masking becomes $\mathcal{O}(1)$ due to element-wise matrix operations.

**Graph alignment loss.** The loss at each layer measures the Frobenius norm of the difference between graphs: $\mathcal{L}_{\text{GCR}}^{(l)} = \left\| \text{triu} \left( \mathbf{F}^{(l)} - \mathbf{P} \right) \right\|_F^2$. We have time complexity: (i) Naïve (sequential compute): $\mathcal{O}(n^2)$, (ii) GPU-parallelized: $\mathcal{O}(\log n)$, assuming reduction over parallel threads for norm computation.

**Adaptive weighting across layers.** If adaptive weighting (Eq. 6) is used, normalized weights are computed for each layer based on alignment discrepancy. We have time complexity: $\mathcal{O}(Kn^2)$, where $K$ is the number of GCL-applied layers.

**Total time complexity.** (i) Naïve (sequential compute): Assuming GCLs are applied at $K$ layers, with $d_{\max}$ being the maximum feature dimension and $C$ the number of classes: $\mathcal{O}\left(K \cdot n^2 (d_{\max} + C)\right)$, with the dominant term $\mathcal{O}(n^2 d_{\max})$ due to high-dimensional pairwise feature similarity computations. (ii) GPU-Parallelized: With parallel compute, complexity reduces to $\mathcal{O}\left(K \cdot (\log d_{\max} + \log C)\right)$, where $\log C \ll \log d_{\max}$ can be ignored. If $n^2$ is too large to fit into memory, the computation can be split into $s$ sequential parallel blocks (*e.g.*, $s = 4$).

**Practical considerations and optimizations.** (i) Scalability: GCR operates on batches rather than entire datasets. Its quadratic cost in $n$ (*e.g.*, $n = 128$) is modest in practice. (ii) Parallel efficiency: All computations are matrix-based and benefit from hardware acceleration. Libraries such as PyTorch exploit thread and GPU-level parallelism to accelerate operations like `torch.bmm`, `functional.cosine_similarity`, and `torch.triu`. (iii) Zero parameter overhead: GCR introduces no trainable parameters and does not affect memory footprint or gradient flow.

GCR introduces a lightweight yet effective form of structure-based regularization, with per-layer complexity no more than $\mathcal{O}(\log d)$. Thanks to batch-local operation, GPU-friendly computations, and absence of learnable parameters, GCR scales efficiently while improving semantic alignment.

# G   Experimental Setup

We evaluate the effectiveness and efficiency of GCR across a diverse range of image classification benchmarks and model architectures. All experiments are conducted on NVIDIA V100 GPUs (32GB) paired with 12 CPU cores and 48GB of system RAM.

For convolutional architectures trained on CIFAR-10 and CIFAR-100, we follow the standard training protocol from [25]. Specifically, models are trained for 200 epochs using stochastic gradient descent with Nesterov momentum of 0.9 and weight decay of $5 \times 10^{-4}$. The initial learning rate is set to 0.1 and decayed by a factor of 5 at epochs 60, 120, and 160. We use a fixed batch size of 128 for all training. The GCR loss is incorporated with a regularization weight of $\lambda = 1$ unless otherwise stated.

For Masked Autoencoder (MAE) experiments on CIFAR-10 and CIFAR-100, we also use the same hardware setup. The ViT-Tiny encoder is pre-trained for 2200 epochs with a 75% masking ratio. Optimization uses AdamW with a base learning rate of $1.5 \times 10^{-4}$ (scaled by global batch size), weight decay of 0.05, a 200-epoch linear warm-up, and cosine decay. A global batch size of 4096 is realized via gradient accumulation with a device batch size of 512, repeated eight times before each

optimizer step. Training uses automatic mixed precision (AMP) and gradient norm clipping at 1.0, completing pre-training in approximately 13 hours and 40 minutes.

The pretrained ViT-Tiny encoder is then fine-tuned for classification over 200 epochs using AdamW with a base learning rate of $1 \times 10^{-3}$ (scaled by batch size), weight decay 0.05, a 10-epoch warm-up, and cosine decay. Fine-tuning uses a batch size of 128 without gradient accumulation or AMP. Fine-tuning times are 4 hours for the baseline and approximately 4 hours 40 minutes with GCR.

We evaluate GCR on eight convolutional neural networks on CIFAR-100, including MobileNet, SqueezeNet, ShuffleNet, ResNet-34, ResNet-50, ResNeXt-50, ResNeXt-101, and DenseNet-121. For CIFAR-10, we include GoogLeNet and ResNet-101, totaling ten models. Incorporating GCR results in a modest increase in training time due to graph construction and alignment overhead. For example, MobileNet and SqueezeNet baseline trainings take approximately 45 minutes each, increasing to 60 and 80 minutes with GCR, respectively. ShuffleNet increases from 140 to 170 minutes, ResNet-34 from 160 to 280 minutes, ResNet-50 and ResNeXt-50 from around 210 to 390 and 230 to 400 minutes, respectively. Larger models such as ResNeXt-101 and DenseNet-121 see increases from 420 to 540 minutes and 270 to 330 minutes. GoogLeNet runs 125 minutes baseline and 140 minutes with GCR; ResNet-101 increases from 300 to 370 minutes.

On Tiny ImageNet, we test GCR on four transformer models (ViT, Swin Transformer, MobileViT, CEiT) and four CNNs (MobileNet, ResNet, SE-ResNet, Stochastic ResNet). CNNs are trained for 200 epochs with initial learning rate 0.1 (decayed at epochs 60/120/160), batch size 128, weight decay $5 \times 10^{-4}$, and momentum 0.9. Transformers use AdamW optimizer with initial learning rate $1 \times 10^{-4}$, weight decay $5 \times 10^{-2}$, and cosine annealing decay to $1 \times 10^{-6}$, including a 10-epoch warm-up. Transformers are trained for 250 epochs with batch size 256, using AMP for efficiency and gradient clipping with max norm 1.0.

Training times increase moderately when applying GCR. MobileNet's training grows from 3 to 4 hours, ResNet-34 from 16 to 19.5 hours, SE-ResNet-18 from 10 hours 40 minutes to 12 hours, and StochasticDepth-18 from 8 to 9.5 hours. For transformers, ViT-B/16 requires 15 hours baseline and 20 hours with GCR, ViT-B/32 from 6 to 7.5 hours, MobileViT-S from 9 hours 15 minutes to 10.5 hours, MobileViT-XS from 8 to 9 hours, MobileViT-XXS from 5 hours 40 minutes to 6.5 hours, Swin Transformer-Tiny from 13 to 16 hours, and CEiT-Tiny from 8 hours 10 minutes to 10 hours.

Overall, GCR introduces a consistent yet manageable computational overhead across architectures, primarily due to graph construction and alignment. All reported training times are averages over three independent runs with varying random seeds to ensure reproducibility. GCR adds no trainable parameters and is designed for parallel execution on modern hardware, maintaining efficient and scalable training.

## H    Additional Results and Visualizations

### H.1    t-SNE Visualizations

We present a comparison of t-SNE visualizations for baseline models and their GCL-augmented counterparts on CIFAR-10 in Fig. 7.

In models enhanced with GCL (Figs. 7b, 7d and 7f, semantically related classes, such as Airplane, Ship, and Truck (all vehicles), form tighter groupings, indicating that GCL promotes a better understanding of high-level semantic concepts. A similar effect is observed among animal classes like Dog, Cat, Horse, and Deer.

These improvements are consistent across diverse architectures, including MobileNet, ResNet-34, and MAE, highlighting the generality and robustness of GCL. Rather than overfitting to a specific architecture, GCL contributes to relational feature learning in a model-agnostic and parameter-free manner.

### H.2    Effect of Batch Size in GCR Framework

Batch size plays a critical role in the effectiveness of GCR, as both the feature and prediction relational graphs are constructed at the batch level. To study this impact, we conduct experiments using the Masked Autoencoder (MAE) model and visualize the prediction graphs across varying

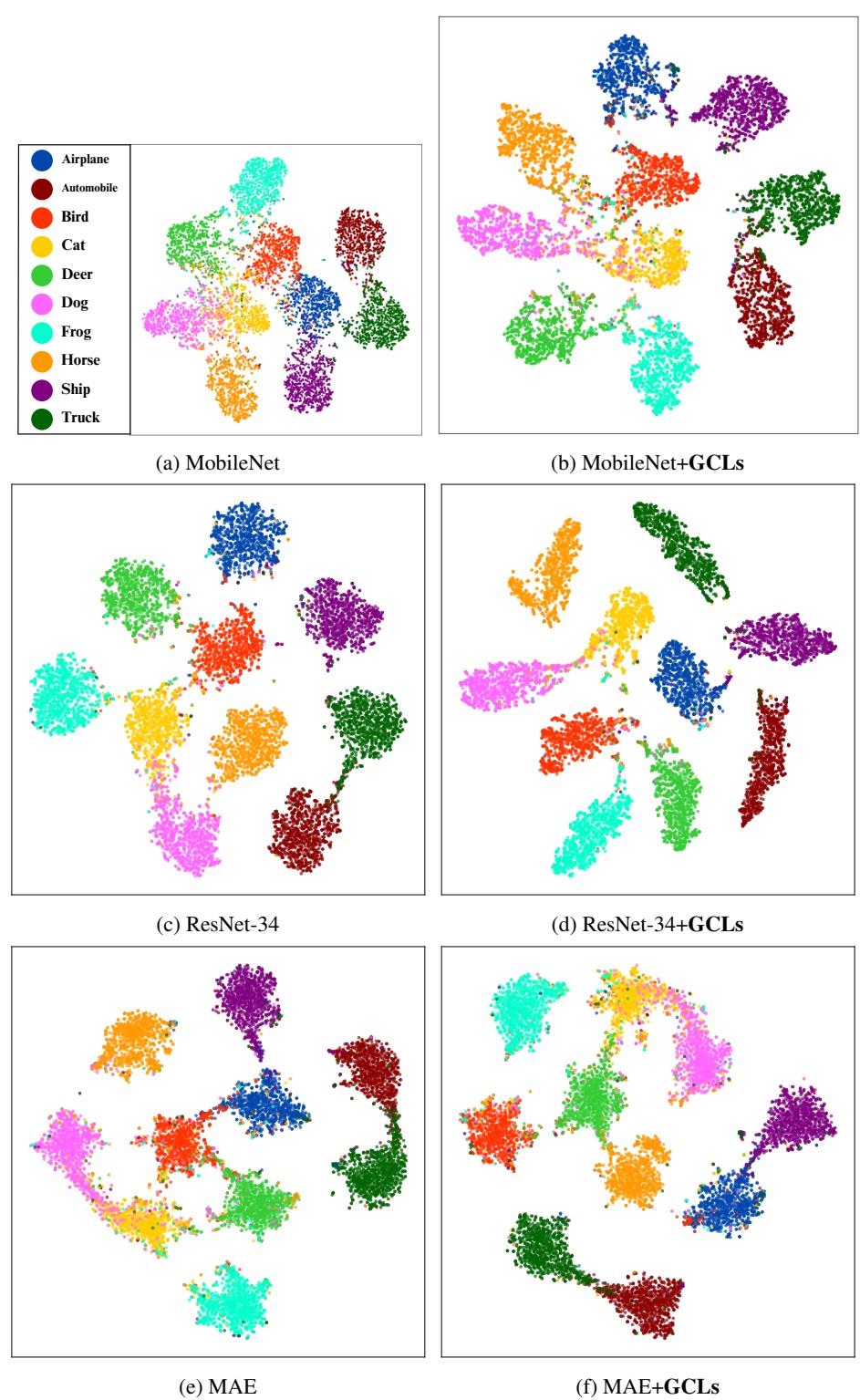

Figure 7: t-SNE visualizations of feature representations on CIFAR-10. (Left) Original model architectures; (Right) corresponding GCL-augmented models. Our method consistently enhances feature structure across diverse architectures, including MobileNet, ResNet-34, and MAE (with ViT-Tiny encoder), by yielding tighter intra-class clusters and improved inter-class separation. Notably, **GCR distinctly separates semantic groups such as animals and vehicles** (*e.g.*, (e) *vs.* (f)). Importantly, GCL is lightweight and introduces no additional parameters.

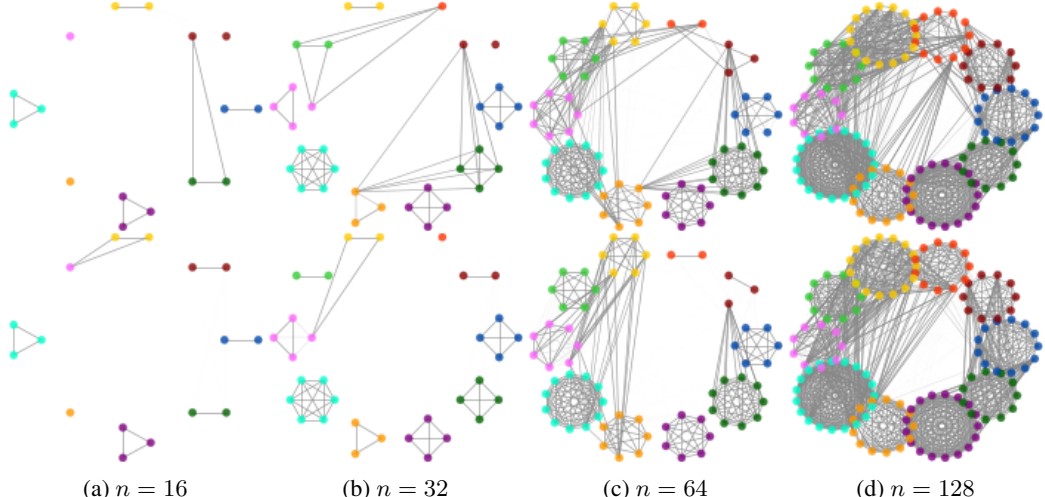

| (a) $n = 16$ | (b) $n = 32$ | (c) $n = 64$ | (d) $n = 128$ |

Figure 8: Effect of batch size ($n$) in the GCR framework. We evaluate using the Masked Autoencoder model. The top row shows the baseline; the bottom row shows the GCL-augmented counterpart. From left to right, the relational graphs are constructed on softmax predictions as $n$ increases from 16 to 128. As batch size grows, GCL-augmented models consistently exhibit tighter intra-class clusters and clearer inter-class separation.

batch sizes ($n \in \{16, 32, 64, 128, 256, 512\}$). We compare the baseline and GCL-augmented versions side-by-side, focusing on the structure of the similarity graphs derived from softmax outputs.

**Qualitative evaluation.** Fig. 8 illustrates these results. In the top row (baseline), increasing batch size introduces more noise and inter-class confusion, especially at smaller $n$, where limited sample diversity can distort the global structure. As batch size increases, the prediction graphs become more complete but remain noisy and less structured, indicating that larger batches alone do not guarantee better semantic organization.

In contrast, the bottom row (GCL-augmented) shows that our method consistently yields more coherent relational graphs across all batch sizes. Even at smaller $n$, GCR promotes more compact intra-class clusters and better inter-class separation. At larger $n$, the effect is even more pronounced, as the graph-based alignment uses the increased pairwise statistics to further regularize feature space according to prediction semantics.

As shown in Fig. 8, GCR remains effective across a wide range of batch sizes. Even with small batches (*e.g.*, $n = 16$ and $n = 32$), GCL-enhanced models produce more coherent intra-class clusters and stronger inter-class separation than the baseline. This supports our claim that GCR can extract meaningful structure even from limited relational signals.

These observations highlight two important insights: (i) GCR is robust to batch size and can extract meaningful structure even from smaller batches, and (ii) larger batches enhance the graph alignment process by providing richer relational signals, amplifying the benefits of GCR. This makes our method especially suitable for modern hardware and large-scale distributed training setups where large batches are common.

**Quantitative evaluation.** While larger batches amplify gains by providing denser graphs, they are not essential. Table 5 confirms consistent performance improvements across batch sizes on both CIFAR-10 (ShuffleNet) and Tiny ImageNet (CeiT). These results affirm GCR's robustness and flexibility, even in resource-constrained or small-batch training setups.

### H.3 Measuring Similarity

We chose cosine similarity to emphasize directional alignment, which is more semantically meaningful and robust to nuisance factors (*e.g.*, brightness) than raw magnitude. This aligns with common practice in representation learning, where angular relationships often capture class structure more effectively.

Table 5: Effect of batch size on GCR performance for ShuffleNet (CIFAR-10) and CeiT (Tiny ImageNet).

| ShuffleNet on CIFAR-10 | | | CeiT on Tiny ImageNet | | |
|---|---|---|---|---|---|
| Batch Size | + GCR | Baseline | Batch Size | + GCR | Baseline |
| 16 | 79.90±0.38 | 78.88±0.41 | 16 | 44.84±0.31 | 43.78±0.35 |
| 32 | 87.91±0.36 | 86.91±0.37 | 32 | 47.55±0.29 | 46.89±0.31 |
| 64 | 91.26±0.25 | 90.64±0.35 | 64 | 49.19±0.25 | 48.09±0.30 |
| 128 | 92.79±0.20 | 91.21±0.28 | 128 | **51.22±0.20** | 49.95±0.29 |
| 256 | **92.89±0.25** | 92.07±0.27 | 256 | 50.77±0.19 | 49.62±0.24 |
| 512 | 92.33±0.23 | 91.94±0.25 | 512 | 50.65±0.22 | 49.34±0.24 |

Table 6: Performance comparison of different GCL integration strategies with various kernels.

| Method | Cosine | RBF | Polynomial | Sigmoid | Laplacian |
|---|---|---|---|---|---|
| Baseline | 65.95±0.25 | 65.95±0.25 | 65.95±0.25 | 65.95±0.25 | 65.95±0.25 |
| Early GCL | **67.53±0.21** | 66.66±0.28 | 66.59±0.29 | 66.63±0.30 | 66.42±0.28 |
| Mid GCL | **67.91±0.19** | 67.04±0.24 | 66.97±0.24 | 67.01±0.36 | 66.80±0.29 |
| Late GCL | **68.32±0.20** | 67.45±0.23 | 67.38±0.29 | 67.42±0.29 | 67.21±0.31 |
| Early + Mid | **67.62±0.23** | 66.75±0.24 | 66.68±0.21 | 66.72±0.27 | 66.51±0.29 |
| Mid + Late | **68.26±0.18** | 67.39±0.23 | 67.32±0.23 | 67.36±0.31 | 67.15±0.27 |
| Early + Late | **67.21±0.24** | 66.34±0.21 | 66.27±0.28 | 66.31±0.28 | 66.10±0.26 |
| Full GCL | **68.25±0.21** | 67.38±0.22 | 67.31±0.25 | 67.35±0.27 | 67.14±0.25 |

While kernel methods (*e.g.*, RBF, polynomial) offer expressive similarity functions, our GCLs operate on features already shaped by deep non-linear transformations. Thus, we prioritize simplicity and generality: cosine is efficient, *hyperparameter-free*, and preserves our goal of making GCLs a lightweight, plug-and-play regularizer.

We tested multiple kernels on MobileNet with CIFAR-100 and found cosine consistently outperforms others, further supporting our design choice.

## H.4 GCR Reduces Inter-Class Noise

We now provide quantitative evidence supporting our claim that GCR reduces inter-class noise, beyond prior visualizations.

**Clustering and separability metrics.** We use the Silhouette score (higher values indicate tighter intra-class clustering and clearer separation from other classes) and the Separability ratio (measuring inter-class *vs*. intra-class distance; higher is better).

Results on CIFAR-10 across ten models show that GCR consistently improves feature separability and cohesion. For example, on ResNet-34, Silhouette($\uparrow$) increases from 0.60 to 0.73, and SepRatio($\uparrow$) from 3.10 to 4.41, confirming clearer class boundaries.

**Confusion matrices.** Confusion matrices indicate reduced inter-class confusion. For example, "cat-dog" confusion decreases from 0.09 to 0.07, and diagonal accuracies improve across several classes (*e.g.*, "auto": $0.96 \rightarrow 0.97$).

## H.5 GCR on Earlier Layers

**Motivation for early-layer regularization.** While later layers are more semantic, we find that GCR sometimes works best in earlier layers, especially on Tiny ImageNet and low-capacity models. This effect arises due to several factors. Early features often exhibit higher noise and misalignment, which GCR's adaptive weighting naturally targets. Regularization at these layers helps prune spurious low-level features, setting the network on a better optimization path. Moreover, prediction-driven self-prompting allows final-layer structure to refine earlier layers via backpropagated relational

Table 7: Quantitative metrics showing improvements in feature clustering and confidence with GCR.

| Model | Silhouette | | SepRatio | | Confidence | |
|---|---|---|---|---|---|---|
| | Baseline | **+ GCR** | Baseline | **+ GCR** | Baseline | **+ GCR** |
| DenseNet-121 | 0.4724 | 0.5001 | 2.2278 | 2.3325 | 0.9746 | 0.9805 |
| ShuffleNet | 0.2806 | 0.4083 | 1.7692 | 2.0472 | 0.9568 | 0.9619 |
| SqueezeNet | -0.1245 | -0.0825 | 1.0008 | 1.0494 | 0.9603 | 0.9660 |
| ResNet-34 | 0.6032 | 0.7314 | 3.1015 | 4.4144 | 0.9801 | 0.9870 |
| ResNet-50 | 0.5314 | 0.6186 | 2.5480 | 3.2294 | 0.9789 | 0.9835 |
| ResNet-101 | 0.5641 | 0.6069 | 2.7705 | 3.0793 | 0.9803 | 0.9859 |
| ResNeXt-50 | 0.5298 | 0.5604 | 2.6323 | 2.7941 | 0.9788 | 0.9814 |
| ResNeXt-101 | 0.5668 | 0.6951 | 2.8387 | 3.8703 | 0.9811 | 0.9880 |
| GoogLeNet | -0.0255 | -0.0055 | 1.1982 | 1.2065 | 0.9720 | 0.9749 |
| Avg | 0.3776 | **0.4481** | 2.2319 | **2.6692** | 0.9737 | **0.9788** |

Table 8: CIFAR-10 confusion matrix for the baseline model.

| | plane | auto | bird | cat | deer | dog | frog | horse | ship | truck |
|---|---|---|---|---|---|---|---|---|---|---|
| plane | **0.93** | 0.01 | 0.02 | 0.01 | | | | | 0.03 | 0.01 |
| auto | 0.01 | **0.96** | | | | | | | | 0.03 |
| bird | 0.02 | | **0.88** | 0.03 | 0.02 | 0.01 | 0.02 | 0.01 | | |
| cat | 0.01 | | 0.02 | **0.83** | 0.02 | 0.08 | | | 0.01 | 0.01 |
| deer | 0.01 | | 0.01 | 0.02 | **0.93** | 0.01 | 0.01 | 0.01 | | |
| dog | | | 0.01 | 0.09 | 0.02 | **0.86** | | 0.01 | | |
| frog | 0.01 | | 0.02 | 0.01 | 0.01 | | **0.95** | | | |
| horse | | | 0.01 | 0.02 | 0.01 | 0.02 | | **0.94** | | |
| ship | 0.02 | 0.01 | | | | | | | **0.95** | 0.01 |
| truck | 0.01 | 0.04 | | | | | | 0.01 | | **0.94** |

Table 9: CIFAR-10 confusion matrix for the model **with GCR**.

| | plane | auto | bird | cat | deer | dog | frog | horse | ship | truck |
|---|---|---|---|---|---|---|---|---|---|---|
| plane | **0.94** | | 0.02 | 0.01 | | | | | 0.01 | 0.01 |
| auto | | **0.97** | | | | | | | | 0.02 |
| bird | 0.01 | | **0.89** | 0.03 | 0.02 | 0.02 | 0.01 | 0.01 | | |
| cat | 0.01 | | 0.01 | **0.85** | 0.01 | 0.08 | | 0.01 | 0.01 | 0.01 |
| deer | 0.01 | | 0.01 | 0.02 | **0.94** | 0.01 | 0.01 | 0.01 | | |
| dog | | | 0.01 | 0.07 | 0.01 | **0.88** | | 0.01 | | |
| frog | 0.01 | | 0.02 | 0.01 | 0.01 | | **0.95** | | | |
| horse | | | 0.01 | 0.02 | 0.01 | 0.01 | | **0.94** | | |
| ship | 0.02 | | | | | | | | **0.96** | 0.01 |
| truck | 0.01 | 0.02 | | 0.01 | | | | | 0.01 | **0.95** |

Table 10: Impact of early *vs*. late GCR on feature robustness in ShuffleNet. Pre-freeze and post-freeze top-1 accuracy are reported, along with the performance drop.

| Model | Pre-freeze | Post-freeze | Performance Drop |
|---|---|---|---|
| Model A (Early-GCR) | 66.8% | 66.1% | **0.7%** |
| Model B (Late-GCR) | 66.4% | 65.1% | **1.2%** |

Table 11: Performance comparison on CIFAR-10, CIFAR-100, and ImageNet-1K.

| Method | CIFAR-10 | CIFAR-100 | ImageNet-1K |
|---|---|---|---|
| **ResNet-18** | | | |
| CNN2GNN [101] | 95.51±0.42 | 74.80±0.81 | 60.12±1.02 |
| CNN2GNN **+ GCR** | 95.87±0.31 | 76.23±0.38 | 62.47±0.47 |
| CNN2Transformer [101] | 95.79±0.24 | 77.39±0.20 | 71.12±0.35 |
| CNN2Transformer **+ GCR** | **95.96±0.35** | **78.23±0.30** | **72.33±0.31** |
| **ResNet-34** | | | |
| CNN2GNN [101] | 96.39±0.41 | 77.87±0.91 | 61.02±0.77 |
| CNN2GNN **+ GCR** | 96.67±0.36 | 78.14±0.54 | 62.88±0.46 |
| CNN2Transformer [101] | 96.73±0.37 | 80.10±0.45 | 75.42±0.15 |
| CNN2Transformer **+ GCR** | **96.97±0.36** | **81.27±0.29** | **76.67±0.26** |

signals. Shallow models benefit more from early guidance because they downsample aggressively and lack strong inductive biases.

**Experiments and analysis.** To test whether GCR's impact correlates with a layer's semantic misalignment, we trained a CeiT model on Tiny ImageNet without GCR and measured the baseline discrepancy for each block $l$: $\delta(l) = \|\mathbf{F}^{(l)} - \mathbf{P}\|_F^2$. We then applied GCR to individual blocks and recorded the top-1 accuracy gain $\Delta\text{Acc}(l)$. Results show that early layers, bridging low-level features to class concepts, exhibit the highest misalignment and largest gains: (i) Block 1: $\delta_1 = 0.45$, gain +1.2%, (ii) Block 2: $\delta_2 = 0.30$, gain +0.9%. A strong Pearson correlation of 0.62 between $\{\delta(l)\}$ and $\{\Delta\text{Acc}(l)\}$ quantitatively confirms that GCR is more effective where feature-prediction misalignment is greater.

Next, we evaluated whether early GCR creates more robust features that benefit later layers, producing a "feature cleaning" effect. We trained two ShuffleNet models (5 blocks each) and then froze the regularized blocks to assess their standalone quality: (i) Model A (Early-GCR): GCLs on Blocks 1-2, frozen after 100 epochs, then fine-tuned remaining blocks. (ii) Model B (Late-GCR): GCLs on Blocks 4-5, frozen and fine-tuned similarly. Model A's smaller drop indicates that early GCR features are more robust and semantically coherent, reducing dependence on later-stage regularization (Table 10). In contrast, Model B's larger drop suggests that late-GCR performance relies heavily on continued regularization, with earlier features remaining entangled.

These results show that GCR scales well and confirm our core insight: aligning feature geometry with prediction semantics strengthens generalization.

## H.6 Results on CIFAR-10, CIFAR-100, and ImageNet-1K

**Distinction from graph-based methods.** Although graph-based methods have been extensively studied, our proposed GCR departs from this line of work by introducing a fundamentally different mechanism. Existing approaches often depend on static external graphs or rely on iterative message passing as in GNNs. By contrast, GCR uses a novel self-prompted regularization strategy, where the model's own predictions dynamically construct a class-aware graph that supervises its intermediate feature representations.

This design brings several innovations. First, instead of relying on pre-defined structures or memory banks, GCR builds graphs on the fly from the model's softmax outputs within each batch, making the supervision inherently adaptive to the evolving state of the model. Second, GCR operates as a purely parameter-free regularizer rather than as a feature transformer. It introduces no additional learnable parameters, remains agnostic to architecture, and can be seamlessly integrated into diverse models with minimal overhead. Finally, GCR enforces a unique cross-space alignment: similarity graphs in the feature space are aligned with semantic graphs in the prediction space, coupling representation learning with prediction dynamics. In this way, GCR turns predictions into structured supervisory signals that guide feature learning, offering a lightweight yet powerful alternative to conventional graph-based classification methods.

Table 12: Results on ImageNet-1K.

| Model | ViG-Ti [37] | ViG-Ti + **GCR** | ViG-S [37] | ViG-S + **GCR** | ViG-B [37] | ViG-B + **GCR** |
|---|---|---|---|---|---|---|
| Accuracy | 73.9 | **74.9** | 80.4 | **81.7** | 82.3 | **84.0** |

Table 13: Comparison of GCR with one-hot labels *vs.* soft predictions on CIFAR-100 (average over 10 runs).

| | MobileNet | ShuffleNet | SqueezeNet |
|---|---|---|---|
| Baseline | 65.95±0.25 | 70.11±0.30 | 69.43±0.27 |
| GCR (one-hot labels) | 67.35±0.24 | 71.28±0.26 | 70.49±0.25 |
| GCR (soft predictions, **ours**) | **68.32**±0.20 | **71.96**±0.27 | **71.03**±0.24 |

We provide direct comparisons between our GCR-augmented models and recent graph-based classification methods across CIFAR-10, CIFAR-100, and ImageNet-1K (Table 11 and Table 12).

The results clearly show the consistent and complementary benefits of GCR when integrated into graph-based backbones. They highlight key insights: (i) Consistent across architectures: GCR improves performance across diverse backbones (CNN2GNN [101], CNN2Transformer [101], ViG [37]), demonstrating broad applicability. (ii) Complementary to existing methods: Its self-regularization mechanism enhances CNN2GNN and CNN2Transformer, indicating orthogonality to existing graph-based pipelines. (iii) Scalable to large-scale tasks: Notable gains on ImageNet-1K (*e.g.*, +1.7% with ViG-B) show GCR's effectiveness in complex, real-world settings. (iv) Lightweight and model-agnostic: GCR introduces no extra parameters and integrates easily into existing architectures.

Together, these findings reinforce GCR's novelty and practical value as a flexible, self-prompted regularization framework applicable across scales and architectures.

### H.7 Feature Similarity and the Information Bottleneck

**Soft predictions *vs.* one-hot labels.** For a pair of features $\mathbf{f}_i$ and $\mathbf{f}_j$ (dropping ReLU for brevity), we encourage $\cos(\mathbf{f}_i, \mathbf{f}_j) \approx \cos(\mathbf{s}_i, \mathbf{s}_j)$, where $\mathbf{s}_i$ and $\mathbf{s}_j$ are soft scores, not one-hot labels. These soft scores allow intra-class variations to persist while emphasizing task-relevant semantics. By imposing this constraint, we create an information bottleneck: the smaller the angle between $\mathbf{f}_i$ and $\mathbf{f}_j$, the stronger the bottleneck. The network is thus forced to encode the most important variations while discarding nuisance factors. For example, in the Chihuahua *vs.* Great Dane scenario, features such as eyes, paws, nose, nails, and teeth are emphasized and encoded consistently across all dog species, whereas leg size or body shape variations are considered less relevant. This ensures that salient features are accurately captured and distinguishable across classes. If $\mathbf{x}_i$ and $\mathbf{x}_j$ belong to different classes (*e.g.*, dog *vs.* wolf), $\cos(\mathbf{s}_i, \mathbf{s}_j)$ will be lower but typically greater than zero, allowing some semantic similarity while relaxing the information bottleneck.

Using $\cos(\mathbf{f}_i, \mathbf{f}_j) \approx \cos(\mathbf{y}_i, \mathbf{y}_j)$ with one-hot labels would collapse intra-class variations (see Table 13). Instead, using soft predictions $\cos(\mathbf{f}_i, \mathbf{f}_j) \approx \cos(\mathbf{s}_i, \mathbf{s}_j)$ allows sufficient variation to be encoded.

**Prediction similarity and diverse substructures.** Even in classes with diverse substructures, prediction similarity mandates feature similarity: (i) In CNNs, as one moves toward the classifier, layers become increasingly shift- and permutation-invariant. Thus, deeper layers filter nuisance factors irrelevant to the classification task. (ii) The information bottleneck ensures the network focuses on salient, task-relevant features and discards uninformative variations. (iii) The bottleneck strength is controlled by $\lambda$ in Eq. (7). Moderate values ($\lambda = 1$) perform best (see Table 14).

Table 14: Impact of bottleneck strength $\lambda$ on CIFAR-100 (ResNet-34).

| $\lambda$ | 0 | 0.1 | 0.3 | 0.5 | 0.7 | 1 | 3 | 5 | 7 | 10 |
|---|---|---|---|---|---|---|---|---|---|---|
| Top-1 Acc (%) | 76.76 | 76.80 | 76.87 | 77.32 | 77.61 | **78.38** | 76.74 | 76.02 | 75.90 | 75.37 |

Table 15: Effect of varying $\tau$ on CIFAR-100 with ResNet-34.

| $\tau$ | 0 | 1e−5 | 1e−4 | 1e−3 | 1e−2 |
|---|---|---|---|---|---|
| Accuracy | 78.38 | 78.32 | 78.27 | 78.13 | 76.04 |

These results confirm that GCR preserves intra-class variations while enforcing focus on the most important semantic features, improving robustness and accuracy without collapsing representations.

## H.8 Lipschitz Continuity and Our Mechanism

$L$**-Lipschitz continuous networks.** It is well established that enforcing $L$-Lipschitz continuity in a network, such as a discriminator, helps regularize its layers and prevents overfitting. For example, Lipschitz Generative Adversarial Nets (LGANs) [128], along with many subsequent works on classification and fine-tuning under $L$-Lipschitzness.

This condition can be imposed on feature vectors $\phi(\cdot)$ of a chosen layer $l$ by design, leading to

$$\|\phi_l(\mathbf{x}) - \phi_l(\mathbf{x}')\|_2 \leq L\|\mathbf{x} - \mathbf{x}'\|_2, \quad \forall\, \mathbf{x}, \mathbf{x}' \in \mathcal{D}, \tag{23}$$

where $L$ is the Lipschitz constant. Intuitively, if $L$ is small, then a small input change $\mathbf{x} \to \mathbf{x}'$ produces a small change in the features $\phi_l(\mathbf{x}) \to \phi_l(\mathbf{x}')$, while a large change in input produces a proportionally large change in the features. This stabilizes the network response.

A practical way to encourage such behavior is through an auxiliary regularization loss added to the main task loss:

$$\mathbb{E}_{\mathbf{x},\mathbf{x}'}\big(\|\phi_l(\mathbf{x}) - \phi_l(\mathbf{x}')\|_2 - L\|\mathbf{x} - \mathbf{x}'\|_2\big)^2, \tag{24}$$

which promotes the approximation

$$\|\phi_l(\mathbf{x}) - \phi_l(\mathbf{x}')\|_2 \approx L\|\mathbf{x} - \mathbf{x}'\|_2. \tag{25}$$

For simplicity, we omit ReLU and cosine similarity in what follows and use the $\ell_2$ distance, though cosine distance can be incorporated without difficulty. We promote alignment of the form

$$\mathbb{E}_{\mathbf{x},\mathbf{x}'}\big(\|\phi_l(\mathbf{x}) - \phi_l(\mathbf{x}')\|_2 - \beta\|s(\mathbf{x}) - s(\mathbf{x}')\|_2\big)^2, \tag{26}$$

where $s(\cdot)$ denotes the softmax score. This yields an information bottleneck of the form

$$\|\phi_l(\mathbf{x}) - \phi_l(\mathbf{x}')\|_2 \approx \beta\|s(\mathbf{x}) - s(\mathbf{x}')\|_2. \tag{27}$$

Combining the Lipschitz condition with our bottleneck mechanism, we obtain

$$L\|\mathbf{x} - \mathbf{x}'\|_2 \approx \|\phi_l(\mathbf{x}) - \phi_l(\mathbf{x}')\|_2 \approx \beta\|s(\mathbf{x}) - s(\mathbf{x}')\|_2, \tag{28}$$

which implies

$$\|s(\mathbf{x}) - s(\mathbf{x}')\|_2 \approx \frac{L}{\beta}\|\mathbf{x} - \mathbf{x}'\|_2. \tag{29}$$

This demonstrates that our model does not collapse intermediate features into identical representations. Instead, small input variations $\mathbf{x} \to \mathbf{x}'$ yield small variations in both features and softmax scores, while large input differences induce proportionally large differences in the outputs.

The strength of this relation depends on the network's local $L$-Lipschitzness (which may be fixed or vary in non-Lipschitz networks) and the scaling factor $\beta$, which is implicitly controlled by our parameter $\lambda$. Therefore, our information bottleneck mechanism ensures stable yet discriminative feature variation.

**Introducing $\tau$: collapse prevention by design.** We enforce $\angle(\phi_l(\mathbf{x}_i), \phi_l(\mathbf{x}_j)) > \tau$ by a simple soft penalty $\sum_l \mathbf{1}(y_i = y_j) \cdot \mathbf{1}(\angle(\phi_l(\mathbf{x}_i), \phi_l(\mathbf{x}_j)) < \tau) \cdot (\angle(\phi_l(\mathbf{x}_i), \phi_l(\mathbf{x}_j)) - \tau)^2$, where $\mathbf{1}(\cdot)$ is the indicator function (*e.g.*, do samples $i$ and $j$ share the same label?) and $\angle(\mathbf{x}, \mathbf{y}) = \cos^{-1}\big(\frac{\mathbf{x}^\top \mathbf{y}}{\|\mathbf{x}\|_2 \cdot \|\mathbf{y}\|_2}\big)$.

The results for CIFAR-100 using ResNet-34 are in Table 15. This result directly offers both anti-collapse control and evidence that the best results are attained for $\tau = 0$ (not pushing angles of the same class apart). Always maintaining a non-zero angle, *e.g.*, $\tau = 1e-2$, is worse. We therefore conclude that our network does not suffer from feature dimensional collapse. However, if such collapse did occur, it could be prevented by a simple penalty, as demonstrated here.

Table 16: Results on CIFAR-100 (100 classes) with MobileNet. Accuracy and intra-/inter-class variance across six layers under different $\tau$.

| $\tau$ (deg) | Acc | 1st | | 2nd | | 3rd | | 4th | | 5th | | 6th | |
|---|---|---|---|---|---|---|---|---|---|---|---|---|---|
| | | intra | inter | intra | inter | intra | inter | intra | inter | intra | inter | intra | inter |
| baseline | 65.95 | 38.71 | 13.81 | 22.30 | 7.56 | 12.00 | 5.27 | 5.10 | 4.24 | 17.71 | 19.49 | 8.64 | 9.69 |
| **ours** (0.00) | 68.14 | 31.21 | 12.02 | 18.18 | 7.61 | 9.98 | 4.28 | 3.94 | 3.25 | 16.82 | 18.69 | 8.30 | 9.32 |
| 0.08 | **68.21** | 31.25 | 12.56 | 18.28 | 6.54 | 9.92 | 4.46 | 4.10 | 3.28 | 18.77 | 19.52 | 9.23 | 9.74 |
| 0.26 | 67.68 | 36.28 | 13.11 | 20.93 | 7.15 | 10.94 | 4.92 | 4.58 | 3.81 | 17.72 | 19.08 | 8.72 | 9.50 |
| 0.81 | 67.37 | 35.39 | 13.72 | 20.44 | 6.92 | 11.02 | 4.89 | 4.70 | 3.89 | 17.74 | 18.87 | 8.70 | 9.41 |
| 2.56 | 67.34 | 35.40 | 13.40 | 20.61 | 7.32 | 10.83 | 4.84 | 4.56 | 3.74 | 17.92 | 19.08 | 8.78 | 9.50 |
| 8.11 | 67.39 | 39.04 | 14.86 | 23.42 | 8.04 | 12.00 | 5.24 | 5.16 | 4.38 | 16.77 | 18.58 | 8.28 | 9.27 |

Table 17: Results on CIFAR-100 (20 super-classes) with MobileNet. Accuracy and intra-/inter-class variance across six layers under different $\tau$.

| $\tau$ (deg) | Acc | 1st | | 2nd | | 3rd | | 4th | | 5th | | 6th | |
|---|---|---|---|---|---|---|---|---|---|---|---|---|---|
| | | intra | inter | intra | inter | intra | inter | intra | inter | intra | inter | intra | inter |
| baseline | 77.46 | 30.30 | 8.86 | 17.94 | 4.30 | 9.25 | 2.97 | 3.01 | 2.46 | 9.09 | 12.27 | 4.45 | 6.12 |
| **ours** (0.00) | 77.54 | 30.36 | 8.65 | 17.95 | 4.43 | 9.34 | 2.99 | 3.22 | 2.56 | 9.69 | 12.57 | 4.73 | 6.27 |
| 0.08 | **77.72** | 31.59 | 9.22 | 18.24 | 4.42 | 9.57 | 3.08 | 3.33 | 2.61 | 9.91 | 12.56 | 4.87 | 6.27 |
| 0.26 | 77.36 | 32.04 | 9.80 | 18.67 | 4.58 | 9.57 | 3.07 | 3.28 | 2.70 | 9.85 | 12.69 | 4.83 | 6.33 |
| 0.81 | 77.69 | 31.36 | 9.59 | 18.17 | 4.38 | 9.50 | 3.04 | 3.15 | 2.60 | 9.70 | 12.63 | 4.77 | 6.30 |
| 2.56 | 77.31 | 31.69 | 9.38 | 18.56 | 4.36 | 9.39 | 3.03 | 3.30 | 2.68 | 9.94 | 12.55 | 4.88 | 6.26 |
| 8.11 | 77.60 | 32.39 | 8.66 | 18.34 | 4.47 | 9.35 | 3.01 | 3.26 | 2.65 | 9.83 | 12.48 | 4.81 | 6.22 |

## H.9  Preventing Dimensional Collapse by Design

**Anti-collapse mechanism.** We introduce a penalty to directly prevent by-design a possibility of dimensional collapse. To this end, we conduct an experiment on CIFAR-100, grouped into 20 super-classes, *e.g.*, vehicles (bicycle, bus, motorcycle, pickup truck, train), insects (bee, beetle, butterfly, caterpillar, cockroach), flowers (orchids, poppies, roses, sunflowers, tulips), *etc*.

To this end, we force the within-class angle $\angle$ between pairs of features to be at least $\tau$. Specifically, we add to our GCR loss the following soft penalty: $\beta \frac{1}{\zeta} \sum_l \mathbf{1}(y_i = y_j) \cdot \mathbf{1}(\angle(\phi_l(\mathbf{x}_i), \phi_l(\mathbf{x}_j)) < \tau) \cdot$ $(\angle(\phi_l(\mathbf{x}_i), \phi_l(\mathbf{x}_j)) - \tau)^2$, which ensures that $\angle(\phi_l(\mathbf{x}_i), \phi_l(\mathbf{x}_j)) > \tau$ within each class. Here, $\zeta$ is a normalization factor accounting for counts of same-class elements. For cosine similarity, we also apply ReLU in the above equations to prevent negative angles.

**Setup.** We use pre-trained baselines for comparisons. For our model, we train GCR-augmented models on CIFAR-100 for: (i) the 100-class task, and (ii) the 20 super-class task. The GCR-augmented model is equipped with a by-design collapse prevention mechanism. We choose $\beta = 0.3$ (generally optimal in our experiments). Subsequently, we vary the minimum angle $\tau$ to enforce a lower bound on intra-class variance. We report both intra- and inter-class variance across all six layers of MobileNet (used for fast experiments), considering both the original 100 classes and the 20 super-classes. Additionally, we include a baseline comparison without applying our GCR framework.

**Findings.** The results on both the 100-class and 20-superclass tasks show that GCR improves accuracy while maintaining meaningful feature diversity. While enforcing a minimum within-class variance with $\tau = 0.08$ helps slightly, the variance for $\tau = 0.00$ (ours) never collapsed in our experiments and remains close to the best case $\tau = 0.08$. Importantly, setting larger $\tau$ values, which enforce large intra-class variance, degrades accuracy. Maintaining certain within-class feature variance by a soft penalty makes sense. At the same time, the dimensional collapse does not happen in our model even without that penalty.

# I  Limitations and Future Works

## I.1  Failure Modes and Marginal Gains

**Highly noisy data.** GCR aligns feature graphs with a masked prediction graph, using it as a semantic reference. The quality of this reference is critical. Our method assumes reliable ground-truth labels

for the mask $\mathbf{M}$. If the training data contains spurious correlations or mislabeled examples, the alignment may reinforce these errors. Under high label noise, the prediction graph $\mathbf{P}$ is based on a flawed mask, producing a corrupted supervisory signal that misguides feature representations toward incorrect semantics. This failure mode can lead to marginal gains or even performance degradation in our supervised framework.

**Highly class-imbalanced data.** Since GCR builds relational graphs at the batch level, the global context is limited to within-batch relationships. In highly imbalanced datasets, batches may contain few or no minority-class samples, resulting in sparse or uninformative prediction graphs for those classes. Consequently, the alignment loss is dominated by majority classes, potentially harming minority-class representations and leading to marginal overall gains.

**Other scenarios leading to marginal gains.** Beyond noisy and imbalanced data, GCR may yield marginal improvements in the following cases: (i) Simple datasets with high baseline performance: GCR targets noisy inter-class similarities and semantic structure. On simpler datasets with strong baseline models and well-separated features, there is less room for improvement. Our results confirm this, with larger gains on complex datasets such as CIFAR-100 and Tiny ImageNet compared to CIFAR-10, where baseline accuracy was already high. (ii) Extremely small batch sizes: Relational graphs rely on sufficient pairwise relationships. Very small batches provide limited data context, reducing graph stability and weakening regularization.

## I.2 Applicability to Unsupervised and Self-Supervised Learning

Our current GCR formulation relies on ground-truth labels for intra-class masking of the prediction graph. This design was intentional to first establish and validate GCR's core effectiveness in a fully supervised setting, where we demonstrated consistent, significant gains across diverse architectures and datasets.

However, this reliance is not a fundamental limitation of GCR but rather a characteristic of the current implementation. The core mechanism, aligning feature graph geometry with prediction graph semantics, is flexible. The mask $\mathbf{M}$ can be generated without ground-truth labels, enabling extensions to unsupervised or self-supervised learning. Specifically, we propose two potential adaptations: (i) Pseudo-labeling: In semi/self-supervised settings, high-confidence model predictions can generate pseudo-labels to construct $\mathbf{M}$, masking pairs sharing the same pseudo-label. (ii) Unsupervised clustering: Feature representations can be clustered dynamically (*e.g.*, via k-means), with $\mathbf{M}$ derived from cluster assignments to enforce consistency among similar samples.

While this work focuses on supervised learning to establish GCR's principle, the framework is modular. Replacing the ground-truth mask with pseudo-labels or clustering-based masks readily extends GCR to unsupervised domains. This important direction highlights GCR's broader potential and flexibility.

## I.3 Future Works

While GCR introduces a novel and effective mechanism for enhancing semantic structure in deep representations, it also presents several limitations that open up promising avenues for future exploration.

First, GCR currently relies on batch-level relational graphs constructed from intermediate features and softmax predictions. This inherently restricts the global context to within-batch relationships, which may limit its effectiveness in settings with small batch sizes or highly imbalanced class distributions. Exploring memory-augmented or streaming graph variants that accumulate semantic structure across batches could improve scalability and robustness.

Second, although the adaptive weighting mechanism allows the model to prioritize layers with high misalignment, it is currently driven solely by Frobenius distance. More expressive structural metrics, such as spectral divergence, Earth Mover's distance, or alignment of Laplacian eigenvectors, could offer deeper insights into graph discrepancy and guide more fine-grained supervision.

Third, GCR assumes access to ground-truth labels for masking during training, limiting its current formulation to fully supervised learning. Extending GCR to semi-supervised, self-supervised, or weakly

supervised regimes by deriving prediction graphs from confident pseudo-labels or unsupervised clustering remains an exciting direction.

Fourth, while GCR is model-agnostic and does not alter the architecture, it introduces additional computation from graph construction and alignment. Although lightweight and parallelizable, this cost may still pose challenges in latency-sensitive applications. Investigating more efficient or approximate graph alignment schemes could alleviate this concern.

Finally, our method focuses on classification tasks. Adapting GCR to other modalities and tasks, such as segmentation, detection, or multimodal fusion, requires further study. In particular, integrating GCR with token-level or region-level predictions in transformer models may offer novel insights into semantic alignment beyond image classification.

In future work, we aim to address these limitations by developing more scalable graph construction techniques, expanding the framework to broader learning paradigms, and refining the theoretical understanding of structural supervision in deep networks.

## J    Broader impacts

GCR introduces a lightweight and architecture-agnostic framework for improving semantic consistency in deep learning models. By aligning intermediate features with structured prediction graphs, GCR encourages networks to learn more coherent, interpretable, and generalizable representations. This has the potential to improve model reliability in critical applications such as medical diagnosis, autonomous driving, and scientific data analysis, where semantic structure and interpretability are crucial.

From an ethical standpoint, GCR does not introduce any explicit biases beyond those already present in the training data. However, like all supervision-driven methods, its effectiveness depends on the quality of labels. If training data contains spurious correlations or mislabeled examples, the alignment process may reinforce rather than correct those artifacts. Future work should investigate ways to make GCR more robust to noisy or biased supervision signals.

GCR is computationally efficient and compatible with existing training protocols, which facilitates deployment in low-resource settings or on edge devices. However, care should be taken to assess environmental costs when scaling to very large models or datasets. In addition, the reliance on pairwise relationships may raise privacy concerns in sensitive domains where sample-wise relationships reveal protected attributes. Applying GCR to privacy-preserving or federated learning settings is a promising direction for ensuring responsible AI development.

