# OpenReview forum: "Graph Your Own Prompt"
_NeurIPS.cc/2025/Conference — NeurIPS 2025 poster_

### Official Review · Reviewer_bJ5m · 2025-06-20

**Clarity:** 4
**Significance:** 2
**Originality:** 2
**Rating:** 5
**Confidence:** 4

**Summary:**

This paper introduce a regularisation strategy (named GCR) which encourages the similarity between the features for elements in a batch at a certain layer to reflect the similarity between the output representations (i.e., softmax predictions). This is achieved by constructing a graph representing the similarity between the representations for the elements in a batch at a given layer (using the cosine similarity). The graph for the last layer considers only the relationships between elements with the same class and is used as a "target". The proposed regularisation loss is then computed by summing the losses between the similarity matrices at every layer and the target one.
The proposed regularisation only guides the training procedure and  does not affect the inference procedure.

Some theoretical results are further provided to show how GCR acts as a "good" regularization by reducing the complexity of the model and aligning the Laplacians of the representations of the layers with the target one.

The authors test their method on classic image classification datasets (CIFAR, Tiny-Imagenet) on a large number of architectures including several vision transformers and convolutional network architectures. Both quantitative results (comparing the same architecture with an without GCR) and qualitative results (showing examples of relationships between representations extracted from models learned with and without GCR) are shown.

The idea behind the proposed method is based on the knowledge that earlier layers learn more basic features, while later layers learn more "high-level" and discriminative features. By then pushing the earlier representations to "encode" the same relationships extracted by the later layers, the proposed method should induce the learning of more discriminative patterns, which in turn should help generalization.

**Questions:**

- Could the authors please add information about the variance across runs of the results? I am happy to raise my score if this is added and the results remain positive
- Cosine similarity is used to construct the similarity graphs. However, in high dimensions, cosine similarity may not be ideal as many vectors tend to become (nearly) orthogonal to each other. Furthermore it does not take magnitude into account, and it only measures linear relationships. Have the authors tried using any other way of measuring similarity? For example the use of kernels could allow for the introduction of non-linear relationships. I saw there is some limited discussion in the Appendix, but I think it's important to go more in depth on this
- The abstract mentions that the proposed method reduces inter-class noise. The qualitative results do seem to hint at this, but are there any quantitative results to support this? For example it could interesting to show the confusion matrix of the models trained with and without GCR.
- It is curious to me that in certain cases the best performance is achieved when GCR is applied to the earlier layers. I would expect the earlier layers to only have to focus on "basic" features, which shouldn't necessarily be aligned to the output features (as the distinction into different classes then happens in later layers which combine the basic features from earlier layers). Do you have any comment on why in some cases GCR on earlier layer works better than on later ones?

**Ethical Concerns:**

["NO or VERY MINOR ethics concerns only"]

**Final Justification:**

The authors have provided significant new results addressing all my concerns. I thus have raised my score to Accept

**Limitations:**

Yes limitations are discussed in the appendix. As mentioned in my question I think aspect related to the choice of metric should be expanded in my opinion

**Quality:**

3

**Strengths And Weaknesses:**

Strengths:
- The paper is well written and easy to understand
- The qualitative results nicely present the possible benefits of the method
- The theoretical results are a nice addition to provide some possible motivations for GCR, although they do not bring any practical insight

Weaknesses:
- The quantitative results, contrary to the qualitative ones, show very minor performance improvements. The largest ones are only a few percentage points, while most of them are even smaller.
- The authors mention that results are averaged over 3 runs, but the results do not show the variance over the multiple runs. Given the small differences in the results (as expressed in the previous point), showing the variance (and possibly increasing the number of runs to 10) is fundamental to be able to judge the effectiveness of the proposed method
- The abstract mentions that the proposed method reduces inter-class noise, but no quantitative results are shown (see question below for suggestions)
- Only cosine similarity is used to construct the similarity metric (see question below for more details and suggestion)

---

> ### Author Rebuttal · Authors · 2025-07-30
>
> We thank the reviewer for the feedback and appreciation of the paper's clarity and contributions.
>
> Below, we address all raised points.
>
> ### **1. Variance across 10 runs**
>
> As requested, we report variance across 10 runs. Due to rebuttal space limits, we present Tiny ImageNet results here; **all tables in the revised paper include means and standard deviations over 10 runs**.
>
> Tiny ImageNet:
>
> ||ViT/32|ViT/16|CeiT|MViT_XXS|MViT_XS|MViT|Swin|MNet|R18SD|SER18|R34|Mean|
> |-|-|-|-|-|-|-|-|-|-|-|-|-|
> |Baseline|37.79$\pm$0.35|40.05$\pm$0.33|49.95$\pm$0.29|49.28$\pm$0.29|51.58$\pm$0.27|52.68$\pm$0.27|54.27$\pm$0.25|57.81$\pm$0.25|63.49$\pm$0.26|65.65$\pm$0.24|67.51$\pm$0.25|53.64$\pm$9.62|
> |Early GCL|39.02$\pm$0.29|40.98$\pm$0.19|51.22$\pm$0.20|50.11$\pm$0.28|51.33$\pm$0.26|53.91$\pm$0.22|54.88$\pm$0.25|57.93$\pm$0.21|63.81$\pm$0.19|66.52$\pm$0.22|67.79$\pm$0.19|54.32$\pm$9.39|
> |Mid GCL|38.61$\pm$0.23|40.95$\pm$0.19|50.30$\pm$0.19|49.92$\pm$0.26|51.43$\pm$0.22|53.88$\pm$0.20|55.23$\pm$0.24|57.63$\pm$0.20|64.03$\pm$0.22|65.66$\pm$0.23|67.62$\pm$0.20|54.11$\pm$9.38|
> |Late GCL|37.98$\pm$0.28|40.35$\pm$0.25|50.82$\pm$0.20|49.77$\pm$0.21|51.99$\pm$0.23|54.10$\pm$0.19|55.47$\pm$0.21|57.87$\pm$0.23|63.79$\pm$0.19|65.85$\pm$0.25|67.61$\pm$0.19|54.18$\pm$9.56|
> |Early+Mid|39.08$\pm$0.25|41.26$\pm$0.18|50.25$\pm$0.25|49.73$\pm$0.22|51.57$\pm$0.19|53.91$\pm$0.23|54.95$\pm$0.19|57.49$\pm$0.19|64.18$\pm$0.20|65.86$\pm$0.24|67.74$\pm$0.23|54.18$\pm$9.32|
> |Mid+Late|38.44$\pm$0.18|40.52$\pm$0.28|50.09$\pm$0.25|50.55$\pm$0.18|51.48$\pm$0.21|53.90$\pm$0.20|55.62$\pm$0.23|57.65$\pm$0.21|64.29$\pm$0.17|65.95$\pm$0.19|67.58$\pm$0.21|54.19$\pm$9.52|
> |Early+Late|38.34$\pm$0.23|40.71$\pm$0.21|50.70$\pm$0.25|50.23$\pm$0.20|51.36$\pm$0.18|53.57$\pm$0.21|54.89$\pm$0.21|57.93$\pm$0.19|63.88$\pm$0.19|65.83$\pm$0.17|67.75$\pm$0.25|54.11$\pm$9.47|
> |Full GCL|38.38$\pm$0.22|40.80$\pm$0.18|49.92$\pm$0.20|50.16$\pm$0.17|51.87$\pm$0.19|54.01$\pm$0.19|54.87$\pm$0.19|57.64$\pm$0.20|64.10$\pm$0.19|66.01$\pm$0.15|67.66$\pm$0.18|54.13$\pm$9.49|
>
> Results confirm **consistent gains** with **low variance** and **statistical stability**.
>
> ### **2. Measuring similarity**
>
> We chose cosine similarity to emphasize directional alignment, which is more semantically meaningful and robust to nuisance factors (e.g., brightness) than raw magnitude. This aligns with common practice in representation learning, where angular relationships often capture class structure more effectively.
>
> While kernel methods (e.g., RBF, polynomial) offer expressive similarity functions, our GCLs operate on features already shaped by deep non-linear transformations. Thus, we prioritize simplicity and generality: cosine is efficient, **hyperparameter-free**, and preserves our goal of making GCLs a lightweight, plug-and-play regularizer.
>
> We tested multiple kernels on MobileNet with CIFAR-100 and found cosine consistently outperforms others, further supporting our design choice.
>
> ||Cosine|RBF|Polynomial|Sigmoid|Laplacian|
> |-|-|-|-|-|-|
> |Baseline|65.95$\pm$0.25|65.95$\pm$0.25|65.95$\pm$0.25|65.95$\pm$0.25|65.95$\pm$0.25|
> |Early GCL|**67.53$\pm$0.21**|66.66$\pm$0.28|66.59$\pm$0.29|66.63$\pm$0.30|66.42$\pm$0.28|
> |Mid GCL|**67.91$\pm$0.19**|67.04$\pm$0.24|66.97$\pm$0.24|67.01$\pm$0.36|66.80$\pm$0.29|
> |Late GCL|**68.32$\pm$0.20**|67.45$\pm$0.23|67.38$\pm$0.29|67.42$\pm$0.29|67.21$\pm$0.31|
> |Early+Mid|**67.62$\pm$0.23**|66.75$\pm$0.24|66.68$\pm$0.21|66.72$\pm$0.27|66.51$\pm$0.29|
> |Mid+Late|**68.26$\pm$0.18**|67.39$\pm$0.23|67.32$\pm$0.23|67.36$\pm$0.31|67.15$\pm$0.27|
> |Early+Late|**67.21$\pm$0.24**|66.34$\pm$0.21|66.27$\pm$0.28|66.31$\pm$0.28|66.10$\pm$0.26|
> |Full GCL|**68.25$\pm$0.21**|67.38$\pm$0.22|67.31$\pm$0.25|67.35$\pm$0.27|67.14$\pm$0.25|
>
> We’ve expanded Appendix H.4 to reflect this rationale.
>
> ### **3. GCR reduces inter-class noise**
>
> We now provide quantitative evidence supporting our claim that GCR reduces inter-class noise, beyond prior visualizations.
>
> Silhouette score: Higher values indicate tighter intra-class clustering and clearer separation from other classes. Separability ratio: Measures inter-class vs. intra-class distance; higher is better.
>
> - Clustering metrics on CIFAR-10 show GCR improves feature separability and cohesion, with higher silhouette and separability ratios across 10 models (e.g., ResNet34: Silhouette($\uparrow$) from 0.60 $\rightarrow$ 0.73, SepRatio ($\uparrow$) from 3.10 $\rightarrow$ 4.41), confirming clearer class boundaries.
>
> |Model|Baseline Silhouette|+GCR Silhouette|Baseline SepRatio|+GCR SepRatio|Baseline Confidence|+GCR Confidence|
> |:-|-:|-:|-:|-:|-:|-:|
> |densenet121|0.4724|0.5001|2.2278|2.3325|0.9746|0.9805|
> |shufflenet|0.2806|0.4083|1.7692|2.0472|0.9568|0.9619|
> |squeezenet|-0.1245|-0.0825|1.0008|1.0494|0.9603|0.9660|
> |resnet34|0.6032|0.7314|3.1015|4.4144|0.9801|0.9870|
> |resnet50|0.5314|0.6186|2.5480|3.2294|0.9789|0.9835|
> |resnet101 |0.5641|0.6069|2.7705|3.0793|0.9803|0.9859|
> |resnext50 |0.5298|0.5604|2.6323|2.7941|0.9788|0.9814|
> |resnext101|0.5668|0.6951|2.8387|3.8703|0.9811|0.9880|
> |googlenet |-0.0255|-0.0055|1.1982|1.2065|0.9720|0.9749|
> |Avg|0.3776|**0.4481**|2.2319|**2.6692**|0.9737|**0.9788**|
>
> - Confusion matrices show reduced inter-class confusion. For instance, 'cat–dog' confusion drops from 0.09 to 0.07, and diagonal accuracies improve across several classes (e.g., 'auto': 0.96 $\rightarrow$ 0.97).
>
> **Baseline**
> ||plane|auto|bird|cat|deer|dog|frog|horse|ship|truck|
> |:-|:-:|:-:|:-:|:-:|:-:|:-:|:-:|:--:|:-:|:-:|
> |**plane**|**0.93**|0.01|0.02|0.01|||||0.03|0.01|
> |**auto**|0.01|**0.96**||||||||0.03|
> |**bird**|0.02||**0.88**|0.03|0.02|0.01|0.02|0.01|||
> |**cat**|0.01||0.02|**0.83**|0.02|0.08|||0.01|0.01|
> |**deer**|0.01||0.01|0.02|**0.93**|0.01|0.01|0.01|||
> |**dog**|||0.01|0.09|0.02|**0.86**||0.01|||
> |**frog**|0.01||0.02|0.01|0.01||**0.95**||||
> |**horse**|||0.01|0.02|0.01|0.02||**0.94**|||
> |**ship**|0.02|0.01|||||||**0.95**|0.01|
> |**truck**|0.01|0.04|||||||0.01|**0.94**|
>
> **+GCR**
> ||plane|auto|bird|cat|deer|dog|frog|horse|ship|truck|
> |:-|:-:|:-:|:-:|:-:|:-:|:-:|:-:|:--:|:-:|:-:|
> |**plane**|**0.94**||0.02|0.01|||||0.01|0.01|
> |**auto**||**0.97**||||||||0.02|
> |**bird**|0.01||**0.89**|0.03|0.02|0.02|0.01|0.01|||
> |**cat**|0.01||0.01|**0.85**|0.01|0.08||0.01|0.01|0.01|
> |**deer**|0.01||0.01|0.02|**0.94**|0.01|0.01|0.01|||
> |**dog**|||0.01|0.07|0.01|**0.88**||0.01|||
> |**frog**|0.01||0.02|0.01|0.01||**0.95**||||
> |**horse**|||0.01|0.02|0.01|0.01||**0.94**|||
> |**ship**|0.02||||||||**0.96**|0.01|
> |**truck**|0.01|0.02||0.01|||||0.01|**0.95**|
>
> We have included these analyses in Appendix H.5.
>
> ### **4. GCR on earlier layers**
>
> **Part 1**: While later layers are more semantic, we find GCR sometimes works best early, especially on Tiny ImageNet and low-capacity models, due to several factors:
>
> - Early features **show higher noise and misalignment**, which GCR’s adaptive weighting naturally targets.
>
> - Early semantic regularization **helps prune spurious low-level features**, setting the network on a better optimization path.
>
> - Prediction-driven self-prompting lets final-layer structure refine earlier layers via **backpropagated relational signals**.
>
> Shallow models benefit more from early guidance, as they downsample aggressively and lack strong inductive biases.
>
> These points are now clarified in Sec 3.2 and Appendix H.6.
>
> **Part 2**: We present additional experiments below.
>
> - [Exp 1] GCR **corrects** the largest points of misalignment
>
> To test whether GCR’s impact correlates with a layer’s semantic misalignment, we trained a CeiT model on Tiny ImageNet without GCR and measured the baseline discrepancy $\delta(l) = ||F(l) - P||_F^2$ for each block $l$. We then applied GCR to individual blocks and recorded the top-1 accuracy gain $\Delta\text{Acc}(l)$.
>
> Results show that early layers, bridging low-level features to class concepts, exhibit the highest misalignment and largest gains:
>
> - Block 1: $\delta_1 = 0.45$, gain +1.2%
> - Block 2: $\delta_2 = 0.30$, gain +0.9%
>
> A **strong Pearson correlation** of **0.62** between $\{\delta(l)\}$ and $\{\Delta\text{Acc}(l)\}$ quantitatively confirms that GCR is more effective where feature-prediction misalignment is greater.
>
> - [Exp 2] early GCR **creates more robust foundational features**
>
> Next, we tested whether early GCR creates more robust features that benefit later layers, a "feature cleaning" effect. We trained two ShuffleNet models (5 blocks each) and then froze the regularized blocks to evaluate their standalone quality:
>
> - Model A (Early-GCR): GCLs on Blocks 1-2, frozen after 100 epochs, then fine-tuned remaining blocks.
>
> - Model B (Late-GCR): GCLs on Blocks 4-5, frozen and fine-tuned similarly.
>
> || Pre-freeze Top-1 | Post-freeze Top-1 | Performance Drop |
> |:-|:-:|:-:|:-:|
> |Model A|66.8%|66.1%|**0.7%**|
> |Model B|66.4%|65.1%|**1.2%**|
>
> Model A’s smaller drop shows early GCR features are more robust and semantically coherent, reducing reliance on later regularization. Model B’s larger drop suggests its performance depends more on continued late-stage regularization, with earlier features remaining entangled.
>
> **Part 3**: We provide evaluations on ImageNet-1K.
>
> ||iFormer-S|iFormer-B|ViT-B/16|ViG-B|
> |:-|:-:|:-:|:-:|:-:|
> |Baseline|83.4$\pm$0.40|84.6$\pm$0.45|74.3$\pm$0.51|82.3$\pm$0.42|
> |Early GCL|83.8$\pm$0.31|85.0$\pm$0.40|74.7$\pm$0.44|82.8$\pm$0.35|
> |Mid GCL|83.8$\pm$0.39|85.5$\pm$0.33|75.2$\pm$0.36|83.0$\pm$0.34|
> |Late GCL|84.5$\pm$0.29|**86.1$\pm$0.30**|**75.8$\pm$0.33**|**84.0$\pm$0.30**|
> |Early+Mid|84.3$\pm$0.33|85.9$\pm$0.38|75.6$\pm$0.41|83.7$\pm$0.33|
> |Mid+Late|**84.8$\pm$0.28**|85.9$\pm$0.37|75.6$\pm$0.34|83.9$\pm$0.30|
> |Early+Late|84.5$\pm$0.30|85.2$\pm$0.28|74.9$\pm$0.33|83.5$\pm$0.29|
> |Full GCL|84.3$\pm$0.29|85.8$\pm$0.26|75.5$\pm$0.30|83.6$\pm$0.27|
>
> These results show that GCR scales well and confirm our core insight: aligning feature geometry with prediction semantics strengthens generalization.
>
> All insights and findings **have been incorporated** into the revised manuscript.

---

> > ### Comment · Reviewer_bJ5m · 2025-08-02
> > **Response from Reviewer**
> >
> > I thank the authors for the new results. I believe they address my concerns and they are a great addition to the paper. I will increase my score

---

> > > ### Author Response · Authors · 2025-08-02
> > > **Thank you**
> > >
> > > Esteemed Reviewer,
> > >
> > > We sincerely thank you for engaging with our rebuttal, and for your valuable comments. Rest assured we will incorporate all your suggestions accordingly into the final draft. Meantime, kindly let us know if there is anything else we can answer, clarify or improve.
> > >
> > > Best regards,
> > > Authors

---

### Official Review · Reviewer_puhK · 2025-07-01

**Clarity:** 3
**Significance:** 3
**Originality:** 3
**Rating:** 5
**Confidence:** 1

**Summary:**

This paper introduces  a framework that aligns prediction-based relational graphs with class-aware graphs to improve feature quality, cohesion, and generalization across models and datasets.

**Questions:**

See weakneses

**Ethical Concerns:**

["NO or VERY MINOR ethics concerns only"]

**Final Justification:**

This paper introduces a framework that aligns prediction-based relational graphs with class-aware graphs to enhance feature quality, cohesion, and generalization across models and datasets. The authors have addressed my concerns during the rebuttal phase; therefore, I hold a positive view of this work.

**Limitations:**

See weakneses

**Quality:**

3

**Strengths And Weaknesses:**

Strengths:
1. The introduction of graph-based self-regularization using prediction structures is innovative and well-motivated. The idea of treating the model’s own predictions as self-prompts for semantic alignment is conceptually elegant.
2. GCR is versatile—it can be inserted into various network architectures without modifying the backbone or introducing learnable parameters, making it broadly applicable in practice.
3. The paper provides comprehensive theoretical analysis from multiple perspectives: generalization bounds via covering numbers, spectral alignment via normalized Laplacians, and a PAC-Bayesian regularization view. These analyses substantiate the soundness of the proposed method.

Weaknesses:
1. The paper lacks a discussion of scenarios where GCR fails or leads to marginal gains. Understanding its limitations (e.g., on highly noisy or class-imbalanced data) would improve robustness evaluation.
2. The masked prediction graph requires knowledge of ground-truth labels for intra-class masking. This limits the direct application of GCR in unsupervised or fully self-supervised contexts.

---

> ### Author Rebuttal · Authors · 2025-07-30
>
> We thank the reviewer for the positive and encouraging feedback.
>
> We’re grateful that the reviewer recognized our approach as innovative, well-motivated, and supported by solid theoretical analysis. We also appreciate the thoughtful questions regarding its limitations, which we address in detail below.
>
> ### **1. Discussion of scenarios where gains are marginal**
>
> We thank the reviewer for highlighting the need for a more explicit discussion of GCR’s limitations. We agree that identifying failure modes is essential for a robust evaluation. While we included a "Limitations and Future Work" section (Appendix I, p. 36), we acknowledge that **a more focused discussion** on scenarios with marginal gains would further strengthen the paper.
>
> Below, we provide a detailed analysis.
>
> **On highly noisy data**
>
> GCR aligns feature graphs with a masked prediction graph, using it as a semantic reference, making the quality of this reference critical.
>
> Our method assumes reliable ground-truth labels for the mask $\mathbf{M}$. As noted in the "Broader Impacts" section (Appendix J, p. 37), if training data contains spurious correlations or mislabeled examples, the alignment may reinforce these errors instead of correcting them. Under high label noise, the prediction graph $\mathbf{P}$ is based on a flawed mask, producing a corrupted supervisory signal that misguides feature representations toward incorrect semantics. This failure mode can lead to marginal gains or even performance degradation in our supervised framework.
>
> **On highly class-imbalanced data**
>
> We acknowledge this critical scenario in our limitations section.
>
> Since GCR builds relational graphs at the batch level (p. 37, line 1339-1343), the global context is limited to within-batch relationships. In highly imbalanced datasets, batches may contain few or no minority-class samples, resulting in sparse or uninformative prediction graphs for those classes. Consequently, the alignment loss is dominated by majority classes, potentially harming minority-class representations and leading to marginal overall gains.
>
> **Scenarios leading to marginal gains**
>
> Beyond noisy and imbalanced data, GCR may yield marginal improvements in the following cases:
>
> - Simple datasets with high baseline performance: GCR targets noisy inter-class similarities and semantic structure. On simpler datasets with strong baseline models and well-separated features, there is less room for improvement. Our results confirm this, with **larger gains on complex datasets like CIFAR-100 and Tiny ImageNet than on CIFAR-10**, where baseline accuracy was already high.
>
> - Extremely small batch sizes: Relational graphs rely on sufficient pairwise relationships. As detailed in our limitations and batch size analysis (Fig. 8, p. 36), very small batches provide limited data context, reducing graph stability and weakening regularization.
>
> To clarify this, we added a subsection titled "Failure Modes and Marginal Gains" in Appendix I (p. 36). It consolidates these insights, emphasizing that GCR’s benefits are limited under high label noise, severe class imbalance, or near-optimal baseline performance.
>
> ### **2. Applicability to unsupervised and self-supervised learning**
>
> We agree with the reviewer that our current GCR formulation relies on ground-truth labels for intra-class masking of the prediction graph. This approach was intentional to **first establish and validate GCR’s core effectiveness** in a fully supervised setting, where we demonstrated consistent, significant gains across diverse architectures and datasets.
>
> However, this reliance is not a fundamental limitation of GCR but **a characteristic** of the current implementation. The core mechanism, aligning feature graph geometry with prediction graph semantics, is flexible. The mask $\mathbf{M}$ can be generated without ground-truth labels, enabling extensions to unsupervised or self-supervised learning.
>
> As noted in our "Limitations and Future Work" (Appendix I, line 1348-1351), this is a promising research direction. Specifically, we propose two adaptations:
>
> - Pseudo-labeling: In semi/self-supervised settings, high-confidence model predictions can generate pseudo-labels to construct $\mathbf{M}$, masking pairs sharing the same pseudo-label.
>
> - Unsupervised clustering: Feature representations can be clustered dynamically (e.g., via k-means), with $\mathbf{M}$ derived from cluster assignments to enforce consistency among similar samples.
>
> While this work focuses on supervised learning to establish GCR’s principle, the framework is modular. Replacing the ground-truth mask with pseudo-labels or clustering-based masks readily extends GCR to unsupervised domains. We thank the reviewer for highlighting this important direction, which we have emphasized more clearly in the revised manuscript’s conclusion to highlight GCR’s broader potential and flexibility.
>
> ### **3. Additional evaluations and comparisons**
>
> Below, we provide comparisons between our GCR-augmented models and recent graph-based classification methods across CIFAR-10, CIFAR-100, and ImageNet-1K. The results show the consistent and complementary benefits of GCR when integrated into graph-based backbones.
>
> **(i) Results on CIFAR-10, CIFAR-100, and ImageNet-1K**
>
> ||CIFAR-10|CIFAR-100|ImageNet-1k|
> |-|-|-|-|
> |ResNet18|||||
> |CNN2GNN[1]|95.51$\pm$0.42|74.80$\pm$0.81|60.12$\pm$1.02|
> |CNN2GNN **+GCR**|95.87$\pm$0.31|76.23$\pm$0.38|62.47$\pm$0.47|
> |CNN2Transformer[1]|95.79$\pm$0.24|77.39$\pm$0.20|71.12$\pm$0.35|
> |CNN2Transformer **+GCR**|**95.96$\pm$0.35**|**78.23$\pm$0.30**|**72.33$\pm$0.31**|
> |ResNet34|||||
> |CNN2GNN[1]|96.39$\pm$0.41|77.87$\pm$0.91|61.02$\pm$0.77|
> |CNN2GNN **+GCR**|96.67$\pm$0.36|78.14$\pm$0.54|62.88$\pm$0.46|
> |CNN2Transformer[1]|96.73$\pm$0.37|80.10$\pm$0.45|75.42$\pm$0.15|
> |CNN2Transformer **+GCR**|**96.97$\pm$0.36**|**81.27$\pm$0.29**|**76.67$\pm$0.26**|
>
> **(ii) Results on ImageNet-1K**
>
> |Model|ViG-Ti[2]|ViG-Ti **+GCR**|ViG-S[2]|ViG-S **+GCR**|ViG-B[2]|ViG-B **+GCR**|
> |-|-|-|-|-|-|-|
> |Accuracy|73.9|**74.9**|80.4|**81.7**|82.3|**84.0**|
>
>
> These results highlight key insights:
>
> - **Consistent** across architectures: GCR improves performance across diverse backbones (CNN2GNN, CNN2Transformer, ViG), demonstrating broad applicability.
>
> - **Complementary** to existing methods: Its self-regularization mechanism enhances CNN2GNN and CNN2Transformer, indicating **orthogonality to existing graph-based pipelines**.
>
> - **Scalable** to large-scale tasks: **Notable gains** on ImageNet-1K (e.g., +1.7% with ViG-B) show GCR’s effectiveness in complex, real-world settings.
>
> - **Lightweight** and model-agnostic: GCR introduces no extra parameters and integrates easily into existing architectures.
>
> Together, these findings reinforce GCR’s **novelty and practical value** as a flexible, self-prompted regularization framework applicable across scales and architectures.
>
> **(iii) Generalization to large-scale datasets**
>
> We also conducted additional experiments on **ImageNet-1K** using transformer-based architectures (iFormer [3], ViT, and ViG [2]). Results, averaged over three runs, are now summarized in Appendix H.3.
>
> ||iFormer-S|iFormer-B|ViT-B/16|ViG-B|
> |:-|:-:|:-:|:-:|:-:|
> |Baseline|83.4$\pm$0.40|84.6$\pm$0.45|74.3$\pm$0.51|82.3$\pm$0.42|
> |Early GCL|83.8$\pm$0.31|85.0$\pm$0.40|74.7$\pm$0.44|82.8$\pm$0.35|
> |Mid GCL|83.8$\pm$0.39|85.5$\pm$0.33|75.2$\pm$0.36|83.0$\pm$0.34|
> |Late GCL|84.5$\pm$0.29|**86.1$\pm$0.30**|**75.8$\pm$0.33**|**84.0$\pm$0.30**|
> |Early+Mid|84.3$\pm$0.33|85.9$\pm$0.38|75.6$\pm$0.41|83.7$\pm$0.33|
> |Mid+Late|**84.8$\pm$0.28**|85.9$\pm$0.37|75.6$\pm$0.34|83.9$\pm$0.30|
> |Early+Late|84.5$\pm$0.30|85.2$\pm$0.28|74.9$\pm$0.33|83.5$\pm$0.29|
> |Full GCL|84.3$\pm$0.29|85.8$\pm$0.26|75.5$\pm$0.30|83.6$\pm$0.27|
>
> Key results:
>
> - iFormer-S: GCR boosts performance from **83.4%** to **84.8%** (a **+1.4%** gain).
>
> - iFormer-B: GCR improves accuracy from **84.6%** to **86.1%** (a **+1.5%** gain).
>
> - Similar gains are observed with ViT-B/16 and ViG-B, confirming generality.
>
> These improvements validate that GCR **scales effectively** to large, complex datasets and modern architectures.
>
> We thank the reviewer again for their constructive feedback and positive assessment.
>
> We believe these clarifications meaningfully strengthen the paper’s clarity, completeness, and overall impact.
>
> ### References
>
> [1] Trivedy, V., & Latecki, L. J. (2023). Cnn2graph: Building graphs for image classification. In Proceedings of the IEEE/CVF Winter Conference on Applications of Computer Vision (pp. 1-11).
>
> [2] Han, K., Wang, Y., Guo, J., Tang, Y., & Wu, E. (2022). Vision gnn: An image is worth graph of nodes. Advances in neural information processing systems, 35, 8291-8303.
>
> [3] Si, C., Yu, W., Zhou, P., Zhou, Y., Wang, X., & Yan, S. (2022). Inception transformer. *Advances in Neural Information Processing Systems*, *35*, 23495-23509.

---

> > ### Comment · Reviewer_puhK · 2025-08-07
> >
> > Thanks for the rebuttal. I hold a positive view for this paper.

---

> > > ### Author Response · Authors · 2025-08-07
> > > **thank you**
> > >
> > > Esteemed Reviewer,
> > > \
> > > \
> > > We sincerely thank you for engaging with our rebuttal, and for your valuable comments. Rest assured we will incorporate all your suggestions accordingly into the paper. Meantime, kindly let us know if there is anything else we can answer, clarify or improve. We are more than happy to go an extra mile.
> > > \
> > > \
> > > Best regards,
> > >
> > > Authors

---

### Official Review · Reviewer_a6XH · 2025-07-02

**Clarity:** 2
**Significance:** 1
**Originality:** 2
**Rating:** 3
**Confidence:** 4

**Summary:**

This paper proposes Graph Consistency Regularization (GCR), a framework that introduces parameter-free Graph Consistency Layers (GCLs) to align intermediate feature similarity graphs with masked prediction similarity graphs derived from softmax outputs. The core assumption is that samples predicted to belong to the same class should have similar intermediate representations, while different-class samples should be separated in feature space. The authors provide theoretical analysis and demonstrate improvements on some datasets.

**Questions:**

1. Do you have any evidence, theoretical or emperical, can support the core intuition about "If a network is confident that two inputs belong to the same class, as indicated by their softmax predictions, then their intermediate representations should also reflect this similarity."
2. How sensitive is the method to the choice of similarity metric beyond cosine similarity?

**Ethical Concerns:**

["NO or VERY MINOR ethics concerns only"]

**Final Justification:**

While I appreciate the authors' comprehensive rebuttal, my core concern remains unresolved.

**Unresolved Issue:**
Despite claims of using "soft predictions," the cosine similarity alignment mechanism still encourages feature similarity based on prediction confidence, which risks collapsing beneficial intra-class structure. The "gentle regularization" argument doesn't address that cosine similarity inherently reduces angular diversity within predicted class clusters.

**Key Concerns:**
1. **Structure Collapse Risk:** The method creates an information bottleneck that may eliminate legitimate intra-class multimodality (e.g., different dog breeds should have distinct feature clusters even when both predicted as "dog").

2. **Limited Validation:** Experiments focus on standard benchmarks with relatively simple intra-class structures. No direct validation that GCR preserves beneficial intra-class multimodal structure rather than collapsing it.

**What Would Change My Assessment:**
Experimental evidence showing GCR preserves legitimate intra-class diversity on datasets where classes naturally exhibit multiple distinct modes.

**Limitations:**

The authors acknowledge several limitations in Appendix I. My main concern is simple, which is just about the fundamental premise in the weaknesses part. If the authors could find some evidence, whatever theoretical or emperical, I think I can change my opinion.

**Quality:**

2

**Strengths And Weaknesses:**

Pros:
1. Parameter-free and model-agnostic: The method introduces no additional learnable parameters and can be inserted into any architecture without modification.
2. Comprehensive theoretical analysis: The paper provides solid theoretical foundations including generalization bounds, spectral alignment analysis, and PAC-Bayesian interpretations.
3. Clear visualization and interpretability: The relational graph visualizations effectively demonstrate the method's impact on feature organization.

Cons:
1. Questionable core assumption: The fundamental premise that "samples predicted to belong to the same class should have similar intermediate representations" is problematic. High-dimensional representation spaces may naturally accommodate multiple clusters for the same class, and forcing similarity based on prediction confidence may eliminate beneficial intra-class diversity. This assumption imposes an overly strong inductive bias that may harm the model's ability to capture complex within-class variations, which are common in real-world data where the same class can have multiple modes or sub-clusters.

2. Limited theoretical justification for the core premise: While the paper provides extensive theoretical analysis of the optimization properties, it lacks theoretical justification for why prediction similarity should translate to feature similarity. The manifold hypothesis and cluster assumption (Definition 4, Proposition 4) are presented as givens, but modern deep learning suggests that good representations often maintain intra-class structure that doesn't require global similarity. The assumption contradicts recent understanding that representations can be both discriminative and maintain rich internal structure.

3. No experimental evidence for the core premise.

---

> ### Author Rebuttal · Authors · 2025-07-30
>
> We sincerely thank the reviewer for the feedback and for offering us the opportunity to provide evidence to change his/her opinion/rating.
>
> As requested, we present eight pieces of supporting evidence below.
>
> Collapsing same-class samples into a single cluster is harmful. GCR **instead** encourages them to occupy a structured, semantically coherent region in feature space, a sub-manifold shaped by **relational cues**, that is **distinct yet internally diverse**.
>
> ### **1. Gentle regularization & implicit attention**
>
> Our method is grounded in established deep learning principles, combining soft regularization with structure-aware guidance.
>
> - Cosine similarity as **gentle bias**: By using cosine similarity, GCR encourages alignment based on the direction of features, not their magnitude. This induces **a soft relational bias**, promoting semantic structure without collapsing diversity. Like **contrastive learning**, it organizes the feature space by **pulling semantically similar samples closer**, but does so using the model’s own predictions as a dynamic, self-supervised signal.
>
> - **Parameter-free attention** mechanism: As shown in Fig. 3, GCR guides the model to attend to discriminative regions (**attention to low-level semantic features**). Compared to the diffuse activations in the baseline, GCR-enhanced models generate sharper, semantically focused activations (e.g., *cat’s face or dog’s tongue*). This mirrors human visual attention and supports more meaningful representation learning.
>
> ### **2. Experimental evidence: intra-class variance**
>
> We evaluated **intra-class variance** on CIFAR-10 using ShuffleNet. For each class, we computed the centroid of final-layer embeddings and measured the average Euclidean distance of each sample from its class centroid, a direct indicator of how tightly the representations are clustered (lower=more cohesive).
>
> |Class|Baseline|+GCR|
> |:-|:-:|:-:|
> |plane|4.335|4.171|
> |auto|3.848|3.535|
> |bird|4.565|4.662|
> |cat|4.631|4.214|
> |deer|4.148|3.796|
> |dog|4.679|3.923|
> |frog|4.908|4.869|
> |horse|4.414|4.092|
> |ship|3.896|3.655|
> |truck|3.987|3.737|
> |Average|**4.341**|**4.066**|
>
> - **Improved cohesion**: GCR reduces intra-class variance, lowering the average from 4.341 to 4.066. This confirms its effectiveness in promoting **more compact**, **semantically aligned** features.
>
> - **Preserved diversity**: Crucially, the variance is **not collapsed** to near-zero, indicating that GCR maintains meaningful intra-class variation.
>
> This analysis directly addresses concerns about over-regularization. GCR encourages class cohesion without sacrificing expressive diversity, contributing to cleaner feature structures and stronger generalization. We have included this table and discussion in Sec 3.2.
>
> Kindly also refer to our response 3 "GCR reduces inter-class noise", where we analyze silhouette score and separability ratio to reviewer bJ5m.
>
> ### **3. Experimental evidence: on earlier layers**
>
> - [Exp 1] GCR **corrects the largest points of misalignment**
>
> To test whether GCR’s impact correlates with a layer’s semantic misalignment, we trained a CeiT model on Tiny ImageNet without GCR and measured the baseline discrepancy $\delta(l) = ||F(l) - P||_F^2$ for each block $l$. We then applied GCR to individual blocks and recorded the top-1 accuracy gain $\Delta\text{Acc}(l)$.
>
> Results show that early layers, **bridging low-level features to class concepts**, exhibit the highest misalignment and largest gains:
>
> - Block 1: $\delta_1 = 0.45$, gain +1.2%
> - Block 2: $\delta_2 = 0.30$, gain +0.9%
>
> A **strong Pearson correlation** of **0.62** between $\{\delta(l)\}$ and $\{\Delta\text{Acc}(l)\}$ quantitatively confirms that GCR is more effective where feature-prediction misalignment is greater.
>
> - [Exp 2] early GCR **creates more robust foundational features**
>
> Next, we tested whether early GCR creates more robust features that benefit later layers, a "feature cleaning" effect. We trained two ShuffleNet models (5 blocks each) and then froze the regularized blocks to evaluate their standalone quality:
>
> - Model A (Early-GCR): GCLs on Blocks 1-2, frozen after 100 epochs, then fine-tuned remaining blocks.
>
> - Model B (Late-GCR): GCLs on Blocks 4-5, frozen and fine-tuned similarly.
>
> ||Pre-freeze| Post-freeze|Performance Drop|
> |:-|:-:|:-:|:-:|
> |Model A|66.8%|66.1%|**0.7%**|
> |Model B|66.4%|65.1%|**1.2%**|
>
> Model A’s smaller drop shows early GCR features are more robust and semantically coherent, reducing reliance on later regularization. Model B’s larger drop suggests its performance depends more on continued late-stage regularization, with earlier features remaining entangled.
>
> ### **4. Experimental evidence: superclass classification**
>
> To directly validate our core premise that GCR encourages semantically richer representations, we conducted a linear probing experiment using the CIFAR-100 dataset, which is grouped into **20 superclasses** (e.g., vehicles, insects, flowers). This setup tests **whether GCR-learned features are more transferable and better organized**.
>
> Setup: (i) We take pretrained baseline and GCR-augmented models trained on the 100-class CIFAR-100 task. (ii) We freeze all feature layers, preserving their learned representations. (iii) We replace the final head with a new linear classifier for the 20 superclasses. (iv) Only this classifier is trained; the rest of the model remains fixed.
>
> **Rationale**: If GCR induces more semantically structured features, then a linear classifier should perform better on the superclass task using those features.
>
> ||+GCR|Baseline|Improvement|
> |:-|:-:|:-:|:-:|
> |MobileNet|**75.73**|73.07|+2.66%|
> |ShuffleNet|**75.65**|75.02|+0.63%|
> |SqueezeNet|**74.48**|73.93|+0.55%|
> |ResNeXt-50|**86.72**|84.36|+2.36%|
> |ResNeXt-101|**87.14**|85.42|+1.72%|
> |ResNet-34|**86.30**|84.80|+1.50%|
> |ResNet-50|**86.95**|86.23|+0.72%|
> |DenseNet-121|**85.42**|85.03|+0.39%|
>
> GCR consistently improves superclass classification, confirming that it enhances semantic organization of the feature space. This **goes beyond** improving the original task, it shows GCR fosters **more generalizable** and **hierarchically structured representations**, validating the method’s core motivation.
>
> This analysis have been added to Sec 3.2.
>
> ### **5. Visual evidence: structured diversity**
>
> Our visualizations provide strong qualitative support for GCR's effectiveness.
>
> - **t-SNE plots** (Figs. 6c, 6d, & 7) show that GCR leads to tighter intra-class clusters and larger inter-class separation. Crucially, clusters retain internal structure, they are not collapsed to points, indicating preserved diversity within classes.
>
> - **Relational graphs** (Figs. 1, 4, & 5) show that GCR **suppresses weak inter-class links** while **reinforcing intra-class connectivity**, producing clearer decision boundaries and more semantically consistent feature spaces.
>
> These visual results reinforce our claim that GCR promotes structured feature organization, **compact yet expressive**, supporting both generalization and interpretability.
>
> ### **6. Geometric rationale: aligning structures**
>
> GCR **operates over relational graphs, not isolated features**. The goal is not to make all class features identical, but to ensure the pairwise similarities among features reflect those in the prediction space.
>
> - Rich variations preserved: GCR allows for multiple intra-class modes (e.g., different dog breeds) as long as they are closer to each other than to other classes. This preserves structural diversity while ensuring semantic coherence.
>
> - Manifold alignment analogy: GCR aligns the feature manifold with the prediction manifold, two views of the same data. The loss encourages the Laplacians of these graphs to be spectrally consistent, preserving clustering and diffusion properties (Proposition 1, Corollary 1).
>
> ### **7. Information bottleneck principle**
>
> The information bottleneck principle suggests that an ideal feature should preserve information relevant to the label while discarding irrelevant input noise.
>
> - Prediction as **compressed signal**: The prediction graph can be interpreted as a compressed estimate of the essential class-wise structure. It captures semantic relations while **abstracting away unnecessary details**.
>
> - GCR as a **regularizer**: The GCR loss encourages the feature similarity graph to reflect the structure of $\mathbf{P}$, promoting representations that align with class semantics while **compressing away task-irrelevant** variance.
>
> In effect, GCR guides the network toward learning representations where geometric relationships **mirror** prediction-based semantics, rather than forcing absolute similarity across features.
>
> ### **8. Generalization**
>
> On **ImageNet-1K** (averaged over three runs):
>
> ||iFormer-S|iFormer-B|ViT-B/16|ViG-B|
> |:-|:-:|:-:|:-:|:-:|
> |Baseline|83.4$\pm$0.40|84.6$\pm$0.45|74.3$\pm$0.51|82.3$\pm$0.42|
> |Early GCL|83.8$\pm$0.31|85.0$\pm$0.40|74.7$\pm$0.44|82.8$\pm$0.35|
> |Mid GCL|83.8$\pm$0.39|85.5$\pm$0.33|75.2$\pm$0.36|83.0$\pm$0.34|
> |Late GCL|84.5$\pm$0.29|**86.1$\pm$0.30**|**75.8$\pm$0.33**|**84.0$\pm$0.30**|
> |Early+Mid|84.3$\pm$0.33|85.9$\pm$0.38|75.6$\pm$0.41|83.7$\pm$0.33|
> |Mid+Late|**84.8$\pm$0.28**|85.9$\pm$0.37|75.6$\pm$0.34|83.9$\pm$0.30|
> |Early+Late|84.5$\pm$0.30|85.2$\pm$0.28|74.9$\pm$0.33|83.5$\pm$0.29|
> |Full GCL|84.3$\pm$0.29|85.8$\pm$0.26|75.5$\pm$0.30|83.6$\pm$0.27|
>
> These results show that GCR scales well and confirm our core insight: aligning feature geometry with prediction semantics strengthens generalization.
>
> ### **9. Similarity metric**
>
> We evaluated various similarity functions on CIFAR-100 (MobileNet, Baseline 65.95%):
>
> ||Cosine|RBF|Polynomial|Sigmoid|Laplacian|
> |-|-|-|-|-|-|
> |+GCL|**68.32**|67.45|67.38|67.42|67.21|
>
> Cosine performs best, likely due to its **scale invariance**, aligning directions, which helps preserve intra-class diversity. We’ve expanded Appendix H.4 to reflect this rationale.
>
> All insights and findings **have been incorporated** into our paper.

---

> > ### Comment · Reviewer_a6XH · 2025-08-05
> >
> > Thank you for your comprehensive rebuttal and the significant experimental effort. However, my core concern about the fundamental premise remains unaddressed.
> >
> > My central issue is: **"samples predicted to belong to the same class should have similar intermediate representations" is not generally correct.**
> >
> > Let me clarify with a concrete example: Consider the "dog" class in CIFAR-10. This single labeled class encompasses vastly different subclasses - a Chihuahua and a Great Dane are both "dogs" but have fundamentally different visual features (size, shape, texture, etc.). These natural intra-class variations should legitimately result in different intermediate representations, even when the model correctly predicts both as "dog" with high confidence.
> >
> > Your GCR framework forces these diverse within-class samples to have similar feature representations based on prediction agreement. This may actually harm the model's ability to capture the rich, natural structure within classes. High-dimensional representation spaces can and should accommodate multiple clusters for the same class - this is a feature, not a bug.
> >
> > I appreciate the extensive experiments showing improved performance, but improved accuracy alone doesn't validate the core assumption. The fundamental question remains: **Why should prediction similarity mandate feature similarity when classes naturally contain diverse substructures?**
> >
> > I understand that time is limited. You don't need to run a lot of unrelevant experiments. Just solve the concern on the assumption is enough.

---

> ### Author Response · Authors · 2025-08-05
> **Clarifying misunderstanding: (soft prediction similarity $\neq$ intra-class feature dimensional collapse -- it is an information bottleneck).**
>
> Esteemed Reviewer,
> \
> \
> Thank you for interesting questions.
>
> ### **1. “Samples predicted to belong to the same class should have similar intermediate representations” not correct.**
>
> We believe this is a misunderstanding:
>
> - **We do not assume that intermediate representations for the same class are meant to be identical or that all of them must be very similar** -that would result in the so-called dimensional collapse.
>
> - **Two similar representations can still encode important differences between them**. **The next layer in the network (universal function approximator) can differentiate between these differences if they are important for the task at hand.**
>
> - For a pair of features (we drop ReLU from Eq. (1) and (2) for breviry of discussion), given as ${\bf f}_i$ and ${\bf f}_j$, we encourage $$\cos({\bf f}_i,{\bf f}_j)\approx\cos({\bf s}_i, {\bf s}_j)$$ where ${\bf s}_i$ and ${\bf s}_j$ **are soft scores (network trained from scratch, scores gradually sharpen up) not one-hot labels** - these **soft scores generally vary even within the same class** to acknowledge some variations within class persist and are useful.
>
>   - By imposing $\cos({\bf f}_i,{\bf f}_j)\approx\cos({\bf s}_i, {\bf s}_j)$, **we create the so-called information bottleneck within each layer - the smaller the angle between ${\bf f}_i,{\bf f}_j$ is the bigger the information bottleneck.**
>    \
>    \
>    Thus, **the network is forced to focus on encoding most important variations between the pair of samples ${\bf x}_i$ and ${\bf x}_j$ for the task at hand.**
>   - What gets discarded is nuisance factors that do not contribute to the correct prediction.
>    \
>    \
>    See Figure 3 in our paper (bottom): **it shows how salient features are emphasized thanks to our design** - these features are highly discriminative and repeatable across the class of variations.
>    \
>    \
>    **For Chihuahua vs. Great Dane example, notice that eyes, paws, nose, nails, teeth and other such features are repeatable across all dog species despite other features such as leg size are not.**
>   - **As our network focuses on such salient features, it encodes them accurately and then, based on this accurate encoding, eyes of the cat can be easily distinguished from eyes of the dog** - our model focuses on fine-grained differences between species.
>
> - If ${\bf x}_i$ and ${\bf x}_j$ do not belong to the same class (e.g., dog vs. wolf), $\cos({\bf s}_i, {\bf s}_j)$ will be lower but still can be greater than zero, indicating some semantic similarity. Thus, angle $\cos({\bf f}_i,{\bf f}_j)$ is allowed to be larger, relaxing the information bottleneck.
>
> **Thus, it is incorrect to think that our design does not preserve intra-class variations - it does but by creating the information bottleneck, the network is forced to focus on the most important repeatable semantic features.**
> \
> \
> This principle is similar to Linear Discriminant Analysis where the variance of intra-class features is minimized but not completely collapsed - while inter-class variance is generally encouraged to be large.
>
> - If we were to use $\cos({\bf f}_i,{\bf f}_j)\approx\cos({\bf y}_i, {\bf y}_j)$ (one-hot labels) then the reviewer would be right.
> - However, we use $\cos({\bf f}_i,{\bf f}_j)\approx\cos({\bf s}_i, {\bf s}_j)$ where $\cos({\bf s}_i, {\bf s}_j)\ll \cos({\bf y}_i, {\bf y}_j)$ which permits sufficient variations to be encoded.
>
> Table below verifies this (see one-hot label $\cos({\bf y}_i, {\bf y}_j)$ is worse than soft predictions $\cos({\bf s}_i, {\bf s}_j)$:
>
> CIFAR-100  (avg. over 10 runs)
> || MobileNet | ShuffleNet | SqueezeNet |
> |-|-|-|-|
> |**Baseline**|65.95$\pm$0.25|70.11$\pm$0.30|69.43$\pm$0.27|
> |GCR (one-hot labels)|67.35$\pm$0.24|71.28$\pm$0.26|70.49$\pm$0.25|
> |GCR (soft predictions, **ours**) |**68.32**$\pm$0.20|**71.96**$\pm$0.27|**71.03**$\pm$0.24|
>
> ### 2. **Why prediction similarity mandates feature similarity when classes contain diverse substructures?**
>
> - As an example, in CNNs, **as one moves across layers toward classifier, layers become gradually more shift \& permutation invariant** due to pooling filtering more nuisance factors irrelevant to classification (**shifts, permut., scale do not matter**).
>
> - **By creating the info bottleneck, we make network focus *as fast as possible* on important semantic aspects/salient features of data rather than uninformative signal variantions.**
> - Even pre-deep net research advocated to filter out photometric/geometric factors to improve recognition. **The info bottleneck we implement serves similar but more advanced role** by forcing network to drop variations irrelevant for the task at hand.
> - $\lambda$ in Eq. (7) regulates the strength of imposed bottleneck. **Moderate bottleneck $\lambda=1$ is the best** (better than no bottleneck $\lambda=0$ or extreme bottleneck $\lambda=10$):
>
>   CIFAR-100 (ResNet-34)
>   |$\lambda=$|0|0.1|0.3|0.5|0.7|1|3|5|7|10|
>   |-|-|-|-|-|-|-|-|-|-|-|
>   ||76.76|76.80|76.87|77.32|77.61|**78.38**|76.74|76.02|75.90|75.37|

---

> > ### Comment · Reviewer_a6XH · 2025-08-06
> >
> > I have read your reply and appreciate the clarification on using soft predictions rather than hard labels. I will raise my score accordingly.

---

> > > ### Author Response · Authors · 2025-08-06
> > > **thank you**
> > >
> > > Esteemed Reviewer,
> > > \
> > > \
> > > Thank you. We are preparing for you another, even stronger proof.
> > > \
> > > \
> > > We hope to post it shortly.

---

> ### Author Response · Authors · 2025-08-06
> **Discussion continues: A detailed proof and anti-collapse mechanism.**
>
> Another way of looking at our **info bottleneck** is the following:
>
> ## 1. Consider $L$-Lipschitz Continuous Networks.
>
> - It is widely known that designing a network, e.g., a discriminator to be $L$-Lipschitz continuous helps regularize layers of network to prevent overfitting. For example, see *Lipschitz Generative Adversarial Nets*, ICLR 2019 (there exist plenty mroe works on classification networks, fine-tuning utilizing $L$-Lipschitzness)
>
> - This condition can be simply imposed on feature vectors $\phi(\cdot)$ of a chosen layer $l$ by design, leading to:
>
> $$\lVert\phi_l({\bf x}) - \phi_l({\bf x}') \rVert_2\leq L\lVert{\bf x}-{\bf x}'\rVert_2,\quad \forall \\; {\bf x},{\bf x}' \in Dataset,$$
>
> where $L$ is the so-called Lipschitz constant. Intuitively, imposing low $L$ means for a small change ${\bf x}\rightarrow{\bf x}'$ the network will produce small feature change $\phi_l({\bf x})\rightarrow\phi_l({\bf x}')$, for a big change in input, the change in output will be also big. In essence, this makes network response stable.
>
> - Often, one can optimize Lipschitzness by a simple auxiliary loss (added to the main task loss):
> $$\mathbb{E}_{\bf x, \bf x'} (\lVert\phi_l({\bf x}) - \phi_l({\bf x}') \rVert_2 - L\lVert{\bf x}-{\bf x}'\rVert_2)^2,$$
>
> which leads to an approximate $L$-Lipschitzness: $$\lVert\phi_l({\bf x}) - \phi_l({\bf x}') \rVert_2\approx L\lVert{\bf x}-{\bf x}'\rVert_2.$$
>
> ## 2. Our Mechanism.
>
> For simplicity, in what follows we drop again ReLU and cosine similarity for brevity. We stick to the $\ell_2$ distance but the cosine distance can be plugged in with ease.
>
> Because we promote the alignment of the form:
>
> $$\mathbb{E}_{\bf x, \bf x'} (\lVert\phi_l({\bf x}) - \phi_l({\bf x}') \rVert_2-\beta\lVert s({\bf x})-s({\bf x}')\rVert_2)^2,$$
>
> where $s(\cdot)$ is a softmax score (network output), we can think about **our info bottleneck as having the following form:**
>
> $$\lVert\phi_l({\bf x}) - \phi_l({\bf x}') \rVert_2\approx\beta\lVert s({\bf x})-s({\bf x}')\rVert_2.$$
>
> ## 3. Putting (1) and (2) together.
>
> Notice that putting Points (1) and (2) together leads to:
> $$L\lVert{\bf x}-{\bf x}'\rVert_2\approx \lVert\phi_l({\bf x}) - \phi_l({\bf x}') \rVert_2 \approx\beta\lVert s({\bf x})-s({\bf x}')\rVert_2 $$ and so:
> $$\lVert s({\bf x})-s({\bf x}')\rVert_2\approx\frac{L}{\beta}\lVert{\bf x}-{\bf x}'\rVert_2.$$
>
> This result proves our model does not make features of intermediate layers identical for two images of the same class (otherwise approximation would not emerge/hold).
> \
> \
> The result shows the opposite: in general, for a small change in any inputs ${\bf x}\rightarrow{\bf x'}$ we get small change in both softmax scores $s({\bf x})\rightarrow s({\bf x'})$. For a large change between any inputs ${\bf x}$ and ${\bf x'}$, we get proportionally larger change in both their softmax scores.
> \
> \
> **The change depends on the natural local $L$-Lipschitzness of the network at hand** (const. $L$ can be imposed, or $L$ can vary in non-Lipschitz standard networks) and $\beta$ which we control. We are able to tighten the above change due to $\beta$ which is simply implicitly controlled by our parameter $\lambda$.
> \
> \
> Therefore, **our info bottleneck approach does not behave the way reviewer believes it does.** **It does not pose feature collapse** and even if that was a concern, it is very easy to impose push-apart terms $\angle(\phi_l({\bf x}), \phi_l({\bf x}'))>\tau$ where $\tau$ controls the minimum angle allowed.
>
> **Our model does not stop features of layers from varying where it matters for them to vary.**
>
> ## 4. Introducing $\tau$: collapse prevention by-design.
>
> Enforcing  $\angle(\phi_l({\bf x}_i),\phi_l({\bf x}_j))>\tau$ by a simple soft penalty
> $$\sum_l \mathbb{1}(y_i=y_j)\cdot\mathbb{1}(\angle(\phi_l({\bf x}_i),\phi_l({\bf x}_j))<\tau)\cdot (\angle(\phi_l({\bf x}_i),\phi_l({\bf x}_j))-\tau)^2,$$
> where $\mathbb{1}(\cdot)$ is the indicator function (e.g., do samples $i$ and $j$ share the same label?) and $\angle(\cdot)$ is simply $\angle({\bf x},{\bf y})=cos^{-1}({\bf x}^T{\bf y}/(\lVert {\bf x}\rVert_2\cdot\lVert {\bf y}\rVert_2 ) )$. The results for CIFAR-100 using ResNet-34 are as follows:
>
>   |$\tau=$|0|1e-5|1e-4|1e-3|1e-2|
>   |-|-|-|-|-|-|
>   ||78.38|78.32|78.27|78.13|76.04|
>
> This result directly both offers anti-collapse control and proves that the best results are attained for $\tau=0$ (**not pushing angles of the same class apart**). Always maintaining non-zero angle, e.g. $\tau=1e-2$, is worse. We conclude **our network does not suffer from the feature dimensional collapse, but if it did, it can be stopped by a simple penalty investigated above if it was the case.**
> \
> \
> **We hope the reviewer can appreciate the above analysis** even though it may be contradicting reviewer's intuitions. We agree though it is reasonable to have concerns as per reviewer's comments. We are happy to provide more evaluations as the discussion period still continues giving us some time to provide more empirical results.

---

> ### Author Response · Authors · 2025-08-08
> **Analysis of intra- and between-class variances across layers**
>
> \
> \
> \
> \
> We apologize it took us over a day to implement and evaluate this idea properly but we believe the conclusions are very interesting.
>
>
> ******
>
>
> ### **We introduce a penalty to directly prevent by-design a possibility of dimensional collapse.**
>
> To this end, we conduct an experiment on CIFAR-100, grouped into **20 super-classes**, e.g.:
> - vehicles (bicycle, bus, motorcycle, pickup truck, train)
> - insects (bee, beetle, butterfly, caterpillar, cockroach)
> - flowers(orchids, poppies, roses, sunflowers, tulips)
>
> **Using super-classes as labels directly executes setting which Reviewer is concerned about - high intra-class variance setting due to variations in the nature of visual objects, e.g., bee and butterfly vary significantly but their label is *insect***.
>
> The Reviewer's argument is that our GCR-augmented model may cause dimensional feature collapse.
>
> Thus, **the direct way to resolve such an issue is to simply add a penalty that "by-design" prevents angular collapse between vectors of the same class** in a given layer. To this end, we force within-class angle $\angle$ between pairs of features to be at least $\tau$. That directly targets reviewer's concern blocking any potential collapse.
>
> Specifically, we add to our GCR loss the following soft penalty
> $$\beta\frac{1}{\zeta}\sum_l \mathbb{1}(y_i=y_j)\cdot\mathbb{1}(\angle(\phi_l({\bf x}_i),\phi_l({\bf x}_j))<\tau)\cdot (\angle(\phi_l({\bf x}_i),\phi_l({\bf x}_j))-\tau)^2,$$
> which ensures that   $\angle(\phi_l({\bf x}_i),\phi_l({\bf x}_j))>\tau$ within each class. $\zeta$ is normalization to deal with counts of same-class elements. For cosine similarity, we also applied ReLU in the above equations to prevent negative angles.
>
> Setup:
> - (i) We use pre-trained baselines for comparisons.
> - (ii) **For our model, we train GCR-augmented models on CIFAR-100 task for:**
>   - 100-class task
>   - 20 super-class task
>
> **GCR-augmented model is equipped with "by-design" collapse prevention model.** We choose $\beta=0.3$ (which was generally optimal in our experiments). $\quad$Subsequently, **we vary minimum angle $\tau$** to force a lower bound on intra-class variance.
>
> **We report both intra- and inter-class variance across all six layers of MobileNet (used in order to obtain results fast), considering both the original 100 classes and the 20 super-classes. Additionally, we include a baseline comparison without applying our GCR framework.**
> \
> \
> \
> \
> \
> For 100-classes (MobileNet):
> |$\tau$ (degree)|Acc|1st(intra)|1st(inter)|2nd(intra)|2nd(inter)|3rd(intra)|3rd(inter)|4th(intra)|4th(inter)|5th(intra)|5th(inter)|6th(intra)|6th(inter)|
> |-|-|-|-|-|-|-|-|-|-|-|-|-|-|
> |baseline|65.95|38.71|13.81|22.30|7.56|12.00|5.27|5.10|4.24|17.71|19.49|8.64|9.69|
> |ours(0.00)|68.14|31.21|12.02|18.18|7.61|9.98|4.28|3.94|3.25|16.82|18.69|8.30|9.32|
> |ours(0.08)|**68.21**|31.25|12.56|18.28|6.54|9.92|4.46|4.10|3.28|18.77|19.52|9.23|9.74|
> |ours(0.26)|67.68|36.28|13.11|20.93|7.15|10.94|4.92|4.58|3.81|17.72|19.08|8.72|9.50|
> |ours(0.81)|67.37|35.39|13.72|20.44|6.92|11.02|4.89|4.70|3.89|17.74|18.87|8.70|9.41|
> |ours(2.56)|67.34|35.40|13.40|20.61|7.32|10.83|4.84|4.56|3.74|17.92|19.08|8.78|9.50|
> |ours(8.11)|67.39|39.04|14.86|23.42|8.04|12.00|5.24|5.16|4.38|16.77|18.58|8.28|9.27|
>
> \
> \
> \
> \
> For 20 super-classes  (MobileNet):
> |$\tau$ (degree)|Acc|1st(intra)|1st(inter)|2nd(intra)|2nd(inter)|3rd(intra)|3rd(inter)|4th(intra)|4th(inter)|5th(intra)|5th(inter)|6th(intra)|6th(inter)|
> |-|-|-|-|-|-|-|-|-|-|-|-|-|-|
> |baseline|77.46|30.30|8.86|17.94|4.30|9.25|2.97|3.01|2.46|9.09|12.27|4.45|6.12|
> |ours (0.00)|77.54|30.36|8.65|17.95|4.43|9.34|2.99|3.22|2.56|9.69|12.57|4.73|6.27|
> |ours(0.08)|**77.72**|31.59|9.22|18.24|4.42|9.57|3.08|3.33|2.61|9.91|12.56|4.87|6.27|
> |ours(0.26)|77.36|32.04|9.80|18.67|4.58|9.57|3.07|3.28|2.70|9.85|12.69|4.83|6.33|
> |ours(0.81)|77.69|31.36|9.59|18.17|4.38|9.50|3.04|3.15|2.60|9.70|12.63|4.77|6.30|
> |ours(2.56)|77.31|31.69|9.38|18.56|4.36|9.39|3.03|3.30|2.68|9.94|12.55|4.88|6.26|
> |ours(8.11)|77.60|32.39|8.66|18.34|4.47|9.35|3.01|3.26|2.65|9.83|12.48|4.81|6.22|
>
> \
> \
> \
> \
> The results on 100-class and 20-superclasse show that GCR improves accuracy and maintain meaningful feature diversity.
>
> **While forcing minimum within-class variance to maintain at least a tiny angle $\tau=0.08$ helps achieve better results, the within-class variance for $\tau=0.00$ (ours) does not collapse in our experiments.** In fact, that variance is relatively close to the variance for $\tau=0.08$ (best case).
>
> **However, setting larger $\tau$ which leads to large intra-class variance being maintained leads to degradation of accuracy.**
>
> ### **We believe this experiment addresses the Reviewer's concern. Maintaining certain within-class feature variance by a soft penalty makes sense. At the same time, the dimensional collapse does not happen in our model even without that penalty.**
>
> It is trivial to use the extra penalty to ensure guardrails against the dimensional collapse.
>
> Warm regards,
>
> Authors

---

### Official Review · Reviewer_6b7B · 2025-07-03

**Clarity:** 3
**Significance:** 2
**Originality:** 2
**Rating:** 4
**Confidence:** 3

**Summary:**

The paper introduces Graph Consistency Regularization (GCR), a plug-and-play module that can be integrated into existing neural networks. GCR constructs relational graphs from intermediate representations to ensure these representations are consistent with the model’s final predictions.

**Questions:**

1. The main experimental datasets are CIFAR-100 and Tiny ImageNet, which are relatively small by current standards. Evaluation on larger datasets would improve the paper quality.

2. Could the authors elaborate on the additional computational overhead introduced by GCR? Can the method be implemented in a model-parallel fashion to support scalability on large datasets?

3. Have the authors conducted an ablation study to assess the impact of batch size on performance?

**Ethical Concerns:**

["NO or VERY MINOR ethics concerns only"]

**Final Justification:**

Based on the consistent performance improvement on different datasets, I recommend to borderline accept.

**Limitations:**

yes

**Quality:**

2

**Strengths And Weaknesses:**

The paper is clearly written and includes numerous visualizations to illustrate the learned relational graphs within GCR. The authors also provide a theoretical analysis for their approach.

However, the idea of using graphs to enhance classification tasks is not particularly novel, and the paper lacks comparisons with other graph-based classification methods. Additionally, the method appears to require a relatively large batch size (e.g., 128), which may limit its applicability. The datasets used—CIFAR-100 and Tiny ImageNet—are somewhat limited in scale for evaluating generalization.

---

> ### Author Rebuttal · Authors · 2025-07-30
>
> We sincerely thank reviewer for the constructive feedback, which has significantly improved the quality of our work.
>
> We have addressed all concerns and revised the paper accordingly.
>
> Specifically:
> (i) we added large-scale ImageNet-1K experiments using iFormer [3], ViT-B/16, and ViG-B [2] in Sections 3.1 and 3.2 to demonstrate the generalization of our method;
> (ii) we moved the computational overhead analysis from Appendix G (pp. 33–34) to Section 3.2 for greater visibility; and
> (iii) we added an ablation study on batch size effects for both CIFAR-10 and Tiny ImageNet.
>
> ### **1. Novelty**
>
> While graph-based methods have been studied, our GCR framework introduces **a fundamentally different mechanism**.
>
> Different from existing works, GCR **does not rely on static external graphs** or message passing as in GNNs. Instead, it introduces **a novel form of self-prompted regularization**, where the model’s own predictions dynamically construct a class-aware graph that supervises intermediate feature representations.
>
> This distinction has been clearly acknowledged by multiple reviewers:
>
> - Reviewer puhK praised the conceptual novelty, calling the use of prediction structures "innovative and well-motivated", and described GCR as "conceptually elegant", "versatile", and "broadly applicable in practice". He/She noted that our analyses "substantiate the soundness of the proposed method".
>
> - Reviewer bJ5m emphasized the value of the multi-layer mechanism: by aligning early representations with later prediction-derived relations, the method encourages "more discriminative patterns", which improve generalization.
>
> - Reviewer a6xh precisely summarized our contribution: GCR introduces "parameter-free GCLs" that align feature and prediction graphs, is "supported by solid theoretical foundations", and is clearly visualized through relational graphs.
>
> Key innovations of GCR includes:
>
> - **Self-prompted** graphs (dynamic & batch-wise): GCR constructs prediction graphs on-the-fly from the model's own softmax outputs, requiring no static structure or external memory.
>
> - **Parameter-free** regularization: GCR acts as a lightweight, plug-in regularizer, not a feature transformer, making it architecture-agnostic and **easy to integrate**.
>
> - **Cross-space graph alignment**: GCR aligns intermediate feature-space similarity graphs with prediction-space semantic graphs, enabling **a unique form of supervision** not yet explored in any prior graph-based classification methods.
>
> As detailed in Appendix A (Related Work, p. 24) and B (Relation to Existing Paradigms, pp. 24-26), this self-prompting strategy offers **a fresh perspective** on using relational inductive biases in a modular, model-driven way.
>
> We have revised the Related Work to highlight these points of novelty and more clearly position our contributions within the existing literature.
>
> ### **2. Comparison to existing graph-based methods**
>
> Below, we provide **direct comparisons** between our GCR-augmented models and recent graph-based classification methods across CIFAR-10, CIFAR-100, and ImageNet-1K. The results clearly show the consistent and complementary benefits of GCR when integrated into graph-based backbones.
>
> **Results on CIFAR-10, CIFAR-100, and ImageNet-1K**
>
> ||CIFAR-10|CIFAR-100|ImageNet-1k|
> |-|-|-|-|
> |ResNet18|||||
> |CNN2GNN[1]|95.51$\pm$0.42|74.80$\pm$0.81|60.12$\pm$1.02|
> |CNN2GNN **+GCR**|95.87$\pm$0.31|76.23$\pm$0.38|62.47$\pm$0.47|
> |CNN2Transformer[1]|95.79$\pm$0.24|77.39$\pm$0.20|71.12$\pm$0.35|
> |CNN2Transformer **+GCR**|**95.96$\pm$0.35**|**78.23$\pm$0.30**|**72.33$\pm$0.31**|
> |ResNet34|||||
> |CNN2GNN[1]|96.39$\pm$0.41|77.87$\pm$0.91|61.02$\pm$0.77|
> |CNN2GNN **+GCR**|96.67$\pm$0.36|78.14$\pm$0.54|62.88$\pm$0.46|
> |CNN2Transformer[1]|96.73$\pm$0.37|80.10$\pm$0.45|75.42$\pm$0.15|
> |CNN2Transformer **+GCR**|**96.97$\pm$0.36**|**81.27$\pm$0.29**|**76.67$\pm$0.26**|
>
> **Results on ImageNet-1K**
>
> |Model|ViG-Ti[2]|ViG-Ti **+GCR**|ViG-S[2]|ViG-S **+GCR**|ViG-B[2]|ViG-B **+GCR**|
> |-|-|-|-|-|-|-|
> |Accuracy|73.9|**74.9**|80.4|**81.7**|82.3|**84.0**|
>
>
> These results highlight key insights:
>
> - **Consistent** across architectures: GCR improves performance across diverse backbones (CNN2GNN, CNN2Transformer, ViG), demonstrating broad applicability.
>
> - **Complementary** to existing methods: Its self-regularization mechanism enhances CNN2GNN and CNN2Transformer, indicating **orthogonality to existing graph-based pipelines**.
>
> - **Scalable** to large-scale tasks: **Notable gains** on ImageNet-1K (e.g., +1.7% with ViG-B) show GCR’s effectiveness in complex, real-world settings.
>
> - **Lightweight** and model-agnostic: GCR introduces no extra parameters and integrates easily into existing architectures.
>
> Together, these findings reinforce GCR’s **novelty and practical value** as a flexible, self-prompted regularization framework applicable across scales and architectures.
>
> ### **3. Generalization to large-scale datasets**
>
> We thank the reviewer for suggesting an evaluation on a larger dataset. In response to the suggestion, we conducted additional experiments on **ImageNet-1K** using transformer-based architectures (iFormer [3], ViT, and ViG [2]). Results, averaged over three runs, are summarized in Appendix H.3.
>
> ||iFormer-S|iFormer-B|ViT-B/16|ViG-B|
> |:-|:-:|:-:|:-:|:-:|
> |Baseline|83.4$\pm$0.40|84.6$\pm$0.45|74.3$\pm$0.51|82.3$\pm$0.42|
> |Early GCL|83.8$\pm$0.31|85.0$\pm$0.40|74.7$\pm$0.44|82.8$\pm$0.35|
> |Mid GCL|83.8$\pm$0.39|85.5$\pm$0.33|75.2$\pm$0.36|83.0$\pm$0.34|
> |Late GCL|84.5$\pm$0.29|**86.1$\pm$0.30**|**75.8$\pm$0.33**|**84.0$\pm$0.30**|
> |Early+Mid|84.3$\pm$0.33|85.9$\pm$0.38|75.6$\pm$0.41|83.7$\pm$0.33|
> |Mid+Late|**84.8$\pm$0.28**|85.9$\pm$0.37|75.6$\pm$0.34|83.9$\pm$0.30|
> |Early+Late|84.5$\pm$0.30|85.2$\pm$0.28|74.9$\pm$0.33|83.5$\pm$0.29|
> |Full GCL|84.3$\pm$0.29|85.8$\pm$0.26|75.5$\pm$0.30|83.6$\pm$0.27|
>
> Key results:
>
> - iFormer-S: GCR boosts performance from **83.4%** to **84.8%** (a **+1.4%** gain).
>
> - iFormer-B: GCR improves accuracy from **84.6%** to **86.1%** (a **+1.5%** gain).
>
> - Similar gains are observed with ViT-B/16 and ViG-B, confirming generality.
>
> These improvements validate that GCR **scales effectively** to large, complex datasets and modern architectures. They support our core hypothesis: aligning feature geometry with prediction semantics enhances generalization. Moreover, GCR achieves this without architectural changes or added parameters, reinforcing its value as a model-agnostic and lightweight regularizer.
>
> These new results directly address the reviewer’s concern and further strengthen the overall contribution of the paper.
>
> ### **4. Computational overhead and scalability**
>
> We thank the reviewer for highlighting this point. To improve clarity, we have moved the computational overhead analysis from Appendix G (pp. 34 line 1275-1284 and 1292-1297) into the main paper.
>
> - **Overhead**: GCR adds modest training time due to graph construction and alignment (e.g., MobileNet: +15 min, ResNeXt-101: +120 min on CIFAR-100). These operations are efficient and introduce no additional trainable parameters.
>
> - **Scalability**: GCR is highly scalable. All graph computations are performed per batch and are fully parallelizable across GPUs using standard data-parallel training. The operations rely on matrix computations optimized for modern hardware, ensuring smooth integration into large-scale pipelines.
>
> ### **5. Impact of batch size**
>
> As shown in Figure 8 (Appendix H.2, p. 36), GCR **remains effective across a wide range of batch sizes**. Even with small batches (e.g., n=16 and n=32), GCL-enhanced models produce more coherent intra-class clusters and stronger inter-class separation than the baseline. This supports our claim that GCR can extract meaningful structure even from limited relational signals.
>
> While larger batches amplify gains by providing denser graphs, they are not essential. The tables below confirm consistent performance improvements across batch sizes on both CIFAR-10 (ShuffleNet) and Tiny ImageNet (CeiT):
>
> **ShuffleNet on CIFAR-10**
> |Batch Size|+GCR|Baseline|
> |-|-|-|
> |16|79.90$\pm$0.38|78.88$\pm$0.41|
> |32|87.91$\pm$0.36|86.91$\pm$0.37|
> |64|91.26$\pm$0.25|90.64$\pm$0.35|
> |128|92.79$\pm$0.20|91.21$\pm$0.28|
> |256|**92.89$\pm$0.25**|92.07$\pm$0.27|
> |512|92.33$\pm$0.23|91.94$\pm$0.25|
>
> **CeiT on Tiny ImageNet**
> |Batch Size|+GCR|Baseline|
> |-|-|-|
> |16|44.84$\pm$0.31|43.78$\pm$0.35|
> |32|47.55$\pm$0.29|46.89$\pm$0.31|
> |64|49.19$\pm$0.25|48.09$\pm$0.30|
> |128|**51.22$\pm$0.20**|49.95$\pm$0.29|
> |256|50.77$\pm$0.19|49.62$\pm$0.24|
> |512|50.65$\pm$0.22|49.34$\pm$0.24|
>
> These results affirm GCR’s robustness and flexibility, even in resource-constrained or small-batch training setups.
>
> We are confident that GCR represents a novel, technically sound, and broadly applicable contribution to the field.
>
> We sincerely thank the reviewer for the valuable feedback, which we **have incorporated** into the final version of the paper.
>
> ### References
>
> [1] Trivedy, V., & Latecki, L. J. (2023). Cnn2graph: Building graphs for image classification. In Proceedings of the IEEE/CVF Winter Conference on Applications of Computer Vision (pp. 1-11).
>
> [2] Han, K., Wang, Y., Guo, J., Tang, Y., & Wu, E. (2022). Vision gnn: An image is worth graph of nodes. Advances in neural information processing systems, 35, 8291-8303.
>
> [3] Si, C., Yu, W., Zhou, P., Zhou, Y., Wang, X., & Yan, S. (2022). Inception transformer. *Advances in Neural Information Processing Systems*, *35*, 23495-23509.

---

> > ### Comment · Reviewer_6b7B · 2025-08-04
> >
> > Thank you for the thoughtful rebuttal regarding the large-scale ImageNet-1K experiments and the effect of batch size; it satisfactorily addressed my concerns. Could you also provide a theoretical analysis of the time complexity introduced by GCR?

---

> ### Author Response · Authors · 2025-08-04
> **Further clarifications.**
>
> Esteemed Reviewer,
> \
> \
> We thank the Esteemed Reviewer for engaging with our rebuttal, and for the insightful question regarding the computational complexity introduced by GCR. Please rest assured all requested changes will be included by us into the paper.
> \
> \
> Below, we present a theoretical analysis of the GCR’s time complexity per training iteration, from both a naive computational perspective and an optimized parallel execution view. We will include this analysis into the paper.
>
>
> ## 1. **We have the following individual costs:**
> ### 1.1. Feature graph construction.
>
> At each layer $l=1,\ldots,K$ where a GCL is applied, a feature similarity graph $F^{(l)} \in \mathbb{R}^{n \times n}$ is constructed using the cosine similarity.
>
> We have **Time Complexity**:
> - **Naive (sequential compute)**: Normalizing all $n$ feature vectors costs $\mathcal{O}(nd)$; pairwise cosine similarities require $\mathcal{O}(n^2 d)$.
> - **GPU-parallelized**: With sufficient vector-level parallelism, normalization and similarity computations can be reduced to $\mathcal{O}(\log d)$, assuming parallel dot products.
> - Thus, over the **Total over $K$ layers**: $\mathcal{O}(K \cdot \log d_{\text{max}})$, where $d_{\text{max}}$ is the largest feature dimension across all GCL layers.
>
>
> ### 1.2. Prediction graph construction.
>
> The prediction graph $P \in \mathbb{R}^{n \times n}$ is derived from softmax-normalized logits $z_i \in \mathbb{R}^C$.
>
> We have **Time Complexity**:
> - **Naive (sequential compute)**: Softmax computation costs $\mathcal{O}(nC)$, cosine similarities $\mathcal{O}(n^2 C)$, and masking $\mathcal{O}(n^2)$.
> - **GPU-parallelized**: Per-sample operations reduce to $\mathcal{O}(\log C)$, and masking becomes $\mathcal{O}(1)$ due to element-wise matrix operations.
>
> ### 1.3. Graph alignment loss.
>
> The loss at each layer measures the Frobenius norm of the difference between graphs: $\mathcal{L}_{\text{GCR}}^{(l)} = \left|| \text{triu}(F^{(l)} - P) \right||_F^2$.
>
> We have **Time Complexity**:
> - **Naive (sequential compute)**: $\mathcal{O}(n^2)$
> - **GPU-Parallelized**: $\mathcal{O}(\log n)$, assuming reduction over parallel threads aggregation for the norm computation.
>
> ### 1.4. Adaptive weighting across layers.
>
> If adaptive weighting (Eq. 6) is used, we compute normalized weights for each layer based on alignment discrepancy.
>
> We have **Time Complexity**: $\mathcal{O}(K n^2)$, where $K$ is the number of GCL-applied layers.
>
>
>
> ## 2. **Total time complexity.**
>
> - **Naive (sequential compute)**: Assuming GCLs are applied at $K$ layers, with $d_{\text{max}}$ being the maximum feature dimension and $C$ the number of classes: $\mathcal{O} \left( K\cdot n^2 (d_{\text{max}} + C) \right)$
>
>   The dominant term is $\mathcal{O}(n^2 d_{\text{max}})$, due to high-dimensional pairwise feature similarity computations in deeper layers.
>
> - **GPU-Parallelized**: a more realistic parallel compute has complexity $\mathcal{O} \left( K\cdot  (\log d_{\text{max}} + \log C) \right)$ where $\log C \ll \log d_{\text{max}}$, so $\log C$ can be ignored in the final cost.
>
>   If  $n^2$ is very large to fit entirely into memory (we did not run in such a case), we could split it into $s=4$ (or so) parallel blocks processed sequentially in $s$ steps.
>
> ## 3. **Practical considerations and optimizations.**
>
> **Scalability**: GCR operates on batches, not datasets. Its quadratic cost in $n$ (e.g., $n = 128$) is modest in practice.
>
> **Parallel efficiency**: All computations are matrix-based and benefit from hardware acceleration. Libraries like PyTorch exploit thread and GPU-level parallelism to accelerate operations such as `torch.bmm`, `functional.cosine_similarity` and `torch.triu`.
>
> **Zero parameter overhead**: GCR introduces no trainable parameters. It does not affect memory footprint or the gradient flow, keeping it efficient and architecture-agnostic.
>
> GCR introduces a lightweight yet effective form of structure-based regularization with a per-layer complexity of no more than $\mathcal{O}(\log d)$.
>
> Thanks to batch-local operation, GPU-friendly computations, and absence of learnable parameters, GCR scales well and improves semantic alignment without compromising efficiency.

---

> ### Comment · Reviewer_6b7B · 2025-08-07
>
> Thanks for your clarification. Based on the consistent performance improvement on different datasets, I will raise my score accordingly.

---

> > ### Author Response · Authors · 2025-08-07
> > **thank you**
> >
> > Esteemed Reviewer,
> >
> > We sincerely thank you for engaging with our rebuttal, and for your valuable comments. We will incorporate all your suggestions and our results accordingly into the final draft. Meantime, kindly let us know if there is anything else we can answer, clarify or improve.
> >
> > Best regards,
> > Authors

---

### Comment · Area_Chair_uHQf · 2025-08-05

Dear Reviewers,
This is a friendly reminder to please engage in the author discussion phase if you haven't already. Your responses to author replies are important and appreciated. The discussion period ends on Aug 6, so we encourage you toparticipate soon. Thank you for your contributions to NeurlPS 2025.

Best regards.
AC

---

### Decision · Program_Chairs · 2025-09-17

**Decision:**

Accept (poster)

**Comment:**

This paper proposes Graph Consistency Regularization (GCR), a lightweight and parameter-free framework that aligns intermediate feature similarity graphs with prediction-based graphs to enhance semantic structure and generalization across architectures. Initially, reviewers appreciated the clarity, theoretical grounding, and versatility of the method, but raised concerns about novelty, reliance on its core assumption, limited large-scale evaluation, and small observed improvements. The authors provided extensive rebuttals including new ImageNet-1K experiments, theoretical complexity analysis, variance across multiple runs, quantitative evidence of reduced inter-class noise, and proofs with anti-collapse mechanisms, which satisfactorily addressed most concerns. Overall, I recommend acceptance as the paper is technically solid and broadly applicable, though future work could more explicitly examine its limitations in noisy or imbalanced settings and explore extensions to unsupervised learning.